# A Decomposition of Forecast Error in Prediction Markets

**Miroslav Dudík**
Microsoft Research, New York, NY
mdudik@microsoft.com

**Sébastien Lahaie**
Google, New York, NY
slahaie@google.com

**Ryan Rogers**
University of Pennsylvania, Philadelphia, PA
rrogers386@gmail.com

**Jennifer Wortman Vaughan**
Microsoft Research, New York, NY
jenn@microsoft.com

## Abstract

We analyze sources of error in prediction market forecasts in order to bound the difference between a security's price and the ground truth it estimates. We consider cost-function-based prediction markets in which an automated market maker adjusts security prices according to the history of trade. We decompose the forecasting error into three components: *sampling error*, arising because traders only possess noisy estimates of ground truth; *market-maker bias*, resulting from the use of a particular market maker (i.e., cost function) to facilitate trade; and *convergence error*, arising because, at any point in time, market prices may still be in flux. Our goal is to make explicit the tradeoffs between these error components, influenced by design decisions such as the functional form of the cost function and the amount of liquidity in the market. We consider a specific model in which traders have exponential utility and exponential-family beliefs representing noisy estimates of ground truth. In this setting, sampling error vanishes as the number of traders grows, but there is a tradeoff between the other two components. We provide both upper and lower bounds on market-maker bias and convergence error, and demonstrate via numerical simulations that these bounds are tight. Our results yield new insights into the question of how to set the market's liquidity parameter and into the forecasting benefits of enforcing coherent prices across securities.

## 1 Introduction

A prediction market is a marketplace in which participants can trade securities with payoffs that depend on the outcomes of future events [19]. Consider the simple setting in which we are interested in predicting the outcome of a political election: whether the incumbent or challenger will win. A prediction market might issue a security that pays out $1 per share if the incumbent wins, and $0 otherwise. The market price $p$ of this security should always lie between 0 and 1, and can be construed as an event probability. If a trader believes that the likelihood of the incumbent winning is greater than $p$, she will buy shares with the expectation of making a profit. Market prices increase when there is more interest in buying and decrease when there is more interest in selling. By this process, the market aggregates traders' information into a consensus forecast, represented by the market price. With sufficient activity, prediction markets are competitive with alternative forecasting methods such as polls [4], but while there is a mature literature on sources of error and bias in polls, the impact of prediction market structure on forecast accuracy is still an active area of research [17].

We consider prediction markets in which all trades occur through a centralized entity known as a *market maker*. Under this market structure, security prices are dictated by a fixed *cost function* and

the current number of outstanding shares [6]. The basic conditions that a cost function should satisfy to correctly elicit beliefs, while bounding the market maker's loss, are now well-understood, chief among them being convexity [1]. Nonetheless, the class of allowable cost functions remains broad, and the literature so far provides little formal guidance on the specific form of cost function to use in order to achieve good forecast accuracy, including how to set the *liquidity parameter* which controls price responsiveness to trade. In practice, the impact of the liquidity parameter is difficult to quantify a priori, so implementations typically resort to calibrations based on market simulations [8, 18]. Prior work also suggests that maintaining coherence among prices of logically related securities has informational advantages [8], but there has been little work aimed at understanding why.

This paper provides a framework to quantify the impact of the choice of cost function on forecast accuracy. We introduce a decomposition of forecast error, in analogy with the bias-variance decomposition familiar from statistics or the approximation-estimation-optimization decomposition for large-scale machine learning [5]. Our decomposition consists of three components. First, there is the *sampling error* resulting from the fact that the market consists of a finite population of traders, each holding a noisy estimate of ground truth. Second, there is a *market-maker bias* which stems from the use of a cost function to provide liquidity and induce trade. Third, there is *convergence error* due to the fact that the market prices may not have fully converged to their equilibrium point.

The central contribution of this paper is a theoretical characterization of the market-maker bias and convergence error, the two components of this decomposition that depend on market structure as defined by the form of the cost function and level of liquidity. We consider a tractable model of agent behavior, originally studied by Abernethy et al. [2], in which traders have exponential utility functions and beliefs drawn from an exponential family. Under this model it is possible to characterize the market's equilibrium prices in terms of the traders' belief and risk aversion parameters, and thereby quantify the discrepancy between current market prices and ground truth. To analyze market convergence, we consider the trader dynamics introduced by Frongillo and Reid [9], under which trading can be viewed as randomized block-coordinate descent on a suitable potential function.

Our analysis is *local* in that the bounds depend on the market equilibrium prices. This allows us to exactly identify the main asymptotic terms of error. We demonstrate via numerical experiments that these asymptotic bounds are accurate early on and therefore can be used to compare market designs.

We make the following specific contributions:

1. We precisely define the three components of the forecasting error.

2. We show that the market-maker bias equals $cb \pm O(b^2)$ as $b \to 0$, where $b$ is the liquidity parameter, and $c$ is an explicit constant that depends on the cost function and trader beliefs.

3. We show that the convergence error decreases with the number of trades $t$ as $\gamma^t$ with $\gamma = 1 - \Theta(b)$. We provide explicit upper and lower bounds on $\gamma$ that depend on the cost function and trader beliefs. In the process, we prove a new local convergence bound for block-coordinate descent.

4. We use our explicit formulas for bias and convergence error to compare two common cost functions: independent markets (IND), under which security prices vary independently, and the logarithmic market scoring rule (LMSR) [10], which enforces logical relationships between security prices. We show that at the same value of the market-maker bias, IND requires at least half-as-many and at most twice-as-many trades as LMSR to achieve the same convergence error.

We consider a specific utility model (exponential utility), but our bias and convergence analysis immediately carry over if we assume that each trader is optimizing a risk measure (rather than an exponential utility function) similar to the setup of Frongillo and Reid [9]. Exponential utility was chosen because it was previously well studied and allowed us to focus on the analysis of the cost function and liquidity. The role of the liquidity parameter in trading off the bias and convergence error has been informally recognized in the literature [7, 10, 13], but our precise definition of market-maker bias and explicit formulas for the bias and convergence error are novel. Abernethy et al. [2] provide results that can be used to derive the bias for LMSR, but not for generic cost functions, so they do not enable comparison of biases of different costs. Frongillo and Reid [9] observe that the convergence error can be locally bounded as $\gamma^t$, but they only provide an upper bound and do not show how $\gamma$ is related to the liquidity or cost function. Our analysis establishes both upper and lower bounds on convergence and relates $\gamma$ explicitly to the liquidity and cost function. This is necessary for a

meaningful comparison of cost function families. Thus our framework provides the first meaningful way to compare the error tradeoffs inherent in different choices of cost functions and liquidity levels.

## 2 Preliminaries

We use the notation $[N]$ to denote the set $\{1, \ldots, N\}$. Given a convex function $f : \mathbb{R}^d \to \mathbb{R} \cup \{\infty\}$, its *effective domain*, denoted $\operatorname{dom} f$, is the set of points where $f$ is finite. Whenever $\operatorname{dom} f$ is non-empty, the *conjugate* $f^* : \mathbb{R}^d \to \mathbb{R} \cup \{\infty\}$ is defined by $f^*(\boldsymbol{v}) := \sup_{\boldsymbol{u} \in \mathbb{R}^d}[\boldsymbol{v}^\mathsf{T} \boldsymbol{u} - f(\boldsymbol{u})]$. We write $\|\cdot\|$ for the Euclidean norm. A centralized mathematical reference is provided in Appendix A.[1]

**Cost-function-based market makers** We study cost-function-based prediction markets [1]. Let $\Omega$ be a finite set of mutually exclusive and exhaustive states of the world. A market administrator, known as *market maker*, wishes to elicit information about the likelihood of various states $\omega \in \Omega$, and to that end offers to buy and sell any number of shares of $K$ *securities*. Securities are associated with coordinates of a payoff function $\boldsymbol{\phi} : \Omega \to \mathbb{R}^K$, where each share of the $k$th security is worth $\phi_k(\omega)$ in the event that the true state of the world is $\omega \in \Omega$. Traders arrive in the market sequentially and trade with the market maker. The market price is fully determined by a convex potential function $C$ called the *cost function*. In particular, if the market maker has previously sold $s_k \in \mathbb{R}$ shares of each security $k$ and a trader would like to purchase a bundle consisting of $\delta_k \in \mathbb{R}$ shares of each, the trader is charged $C(\boldsymbol{s} + \boldsymbol{\delta}) - C(\boldsymbol{s})$. The *instantaneous price* of security $k$ is then $\partial C(\boldsymbol{s})/\partial s_k$. Note that negative values of $\delta_k$ are allowed and correspond to the trader (short) selling security $k$.

Let $\mathcal{M} := \operatorname{conv}\{\boldsymbol{\phi}(\omega) : \omega \in \Omega\}$ be the convex hull of the set of payoff vectors. It is exactly the set of expectations $\mathbb{E}[\boldsymbol{\phi}(\omega)]$ across all possible probability distributions over $\Omega$, which we call *beliefs*. We refer to elements of $\mathcal{M}$ as *coherent prices*. Abernethy et al. [1] characterize the conditions that a cost function must satisfy in order to guarantee important properties such as bounded loss for the market maker and no possibility of arbitrage. To start, we assume only that $C : \mathbb{R}^K \to \mathbb{R}$ is convex and differentiable and that $\mathcal{M} \subseteq \operatorname{dom} C^*$, which corresponds to the bounded loss property.

**Example 2.1** (Logarithmic Market Scoring Rule: LMSR [10]). Consider a *complete market* with a single security for each outcome worth \$1 if that outcome occurs and \$0 otherwise, i.e., $\Omega = [K]$ and $\phi_k(\omega) = \mathbf{1}\{k = \omega\}$ for all $k$. The LMSR cost function and instantaneous security prices are given by

$$C(\boldsymbol{s}) = \log\left(\sum_{k=1}^K e^{s_k}\right) \qquad \text{and} \qquad \frac{\partial C(\boldsymbol{s})}{\partial s_k} = \frac{e^{s_k}}{\sum_{\ell=1}^K e^{s_\ell}}, \ \forall k \in [K]. \tag{1}$$

Its conjugate is the entropy function, $C^*(\boldsymbol{\mu}) = \sum_k \mu_k \log \mu_k + \mathbb{I}\{\boldsymbol{\mu} \in \Delta_K\}$, where $\Delta_K$ is the simplex in $\mathbb{R}^K$ and $\mathbb{I}\{\cdot\}$ is the convex indicator, equal to zero if its argument is true and infinity if false. Thus, in this case $\mathcal{M} = \Delta_K = \operatorname{dom} C^*$.

Notice that the LMSR security prices are coherent because they always sum to one. This prevents arbitrage opportunities for traders. Our second running example does not have this property.

**Example 2.2** (Sum of Independent LMSRs: IND). Let $\Omega = [K]$ and $\phi_k(\omega) = \mathbf{1}\{k = \omega\}$ for all $k$. The cost function and instantaneous security prices for the *sum of independent LMSRs* are given by

$$C(\boldsymbol{s}) = \sum_{k=1}^K \log\left(1 + e^{s_k}\right) \qquad \text{and} \qquad \frac{\partial C(\boldsymbol{s})}{\partial s_k} = \frac{e^{s_k}}{1 + e^{s_k}}, \ \forall k \in [K], \tag{2}$$

$C^*(\boldsymbol{\mu}) = \sum_k[\mu_k \log \mu_k + (1-\mu_k)\log(1-\mu_k)] + \mathbb{I}\{\boldsymbol{\mu} \in [0,1]^K\}$, $\mathcal{M} = \Delta_K$, and $\operatorname{dom} C^* = [0,1]^K$.

When choosing a cost function, one important consideration is *liquidity*, that is, how quickly prices change in response to trades. Any cost function $C$ can be viewed as a member of a parametric family of cost functions of the form $C_b(\boldsymbol{s}) := bC(\boldsymbol{s}/b)$ across all $b > 0$. With larger values of $b$, larger trades are required to move market prices by some fixed amount, and the worst-case loss of the market maker is larger; with smaller values, small purchases can result in big changes to the market price.

**Basic model** In our analysis of error we assume that there exists an unknown true probability distribution $\boldsymbol{p}^{\text{true}} \in \Delta_{|\Omega|}$ over the outcome set $\Omega$. The true expected payoffs of the $K$ market securities are then given by the vector $\boldsymbol{\mu}^{\text{true}} := \mathbb{E}_{\omega \sim \boldsymbol{p}^{\text{true}}}[\boldsymbol{\phi}(\omega)]$.

We assume that there are $N$ traders and that each trader $i \in [N]$ has a private belief $\tilde{\boldsymbol{p}}_i$ over outcomes. We additionally assume that each trader $i$ has a *utility function* $u_i : \mathbb{R} \to \mathbb{R}$ for wealth and would like to maximize expected utility subject to her beliefs. For now we assume that $u_i$ is differentiable and concave, meaning that each trader is risk averse, though later we focus on exponential utility. The expected utility of trader $i$ owning a security bundle $\boldsymbol{r}_i \in \mathbb{R}^K$ and cash $c_i$ is $U_i(\boldsymbol{r}_i, c_i) := \mathbb{E}_{\omega \sim \tilde{\boldsymbol{p}}_i} \left[ u_i \big( c_i + \boldsymbol{\phi}(\omega) \cdot \boldsymbol{r}_i \big) \right]$. We assume that each trader begins with zero cash. This is without loss of generality because we could incorporate any initial cash holdings into $u_i$.

# 3 A Decomposition of Error

In this section, we decompose the market's forecast error into three major components. The first is *sampling error*, which arises because traders have only noisy observations of the ground truth. The second is *market-maker bias*, which arises because the shape of the cost function impacts the traders' willingness to invest. Finally, *convergence error* arises due to the fact that at any particular point in time the market prices may not have fully converged. To formalize our decomposition, we introduce two new notions of equilibrium.

Our first notion of equilibrium, called a *market-clearing equilibrium*, does not assume the existence of a market maker, but rather assumes that traders trade only among themselves, and so no additional securities or cash are available beyond the traders' initial allocations. This equilibrium is described by security prices $\bar{\boldsymbol{\mu}} \in \mathbb{R}^K$ and allocations $(\bar{\boldsymbol{r}}_i, \bar{c}_i)$ of security bundles and cash to each trader $i$ such that, given her allocation, no trader wants to buy or sell any bundle of securities at those prices. Trader bundles and cash are summarized as $\bar{\boldsymbol{r}} = (\bar{\boldsymbol{r}}_i)_{i \in [N]}$ and $\bar{\boldsymbol{c}} = (\bar{c}_i)_{i \in [N]}$.

**Definition 3.1** (Market-clearing equilibrium). A triple $(\bar{\boldsymbol{r}}, \bar{\boldsymbol{c}}, \bar{\boldsymbol{\mu}})$ is a market-clearing equilibrium if $\sum_{i=1}^{N} \bar{\boldsymbol{r}}_i = \boldsymbol{0}$, $\sum_{i=1}^{N} \bar{c}_i = 0$, and for all $i \in [N]$, $\boldsymbol{0} \in \arg\max_{\boldsymbol{\delta} \in \mathbb{R}^K} U_i(\bar{\boldsymbol{r}}_i + \boldsymbol{\delta}, \bar{c}_i - \boldsymbol{\delta} \cdot \bar{\boldsymbol{\mu}})$. We call $\bar{\boldsymbol{\mu}}$ *market-clearing prices* if there exist $\bar{\boldsymbol{r}}$ and $\bar{\boldsymbol{c}}$ such that $(\bar{\boldsymbol{r}}, \bar{\boldsymbol{c}}, \bar{\boldsymbol{\mu}})$ is a market-clearing equilibrium. Similarly, we call $\bar{\boldsymbol{r}}$ a *market-clearing allocation* if there exists a corresponding equilibrium.

The requirements on $\sum_{i=1}^{N} \bar{\boldsymbol{r}}_i$ and $\sum_{i=1}^{N} \bar{c}_i$ guarantee that no additional securities or cash have been created. In other words, there exists some set of trades among traders that would lead to the market-clearing allocation, although the definition says nothing about how the equilibrium is reached.

Since we rely on a market maker to orchestrate trade, our markets generally do not reach the market-clearing equilibrium. Instead, we introduce the notion of *market-maker equilibrium*. This equilibrium is again described by a set of security prices $\boldsymbol{\mu}^\star$ and trader allocations $(\boldsymbol{r}_i^\star, c_i^\star)$, summarized as $(\boldsymbol{r}^\star, \boldsymbol{c}^\star)$, such that no trader wants to trade at these prices given her allocation. The difference is that we now require $\boldsymbol{r}^\star$ and $\boldsymbol{c}^\star$ to be reachable via some sequence of trade with the market maker instead of via trade among only the traders, and $\boldsymbol{\mu}^\star$ must be the market prices after such a sequence of trade.

**Definition 3.2** (Market-maker equilibrium). A triple $(\boldsymbol{r}^\star, \boldsymbol{c}^\star, \boldsymbol{\mu}^\star)$ is a market-maker equilibrium for cost function $C_b$ if, for the market state $\boldsymbol{s}^\star = \sum_{i=1}^{N} \boldsymbol{r}_i^\star$, we have $\sum_{i=1}^{N} c_i^\star = C_b(\boldsymbol{0}) - C_b(\boldsymbol{s}^\star)$, $\boldsymbol{\mu}^\star = \nabla C_b(\boldsymbol{s}^\star)$, and for all $i \in [N]$, $\boldsymbol{0} \in \arg\max_{\boldsymbol{\delta} \in \mathbb{R}^K} U_i\big(\boldsymbol{r}_i^\star + \boldsymbol{\delta}, c_i^\star - C_b(\boldsymbol{s}^\star + \boldsymbol{\delta}) + C_b(\boldsymbol{s}^\star)\big)$. We call $\boldsymbol{\mu}^\star$ *market-maker equilibrium prices* if there exist $\boldsymbol{r}^\star$ and $\boldsymbol{c}^\star$ such that $(\boldsymbol{r}^\star, \boldsymbol{c}^\star, \boldsymbol{\mu}^\star)$ is a market-maker equilibrium. Similarly, we call $\boldsymbol{r}^\star$ a *market-maker equilibrium allocation* if there exists a corresponding equilibrium. We sometimes write $\boldsymbol{\mu}^\star(b; C)$ to show the dependence of $\boldsymbol{\mu}^\star$ on $C$ and $b$.

The market-clearing prices $\bar{\boldsymbol{\mu}}$ and the market-maker equilibrium prices $\boldsymbol{\mu}^\star(b; C)$ are not unique in general, but are unique for the specific utility functions that we study in this paper.

Using these notions of equilibrium, we can formally define our error components. Sampling error is the difference between the true security values and the market-clearing equilibrium prices. The bias is the difference between the market-clearing equilibrium prices and the market-maker equilibrium prices. Finally, the convergence error is the difference between the market-maker equilibrium prices and the market prices $\boldsymbol{\mu}^t(b; C)$ at a particular round $t$. Putting this together, we have that

$$\boldsymbol{\mu}^{\text{true}} - \boldsymbol{\mu}^t(b; C) = \underbrace{\boldsymbol{\mu}^{\text{true}} - \bar{\boldsymbol{\mu}}}_{\text{Sampling Error}} + \underbrace{\bar{\boldsymbol{\mu}} - \boldsymbol{\mu}^\star(b; C)}_{\text{Bias}} + \underbrace{\boldsymbol{\mu}^\star(b; C) - \boldsymbol{\mu}^t(b; C)}_{\text{Convergence Error}}. \tag{3}$$

# 4 The Exponential Trader Model

For the remainder of the paper, we work with the exponential trader model introduced by Abernethy et al. [2] in which traders have exponential utility functions and exponential-family beliefs. Under this model, both the market-clearing prices and market-maker equilibrium prices are unique and can be expressed cleanly in terms of potential functions [9], yielding a tractable analysis. The results of this section are immediate consequences of prior work [2, 9], but our equilibrium concepts bring them into a common framework.

We consider a specific *exponential family* [3] of probability distributions over $\Omega$ defined as $p(\omega; \boldsymbol{\theta}) = e^{\phi(\omega) \cdot \boldsymbol{\theta} - T(\boldsymbol{\theta})}$, where $\boldsymbol{\theta} \in R^K$ is the *natural parameter* of the distribution, and $T$ is the *log partition function*, $T(\boldsymbol{\theta}) := \log\left(\sum_{\omega \in \Omega} e^{\phi(\omega) \cdot \boldsymbol{\theta}}\right)$. The gradient $\nabla T(\boldsymbol{\theta})$ coincides with the expectation of $\phi$ under $p(\cdot; \boldsymbol{\theta})$, and $\operatorname{dom} T^* = \operatorname{conv}\{\phi(\omega) : \omega \in \Omega\} = \mathcal{M}$.

Following Abernethy et al. [2], we assume that each trader $i$ has exponential-family beliefs with natural parameter $\tilde{\boldsymbol{\theta}}_i$. From the perspective of trader $i$, the expected payoffs of the $K$ market securities can then be expressed as the vector $\tilde{\boldsymbol{\mu}}_i$ with $\tilde{\mu}_{i,k} := \sum_{\omega \in \Omega} \phi_k(\omega) p(\omega; \tilde{\boldsymbol{\theta}}_i)$.

As in Abernethy et al. [2], we also assume that traders are risk averse with exponential utility for wealth, so the utility of trader $i$ for wealth $W$ is $u_i(W) = -(1/a_i)e^{-a_i W}$, where $a_i$ is the the trader's risk aversion coefficient. We assume that the traders' risk aversion coefficients are fixed.

Using the definitions of the expected utility $U_i$, the exponential family distribution $p(\cdot; \tilde{\boldsymbol{\theta}}_i)$, the log partition function $T$, and the exponential utility $u_i$, it is straightforward to show [2] that

$$U_i(\boldsymbol{r}_i, c_i) = -\frac{1}{a_i} e^{-T(\tilde{\boldsymbol{\theta}}_i) - a_i c_i} \sum_{\omega \in \Omega} e^{\phi(\omega) \cdot (\tilde{\boldsymbol{\theta}}_i - a_i \boldsymbol{r}_i)} = -\frac{1}{a_i} e^{T(\tilde{\boldsymbol{\theta}}_i - a_i \boldsymbol{r}_i) - T(\tilde{\boldsymbol{\theta}}_i) - a_i c_i}. \quad (4)$$

Under this trader model, we can use the techniques of Frongillo and Reid [9] to construct potential functions which yield alternative characterizations of the equilibria as solutions of minimization problems. Consider first a market-clearing equilibrium. Define $F_i(\boldsymbol{s}) := \frac{1}{a_i} T(\tilde{\boldsymbol{\theta}}_i + a_i \boldsymbol{s})$ for each trader $i$. From Eq. (4) we can observe that $-F_i(-\boldsymbol{r}_i) + c_i$ is a monotone transformation of trader $i$'s utility. Since each trader's utility is locally maximized at a market-clearing equilibrium, the sum of traders' utilities is also locally maximized, as is $\sum_{i=1}^{N}(-F_i(-\boldsymbol{r}_i) + c_i)$. Since the equilibrium conditions require that $\sum_{i=1}^{N} c_i = 0$, the security allocation associated with any market-clearing equilibrium must be a local minimum of $\sum_{i=1}^{N} F_i(-\boldsymbol{r}_i)$. This idea is formalized in the following theorem. The proof follows from an analysis of the KKT conditions of the equilibrium. (See the appendix for all omitted proofs.)

**Theorem 4.1.** *Under the exponential trader model, a market-clearing equilibrium always exists and market-clearing prices are unique. Market-clearing allocations and prices are exactly the solutions of the following optimization problems:*

$$\bar{\boldsymbol{r}} \in \operatorname*{argmin}_{\boldsymbol{r}: \sum_{i=1}^{N} \boldsymbol{r}_i = 0} \left[ \sum_{i=1}^{N} F_i(-\boldsymbol{r}_i) \right], \qquad \bar{\boldsymbol{\mu}} = \operatorname*{argmin}_{\boldsymbol{\mu} \in \mathbb{R}^K} \left[ \sum_{i=1}^{N} F_i^*(\boldsymbol{\mu}) \right]. \quad (5)$$

Using a similar argument, we can show that the allocation associated with any market-maker equilibrium is a local minimum of the function $F(\boldsymbol{r}) := \sum_{i=1}^{N} F_i(-\boldsymbol{r}_i) + C_b\left(\sum_{i=1}^{N} \boldsymbol{r}_i\right)$.

**Theorem 4.2.** *Under the exponential trader model, a market-maker equilibrium always exists and equilibrium prices are unique. Market-maker equilibrium allocations and prices are exactly the solutions of the following optimization problems:*

$$\boldsymbol{r}^\star \in \operatorname*{argmin}_{\boldsymbol{r}} F(\boldsymbol{r}), \qquad \boldsymbol{\mu}^\star = \operatorname*{argmin}_{\boldsymbol{\mu} \in \mathbb{R}^K} \left[ \sum_{i=1}^{N} F_i^*(\boldsymbol{\mu}) + b C^*(\boldsymbol{\mu}) \right]. \quad (6)$$

**Sampling error** We finish this section with an analysis of the first component of error identified in Section 3: the sampling error. We begin by deriving a more explicit form of market-clearing prices:

**Theorem 4.3.** *Under the exponential trader model, the unique market-clearing equilibrium prices can be written as $\bar{\boldsymbol{\mu}} = \mathbb{E}_{\bar{\boldsymbol{\theta}}}[\phi(\omega)]$, where $\bar{\boldsymbol{\theta}} := \left(\sum_{i=1}^{N} \tilde{\boldsymbol{\theta}}_i/a_i\right)/\left(\sum_{i=1}^{N} 1/a_i\right)$ is the risk-aversion-weighted average belief and $\mathbb{E}_{\bar{\boldsymbol{\theta}}}$ is the expectation under $p(\cdot; \bar{\boldsymbol{\theta}})$.*

The sampling error arises because the beliefs $\tilde{\boldsymbol{\theta}}_i$ are only noisy signals of the ground truth. From Theorem 4.3 we see that this error may be compounded by the weighting according to risk aversions, which can skew the prices. To obtain a concrete bound on the error term $\|\boldsymbol{\mu}^{\text{true}} - \bar{\boldsymbol{\mu}}\|$, we need to make some assumptions about risk aversion coefficients, the true distribution of the outcome, and how this distribution is related to trader beliefs. For instance, suppose risk aversion coefficients are bounded both from below and above, the true outcome is drawn from an exponential-family distribution with natural parameter $\boldsymbol{\theta}^{\text{true}}$, and the beliefs $\tilde{\boldsymbol{\theta}}_i$ are independent samples with mean $\boldsymbol{\theta}^{\text{true}}$ and a bounded covariance matrix. Under these assumptions, one can show using standard concentration bounds that with high probability, $\|\boldsymbol{\mu}^{\text{true}} - \bar{\boldsymbol{\mu}}\| = O(\sqrt{1/N})$ as $N \to \infty$. In other words, market-clearing prices approach the ground truth as the number of traders increases. In Appendix B.4 we make the dependence on risk aversion and belief noise more explicit. The analysis of other information structures (e.g., biased or correlated beliefs) is beyond the scope of this paper; instead, we focus on the two error components that depend on the market design.

## 5 Market-maker Bias

We now analyze the market-maker bias—the difference between the marker-maker equilibrium prices $\boldsymbol{\mu}^{\star}$ and market-clearing prices $\bar{\boldsymbol{\mu}}$. We first state a global bound that depends on the liquidity $b$ and cost function $C$, but not on trader beliefs, and show that $\boldsymbol{\mu}^{\star} \to \bar{\boldsymbol{\mu}}$ with the rate $O(b)$ as $b \to 0$. The proof builds on Theorems 4.1 and 4.2 and uses the facts that $C^*$ is bounded on $\mathcal{M}$ (by our assumptions on $C$), and conjugates $F_i^*$ are strongly convex on $\mathcal{M}$ (from properties of the log partition function).

**Theorem 5.1** (Global Bias Bound). *Under the exponential trader model, for any $C$, there exists a constant $c$ such that $\|\boldsymbol{\mu}^{\star}(b;C) - \bar{\boldsymbol{\mu}}\| \leq cb$ for all $b \geq 0$.*

This result makes use of strong convexity constants that are valid over the entire set $\mathcal{M}$, which can be overly conservative when $\boldsymbol{\mu}^{\star}$ is close to $\bar{\boldsymbol{\mu}}$. Furthermore, it gives us only an upper bound, which cannot be used to compare different cost function families. In the rest of this section we pursue a tighter local analysis, based on the properties of $F_i^*$ and $C^*$ at $\bar{\boldsymbol{\mu}}$. Our local analysis requires assumptions that go beyond convexity and differentiability of the cost function. We call the class of functions that satisfy these assumptions *convex$^+$ functions*. (See Appendix A.3 for their complete treatment and a more general definition than provided here.) These functions are related to functions of *Legendre type* (see Sec. 26 of Rockafellar [15]). Informally, they are smooth functions that are strictly convex along directions in a certain space (the *gradient space*) and linear in orthogonal directions. For cost functions, strict convexity means that prices change in response to arbitrarily small trades, while the linear directions correspond to bundles with constant payoffs, whose prices are therefore fixed.

**Definition 5.2.** Let $f : \mathbb{R}^d \to \mathbb{R}$ be differentiable and convex. Its *gradient space* is the linear space parallel to the affine hull of its gradients, denoted as $\mathcal{G}(f) := \text{span}\{\nabla f(\boldsymbol{u}) - \nabla f(\boldsymbol{u}') : \boldsymbol{u}, \boldsymbol{u}' \in \mathbb{R}^d\}$.

**Definition 5.3.** We say that a convex function $f : \mathbb{R}^d \to \mathbb{R}$ is *convex$^+$* if it has continuous third derivatives and $\text{range}(\nabla^2 f(\boldsymbol{u})) = \mathcal{G}(f)$ for all $\boldsymbol{u} \in \mathbb{R}^d$.

It can be checked that if $P$ is a projection on $\mathcal{G}(f)$ then there exists some $\boldsymbol{a}$ such that $f(\boldsymbol{u}) = f(P\boldsymbol{u}) + \boldsymbol{a}^{\intercal}\boldsymbol{u}$, so $f$ is up to a linear term fully described by its values on $\mathcal{G}(f)$. The condition on the range of the Hessian ensures that $f$ is strictly convex over $\mathcal{G}(f)$, so its gradient map is invertible over $\mathcal{G}(f)$. This means that the Hessian can be expressed as a function of the gradient, i.e., there exists a matrix-valued function $H_f$ such that $\nabla^2 f(\boldsymbol{u}) = H_f(\nabla f(u))$ (see Proposition A.8). The cost functions $C$ for both the LMSR and the sum of independent LMSRs (IND) are convex$^+$.

**Example 5.4** (LMSR as a convex$^+$ function). For LMSR, the gradient space of $C$ is parallel to the simplex: $\mathcal{G}(C) = \{\boldsymbol{u} : \mathbf{1}^{\intercal}\boldsymbol{u} = 0\}$. The gradients of $C$ are points in the relative interior of the simplex. Given such a point $\boldsymbol{\mu} = \nabla C(\boldsymbol{s})$, the corresponding Hessian is $\nabla^2 C(\boldsymbol{s}) = H_C(\boldsymbol{\mu}) = (\text{diag}_{k \in [K]} \mu_k) - \boldsymbol{\mu}\boldsymbol{\mu}^{\intercal}$, where $\text{diag}_{k \in [K]} \mu_k$ denotes the diagonal matrix with values $\mu_k$ on the diagonal. The null space of $H_C(\boldsymbol{\mu})$ is $\{c\mathbf{1} : c \in \mathbb{R}\}$, so $C$ is linear in the all-ones direction (buying one share of each security always has cost one), but strictly convex in directions from $\mathcal{G}(C)$.

**Example 5.5** (IND as a convex$^+$ function). For IND, the gradient space is $\mathbb{R}^K$ and the gradients are the points in $(0,1)^K$. In this case, $H_C(\boldsymbol{\mu}) = \text{diag}_k[\mu_k(1 - \mu_k)]$. This matrix has full rank.

Our next theorem shows that for an appropriate vector $\boldsymbol{u}$, which depends on $\bar{\boldsymbol{\mu}}$ and $C$, we have $\boldsymbol{\mu}^{\star}(b;C) = \bar{\boldsymbol{\mu}} + b\boldsymbol{u} + \boldsymbol{\varepsilon}_b$, where $\|\boldsymbol{\varepsilon}_b\| = O(b^2)$. Here, the $O(\cdot)$ is taken as $b \to 0$, so the error term

$\varepsilon_b$ goes to zero faster than the term $b\boldsymbol{u}$, which we call the *asymptotic bias*. Our analysis is *local* in the sense that the constants hiding within $O(\cdot)$ may depend on $\bar{\boldsymbol{\mu}}$. This analysis fully uncovers the main asymptotic term and therefore allows comparison of cost families. In our experiments, we show that the asymptotic bias is an accurate estimate of the bias even for moderately large values of $b$.

**Theorem 5.6** (Local Bias Bound). *Assume that the cost function $C$ is convex$^+$. Then*

$$\boldsymbol{\mu}^\star(b; C) = \bar{\boldsymbol{\mu}} - b(\bar{a}/N)H_T(\bar{\boldsymbol{\mu}})\partial C^*(\bar{\boldsymbol{\mu}}) + \varepsilon_b \ , \quad where \ \|\varepsilon_b\| = O(b^2).$$

*In the expression above, $\bar{a} = N/(\sum_{i=1}^{N} 1/a_i)$ is the harmonic mean of risk-aversion coefficients and $H_T(\bar{\boldsymbol{\mu}})\partial C^*(\bar{\boldsymbol{\mu}})$ is guaranteed to consist of a single point even when $\partial C^*(\bar{\boldsymbol{\mu}})$ is a set.*

The theorem is proved by a careful application of Taylor's Theorem and crucially uses properties of conjugates of convex$^+$ functions, which we derive in Appendix A.3. It gives us a formula to calculate the asymptotic bias for any cost function for a particular value of $\bar{\boldsymbol{\mu}}$, or evaluate the worst-case bias against some set of possible market-clearing prices. It also constitutes an important step in comparing cost function families. To compare the convergence error of two costs $C$ and $C'$ in the next section, we require that their liquidities $b$ and $b'$ be set so that they have (approximately) the same bias, i.e., $\|\boldsymbol{\mu}^\star(b'; C') - \bar{\boldsymbol{\mu}}\| \approx \|\boldsymbol{\mu}^\star(b; C) - \bar{\boldsymbol{\mu}}\|$. Theorem 5.6 tells us that this can be achieved by the linear rule $b' = b/\eta$ where $\eta = \|H_T(\bar{\boldsymbol{\mu}})\partial C'^*(\bar{\boldsymbol{\mu}})\| / \|H_T(\bar{\boldsymbol{\mu}})\partial C^*(\bar{\boldsymbol{\mu}})\|$. For $C = \mathtt{LMSR}$ and $C' = \mathtt{IND}$, we prove that the corresponding $\eta \in [1, 2]$. Equivalently, this means that for the same value of $b$ the asymptotic bias of $\mathtt{IND}$ is at least as large as that of $\mathtt{LMSR}$, but no more than twice as large:

**Theorem 5.7.** *For any $\bar{\boldsymbol{\mu}}$ there exists $\eta \in [1, 2]$ such that for all $b$, $\|\boldsymbol{\mu}^\star(b/\eta; \mathtt{IND}) - \bar{\boldsymbol{\mu}}\| = \|\boldsymbol{\mu}^\star(b; \mathtt{LMSR}) - \bar{\boldsymbol{\mu}}\| \pm O(b^2)$. For this same $\eta$, also $\|\boldsymbol{\mu}^\star(b; \mathtt{IND}) - \bar{\boldsymbol{\mu}}\| = \eta\|\boldsymbol{\mu}^\star(b; \mathtt{LMSR}) - \bar{\boldsymbol{\mu}}\| \pm O(b^2)$.*

Theorem 5.6 also captures an intuitive relationship which can guide the market maker in adjusting the market liquidity $b$ as the number of traders $N$ and their risk aversion coefficients $a_i$ vary. In particular, holding $\bar{\boldsymbol{\mu}}$ and the cost function fixed, we can maintain the same amount of bias by setting $b \propto N/\bar{a}$. Note that $1/a_i$ plays the role of the budget of trader $i$ in the sense that at fixed prices, the trader will spend an amount of cash proportional to $1/a_i$. Thus $N/\bar{a} = \sum_i (1/a_i)$ corresponds to the total amount of available cash among the traders in the market. Similarly, the market maker's worst-case loss, amounting to the market maker's cash, is proportional to $b$, so setting $b \propto \sum_i (1/a_i)$ is natural.

# 6   Convergence Error

We now study the convergence error, namely the difference between the prices $\boldsymbol{\mu}^t$ at round $t$ and the market-maker equilibrium prices $\boldsymbol{\mu}^\star$. To do so, we must posit a model of how the traders interact with the market. Following Frongillo and Reid [9], we assume that in each round, a trader $i \in [N]$, chosen uniformly at random, buys a bundle $\boldsymbol{\delta} \in \mathbb{R}^K$ that optimizes her utility given the current market state $\boldsymbol{s}$ and her existing security and cash allocations, $\boldsymbol{r}_i$ and $c_i$. The resulting updates of the allocation vector $\boldsymbol{r} = (\boldsymbol{r}_i)_{i=1}^N$ correspond to randomized block-coordinate descent on the potential function $F(\boldsymbol{r})$ with blocks $\boldsymbol{r}_i$ (see Appendix D.1 and Frongillo and Reid [9]). We refer to this model as the *all-security (trader) dynamics* ($\mathtt{ASD}$).[2] We apply and extend the analysis of block-coordinate descent to this setting. We focus on convex$^+$ functions and conduct local convergence analysis around the minimizer of $F$. Our experiments demonstrate that the local analysis accurately estimates the convergence rate.

Let $\boldsymbol{r}^\star$ denote an arbitrary minimizer of $F$ and let $F^\star$ be the minimum value of $F$. Also, let $\boldsymbol{r}^t$ denote the allocation vector and $\boldsymbol{\mu}^t$ the market price vector after the $t$th trade. Instead of directly analyzing the convergence error $\|\boldsymbol{\mu}^t - \boldsymbol{\mu}^\star\|$, we bound the suboptimality $F(\boldsymbol{r}^t) - F^\star$ since $\|\boldsymbol{\mu}^t - \boldsymbol{\mu}^\star\|^2 = \Theta(F(\boldsymbol{r}^t) - F^\star)$ for convex$^+$ costs $C$ under $\mathtt{ASD}$ (see Appendix D.7.1).

Convex$^+$ functions are locally strongly convex and have a Lipschitz-continuous gradient, so the standard analysis of block-coordinate descent [9, 11] implies linear convergence, i.e., $\mathbb{E}[F(\boldsymbol{r}^t)] - F^\star \leq O(\gamma^t)$ for some $\gamma < 1$, where the expectation is under the randomness of the algorithm. We refine the standard analysis by (1) proving not only upper, but also lower bounds on the convergence rate, and (2) proving an explicit dependence of $\gamma$ on the cost function $C$ and the liquidity $b$. These two refinements are crucial for comparison of cost families, as we demonstrate with the comparison of $\mathtt{LMSR}$ and $\mathtt{IND}$. We begin by formally defining bounds on local convergence of any randomized iterative algorithm that minimizes a function $F(\boldsymbol{r})$ via a sequence of iterates $\boldsymbol{r}^t$.

**Definition 6.1.** We say that $\gamma_{\text{high}}$ is an *upper bound on the local convergence rate* of an algorithm if, with probability 1 under the randomness of the algorithm, the algorithm reaches an iteration $t_0$ such that for some $c > 0$ and all $t \geq t_0$, $\mathbb{E}\left[F(\mathbf{r}^t) \mid \mathbf{r}^{t_0}\right] - F^\star \leq c\gamma_{\text{high}}^{t-t_0}$. We say that $\gamma_{\text{low}}$ is a *lower bound on the local convergence rate* if $\gamma_{\text{high}} \geq \gamma_{\text{low}}$ holds for all upper bounds $\gamma_{\text{high}}$.

To state explicit bounds, we use the notation $D := \text{diag}_{i \in [N]} a_i$ and $P := I_N - \mathbf{1}\mathbf{1}^\intercal/N$, where $I_N$ is the $N \times N$ identity matrix and $\mathbf{1}$ is the all-ones vector. We write $M^+$ for the pseudoinverse of a matrix $M$ and $\lambda_{\min}(M)$ and $\lambda_{\max}(M)$ for its smallest and largest positive eigenvalues.

**Theorem 6.2** (Local Convergence Bound). *Assume that $C$ is convex$^+$. Let $H_T := H_T(\bar{\boldsymbol{\mu}})$ and $H_C := H_C(\bar{\boldsymbol{\mu}})$. For the all-securities dynamics, the local convergence rate is bounded between*

$$\gamma_{\text{high}}^{\text{ASD}} = 1 - \tfrac{2b}{N} \cdot \lambda_{\min}(PDP) \cdot \lambda_{\min}\big( H_T^{1/2} H_C^+ H_T^{1/2} \big) + O(b^2) \ ,$$

$$\gamma_{\text{low}}^{\text{ASD}} = 1 - \tfrac{2b}{N} \cdot \lambda_{\max}(PDP) \cdot \lambda_{\max}\big( H_T^{1/2} H_C^+ H_T^{1/2} \big) - O(b^2) \ .$$

In our proof, we first establish both lower and upper bounds on convergence of a generic block-coordinate descent that extend the results of Nesterov [11]. We then analyze the behavior of the algorithm for the specific structure of our objective to obtain explicit lower and upper bounds. Our bounds prove linear convergence with the rate $\gamma = 1 - \Theta(b)$. Since the convergence gets worse as $b \to 0$, there is a trade-off with the bias, which decreases as $b \to 0$.

Theorems 5.6 and 6.2 enable systematic quantitative comparisons of cost families. For simplicity, assume that $N \geq 2$ and all risk aversions are $a$, so $\lambda_{\min}(PDP) = \lambda_{\max}(PDP) = a$. To compare convergence rates of two costs $C$ and $C'$, we need to control for bias. As discussed after Theorem 5.6, their biases are (asymptotically) equal if their liquidities are linearly related as $b' = b/\eta$ for a suitable $\eta$. Theorem 6.2 then states that $C_{b'}'$ requires (asymptotically) at most a factor of $\rho$ as many trades as $C_b$ to achieve the same convergence error, where $\rho := \eta \cdot \lambda_{\max}(H_T^{1/2} H_C^+ H_T^{1/2})/\lambda_{\min}(H_T^{1/2} H_{C'}^+ H_T^{1/2})$. Similarly, $C_b$ requires at most a factor of $\rho'$ as many trades as $C_{b'}'$, with $\rho'$ defined symmetrically to $\rho$. For $C = \texttt{LMSR}$ and $C' = \texttt{IND}$, we can show that $\rho \leq 2$ and $\rho' \leq 2$, yielding the following result:

**Theorem 6.3.** *Assume that $N \geq 2$ and all risk aversions are equal to $a$. Consider running* $\texttt{LMSR}$ *with liquidity $b$ and* $\texttt{IND}$ *with liquidity $b' = b/\eta$ such that their asymptotic biases are equal. Denote the iterates of the two runs of the market as $\boldsymbol{\mu}_{\texttt{LMSR}}^t$ and $\boldsymbol{\mu}_{\texttt{IND}}^t$ and the respective market-maker equilibria as $\boldsymbol{\mu}_{\texttt{LMSR}}^\star$ and $\boldsymbol{\mu}_{\texttt{IND}}^\star$. Then, with probability 1, there exist $t_0$ and $t_1 \geq t_0$ such that for all $t \geq t_1$ and sufficiently small $b$*

$$\mathbb{E}_{t_0}\left[\left\|\boldsymbol{\mu}_{\texttt{LMSR}}^{2t(1+\varepsilon)} - \boldsymbol{\mu}_{\texttt{LMSR}}^\star\right\|^2\right] \leq \mathbb{E}_{t_0}\left[\left\|\boldsymbol{\mu}_{\texttt{IND}}^t - \boldsymbol{\mu}_{\texttt{IND}}^\star\right\|^2\right] \leq \mathbb{E}_{t_0}\left[\left\|\boldsymbol{\mu}_{\texttt{LMSR}}^{(t/2)(1-\varepsilon)} - \boldsymbol{\mu}_{\texttt{LMSR}}^\star\right\|^2\right] \ ,$$

*where $\varepsilon = O(b)$ and $\mathbb{E}_{t_0}[\cdot] = \mathbb{E}[\cdot \mid \mathbf{r}^{t_0}]$ conditions on the $t_0$th iterate of a given run.*

This result means that $\texttt{LMSR}$ and $\texttt{IND}$ are roughly equivalent (up to a factor of two) in terms of the number of trades required to achieve a given accuracy. This is somewhat surprising as this implies that maintaining price coherence does not offer strong informational advantages (at least when traders are individually coherent, as assumed here). However, while there is little difference between the two costs in terms of accuracy, there is a difference in terms of the worst-case loss. For $K$ securities, the worst-case loss of $\texttt{LMSR}$ with the liquidity $b$ is $b \log K$, and the worst-case loss of $\texttt{IND}$ with the liquidity $b'$ is $b' K \log 2$. If liquidities are chosen as in Theorem 6.3, so that $b'$ is up to a factor-of-two smaller than $b$, then the worst-case loss of $\texttt{IND}$ is at least $(bK/2) \log 2$, which is always worse than the $\texttt{LMSR}$'s loss of $b \log K$, and the ratio of the two losses increases as $K$ grows.

When all risk aversion coefficients are equal to some constant $a$, then the dependence of Theorem 6.2 on the number of traders $N$ and their risk aversion is similar to the dependence in Theorem 5.6. For instance, to guarantee that $\gamma$ stays below a certain level for varying $N$ and $a$ requires $b = \Omega(N/a)$.

## 7 Numerical Experiments

We evaluate the tightness of our theoretical bounds via numerical simulation. We consider a complete market over $K = 5$ securities and simulate $N = 10$ traders with risk aversion coefficients equal to 1. These values of $N$ and $K$ are large enough to demonstrate the tightness of our results, but small enough that simulations are tractable. While our theory comprehensively covers heterogeneous

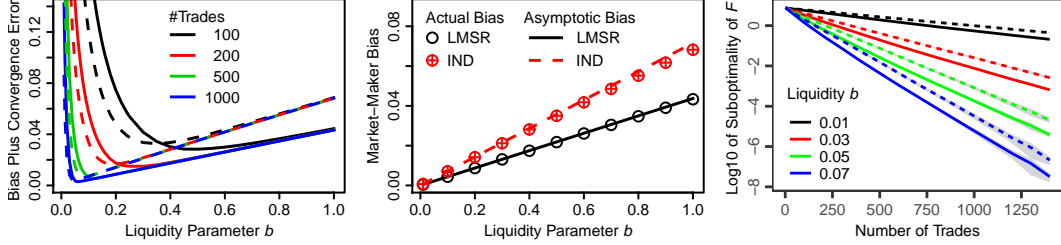

Figure 1: (Left) The tradeoff between market-maker bias and convergence. Solid lines are for LMSR, dashed for IND, the color indicates the number of trades. (Center) Market-maker bias as a function of $b$. (Right) Convergence in the objective. Shading indicates 95% confidence based on 20 trading sequences.

risk aversions and the dependence on the number of traders and securities, we have chosen to keep these values fixed, so that we can more cleanly explore the impact of liquidity and number of trades. We consider the two most commonly studied cost functions: LMSR and IND. We fix the ground-truth natural parameter $\boldsymbol{\theta}^{\text{true}}$ and independently sample the belief $\tilde{\boldsymbol{\theta}}_i$ of each trader from Normal$(\boldsymbol{\theta}^{\text{true}}, \sigma^2 I_K)$, with $\sigma = 5$. We consider a *single-peaked* ground truth distribution with $\theta_1^{\text{true}} = \log(1 - \nu(K - 1))$ and $\theta_k^{\text{true}} = \log \nu$ for $k \neq 1$, with $\nu = 0.02$. Trading is simulated according to the all-security dynamics (ASD) as described at the start of Section 6. In Appendix E, we show qualitatively similar results using a uniform ground truth distribution and single-security dynamics (SSD).

We first examine the tradeoff that arises between market-maker bias and convergence error as the liquidity parameter is adjusted. Fig. 1 (left) shows the combined bias and convergence error, $\|\boldsymbol{\mu}^t - \bar{\boldsymbol{\mu}}\|$, as a function of liquidity and the number of trades $t$ (indicated by the color of the line) for the two cost functions, averaged over twenty random trading sequences. The minimum point on each curve tells us the optimal value of the liquidity parameter $b$ for the particular cost function and particular number of trades. When the market is run for a short time, larger values of $b$ lead to lower error. On the other hand, smaller values of $b$ are preferable as the number of trades grows, with the combined error approaching 0 for small $b$.

In Fig. 1 (center) we plot the bias $\|\boldsymbol{\mu}^\star(b; C) - \bar{\boldsymbol{\mu}}\|$ as a function of $b$ for both LMSR and IND. We compare this with the theoretical approximation $\|\boldsymbol{\mu}^\star(b; C) - \bar{\boldsymbol{\mu}}\| \approx b(\bar{a}/N)\|H_T(\bar{\boldsymbol{\mu}})\partial C^\star(\bar{\boldsymbol{\mu}})\|$ from Theorem 5.6. Although Theorem 5.6 only gives an asymptotic guarantee as $b \to 0$, the approximation is fairly accurate even for moderate values of $b$. In agreement with Theorem 5.7, the bias of IND is higher than that of LMSR at any fixed value of $b$, but by no more than a factor of two.

In Fig. 1 (right) we plot the log of $\hat{\mathbb{E}}[F(\boldsymbol{r}^t)] - F^\star$ as a function of the number of trades $t$ for our two cost functions and several liquidity levels. Even for small $t$ the curves are close to linear, showing that the local linear convergence rate kicks in essentially from the start of trade in our simulations. In other words, there exist some $\hat{c}$ and $\hat{\gamma}$ such that, empirically, we have $\hat{\mathbb{E}}[F(\boldsymbol{r}^t)] - F^\star \approx \hat{c}\hat{\gamma}^t$, or equivalently, $\log(\hat{\mathbb{E}}[F(\boldsymbol{r}^t)] - F^\star) \approx \log \hat{c} + t \log \hat{\gamma}$. Plugging the belief values into Theorem 6.2, the slope of the curve for LMSR should be $\log_{10} \hat{\gamma} \approx -0.087b$ for sufficiently small $b$, and the slope for IND should be between $-0.088b$ and $-0.164b$. In Appendix E, we verify that this is the case.

## 8  Conclusion

Our theoretical framework provides a meaningful way to quantitatively evaluate the error tradeoffs inherent in different choices of cost functions and liquidity levels. We find, for example, that to maintain a fixed amount of bias, one should set the liquidity parameter $b$ proportional to a measure of the amount of cash that traders are willing to spend. We also find that, although the LMSR maintains coherent prices while IND does not, the two are equivalent up to a factor of two in terms of the number of trades required to reach any fixed accuracy, though LMSR has lower worst-case loss.

We have assumed that traders' beliefs are individually coherent. Experimental evidence suggests that LMSR might have additional informational advantages over IND when traders' beliefs are incoherent or each trader is informed about only a subset of events [12]. We touch on this in Appendix C.2, but leave a full exploration of the impact of different assumptions on trader beliefs to future work.

## Footnotes

[1] A longer version of this paper containing the appendix is available on arXiv and the authors' websites.

[2]In Appendix D, we also analyze the *single-security (trader) dynamics* ($\mathtt{SSD}$), in which a randomly chosen trader randomly picks a single security to trade, corresponding to randomized coordinate descent on $F$.

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
