[Supplementary Material · main-final2-appendix.pdf]

# A Mathematical Background

## A.1 Vectors, Matrices, Intervals

Vectors are denoted by small boldface italics $\boldsymbol{u}, \boldsymbol{\mu}, \ldots$, matrices by large italics $A, B, P, \ldots$. The norm $\|\cdot\|$ denotes the standard Euclidean norm for vectors, and the operator norm for matrices, i.e., $\|A\| = \sup_{\boldsymbol{u}: \|\boldsymbol{u}\|=1} \|A\boldsymbol{u}\|$. The *range of a matrix*, denoted $\mathrm{range}(\cdot)$, is the span of its columns. For a symmetric matrix $A$, its pseudoinverse, denoted $A^+$, is the unique symmetric matrix with $\mathrm{range}(A^+) = \mathrm{range}(A)$ such that $A^+A = AA^+ = P$ where $P$ is the projection on $\mathrm{range}(A)$.

For symmetric matrices $A$ and $B$, we write $A \preceq B$ to denote that $B - A$ is positive-semidefinite. We use the notation $A \simeq B \pm C$ to denote $B - C \preceq A \preceq B + C$. For scalars, we similarly write $a \simeq b \pm c$ to mean $a \in [b - c, b + c]$.

We use $\lambda_{\min}(A)$ and $\lambda_{\max}(A)$ to denote the smallest and the largest positive eigenvalue of a symmetric positive-semidefinite matrix. We also write $\lambda_{\min}(A, B)$ and $\lambda_{\max}(A, B)$ for the generalized minimum and maximum eigenvalues defined as follows whenever $B \neq \mathbf{0}$:

$$\lambda_{\min}(A, B) := \min_{\boldsymbol{u} \in \mathrm{range}(B) \setminus \{\mathbf{0}\}} \frac{\boldsymbol{u}^\mathsf{T} A \boldsymbol{u}}{\boldsymbol{u}^\mathsf{T} B \boldsymbol{u}} \ , \qquad \lambda_{\max}(A, B) := \max_{\boldsymbol{u} \in \mathrm{range}(B) \setminus \{\mathbf{0}\}} \frac{\boldsymbol{u}^\mathsf{T} A \boldsymbol{u}}{\boldsymbol{u}^\mathsf{T} B \boldsymbol{u}} \ .$$

The following formulas follow by substituting $\boldsymbol{v} = B^{1/2} \boldsymbol{u}$:

$$\lambda_{\min}(A, B) = \lambda_{\min}\left((B^{1/2})^+ A (B^{1/2})^+\right) \quad \text{if } \mathrm{range}(A) \subseteq \mathrm{range}(B),$$

$$\lambda_{\max}(A, B) = \lambda_{\max}\left((B^{1/2})^+ A (B^{1/2})^+\right) \quad \text{if } \mathrm{range}(A) \supseteq \mathrm{range}(B).$$

By Eq. (180) of Petersen and Pedersen [14], we also have for any matrix $A$ (not necessarily square):

$$\lambda_{\min}(AA^\mathsf{T}) = \lambda_{\min}(A^\mathsf{T}A) \ , \quad \lambda_{\max}(AA^\mathsf{T}) = \lambda_{\max}(A^\mathsf{T}A) \ . \tag{7}$$

The following two results relate $A$ and its pseudoinverse $A^+$:

**Proposition A.1.** *Let $A$ and $B$ be symmetric matrices such that $\mathrm{range}(A) = \mathrm{range}(B)$. Then $A \preceq B$ if and only if $B^+ \preceq A^+$.*

*Proof.* Assume that $\mathrm{range}(A) = \mathrm{range}(B) = \mathcal{L}$. Then we have the following equivalences

$$A \preceq B \ \text{ iff } \ \boldsymbol{u}^\mathsf{T} A \boldsymbol{u} \leq \boldsymbol{u}^\mathsf{T} B \boldsymbol{u} \text{ for all } \boldsymbol{u} \in \mathcal{L} \setminus \{\mathbf{0}\}$$

$$\text{iff } \ 1 \leq \min_{\boldsymbol{u} \in \mathcal{L} \setminus \{\mathbf{0}\}} \frac{\boldsymbol{u}^\mathsf{T} B \boldsymbol{u}}{\boldsymbol{u}^\mathsf{T} A \boldsymbol{u}}$$

$$\text{iff } \ 1 \leq \lambda_{\min}(B, A) = \lambda_{\min}\left((A^{1/2})^+ B (A^{1/2})^+\right)$$

$$= \lambda_{\min}\left(B^{1/2} A^+ B^{1/2}\right) = \lambda_{\min}(A^+, B^+) \qquad \text{(by Eq. 7)}$$

$$\text{iff } \ 1 \leq \min_{\boldsymbol{u} \in \mathcal{L} \setminus \{\mathbf{0}\}} \frac{\boldsymbol{u}^\mathsf{T} A^+ \boldsymbol{u}}{\boldsymbol{u}^\mathsf{T} B^+ \boldsymbol{u}}$$

$$\text{iff } \ B^+ \preceq A^+ \qquad\qquad\qquad\qquad\qquad\qquad \square$$

**Proposition A.2** (Blockwise Inversion). *Let $A, B, C \in \mathbb{R}^{d \times d}$, where $A$ and $C$ are symmetric, and let $S = C - B^\mathsf{T} A^+ B$. Assume the following conditions hold:*

- $\mathrm{range}(A) \cap \mathrm{range}(C) = \{\mathbf{0}\}$.
- $\mathrm{range}(B) \subseteq \mathrm{range}(A)$.
- $\mathrm{range}(B^\mathsf{T}) \subseteq \mathrm{range}(C)$.
- $\mathrm{range}(S) = \mathrm{range}(C)$.

*Then*

$$(A + B + B^\mathsf{T} + C)^+ = A^+ + (I - A^+ B) S^+ (I - A^+ B)^\mathsf{T} \ .$$

*Proof.* This follows by the formula in Section 9.1.3 of Petersen and Pedersen [14] for the block matrix

$$\begin{pmatrix} U^\mathsf{T} A U & U^\mathsf{T} B V \\ V^\mathsf{T} B U & V^\mathsf{T} C V \end{pmatrix}$$

where $U$ and $V$ are matrices with an orthonormal basis of $\mathrm{range}(A)$ and $\mathrm{range}(C)$, respectively. $\square$

The final result of this section relates the optimization of a quadratic form with a matrix $A$ to the quadratic form with the matrix $A^+$:

**Proposition A.3.** *Let $A \in \mathbb{R}^{d \times d}$ be a symmetric positive-semidefinite matrix and let $\boldsymbol{u} \in \text{range}(A)$. Then*

$$\min_{\boldsymbol{\delta} \in \mathbb{R}^d} \left( \boldsymbol{\delta}^\mathsf{T} \boldsymbol{u} + \frac{1}{2} \boldsymbol{\delta}^\mathsf{T} A \boldsymbol{\delta} \right) = -\frac{1}{2} \boldsymbol{u}^\mathsf{T} A^+ \boldsymbol{u} \ .$$

*Proof.* The result follows from convex conjugacy of quadratic convex functions, see page 108 of Rockafellar [15]. $\square$

## A.2 Convex Analysis

If $f : \mathbb{R}^d \to \mathbb{R} \cup \{\infty\}$ then the *epigraph* of $f$ is the set of points in $\mathbb{R}^d \times \mathbb{R}$ that lie on or above the graph of $f$. The function $f$ is called *convex* if its epigraph is convex. It is called *closed* if its epigraph is closed, and *proper* if it is not identically equal to $\infty$. The *effective domain* of $f$ is the set of points where it is finite, denoted $\text{dom } f$. The convex hull of a set $S$, written $\text{conv } S$, is the smallest convex set containing $S$. For instance, the simplex in $\mathbb{R}^d$ is the convex hull of the vectors of standard basis.

An *affine subspace* of $\mathbb{R}^d$ is any set that can be written as $\mathcal{A} = \{\boldsymbol{a} + \boldsymbol{u} : \boldsymbol{u} \in \mathcal{L}\}$ for some fixed vector $\boldsymbol{a}$ where $\mathcal{L}$ is a linear subspace of $\mathbb{R}^d$. We refer to $\mathcal{L}$ as the linear space *parallel to* $\mathcal{A}$. The *affine hull* of a non-empty set $S$, denoted $\text{aff } S$, is the smallest affine set that contains $S$. The *relative interior* of a set $S$, denoted $\text{ri } S$, is the interior of $S$ under the topology of its affine hull. The *relative boundary* of a set $S$ consists of points in the closure of $S$ that are not in its relative interior, $(\text{cl } S) \backslash (\text{ri } S)$. For instance, the affine hull of the simplex consists of vectors $\boldsymbol{u}$ such that $\boldsymbol{u}^\mathsf{T} \mathbf{1} = 1$, where $\mathbf{1}$ is the all-ones vector. The parallel linear space is $\{\boldsymbol{u} : \boldsymbol{u}^\mathsf{T} \mathbf{1} = 0\}$. The simplex has an empty interior, but its relative interior consists of points $\{\boldsymbol{u} \in (0,1)^d : \boldsymbol{u}^\mathsf{T} \mathbf{1} = 1\}$. The relative boundary of the simplex consists of those points in the simplex which have at least one coordinate equal to zero.

For a convex $f : \mathbb{R}^d \to \mathbb{R} \cup \{\infty\}$, the *subdifferential* of $f$ at $\boldsymbol{u}$ is defined as $\partial f(\boldsymbol{u}) := \{\boldsymbol{v} \in \mathbb{R}^d : f(\boldsymbol{u}') \geq f(\boldsymbol{u}) + \boldsymbol{v}^\mathsf{T}(\boldsymbol{u}' - \boldsymbol{u}), \forall \boldsymbol{u}' \in \mathbb{R}^d\}$. Any convex $f : \mathbb{R}^d \to \mathbb{R} \cup \{\infty\}$ is *subdifferentiable*, i.e., its subdifferential is non-empty, at all points in $\text{ri } \text{dom } f$. If the subdifferential is a singleton, it coincides with the gradient. Given a proper convex function $f : \mathbb{R}^d \to \mathbb{R} \cup \{\infty\}$, we define its *convex conjugate* $f^* : \mathbb{R}^d \to \mathbb{R} \cup \{\infty\}$ by $f^*(\boldsymbol{\mu}) := \sup_{\boldsymbol{u} \in \mathbb{R}^d} [\boldsymbol{u}^\mathsf{T} \boldsymbol{\mu} - f(\boldsymbol{u})]$. For a closed proper convex function $f$, its conjugate is also closed, proper and convex, and the following statements are equivalent:

$$\boldsymbol{\mu} \in \partial f(\boldsymbol{u}) \quad \text{iff} \quad \boldsymbol{u} \in \partial f^*(\boldsymbol{\mu}) \quad \text{iff} \quad f(\boldsymbol{u}) + f^*(\boldsymbol{\mu}) = \boldsymbol{u}^\mathsf{T} \boldsymbol{\mu} \ .$$

We will use the following variant of a duality result known as Fenchel's duality:

**Theorem A.4** (Fenchel's duality). *Let $f : \mathbb{R}^d \to \mathbb{R}$ and $g : \mathbb{R}^K \to \mathbb{R} \cup \{\infty\}$ be closed proper convex functions and $A \in \mathbb{R}^{K \times d}$. Assume that there exists some $\boldsymbol{u} \in \mathbb{R}^d$ such that $A\boldsymbol{u} \in \text{ri}(\text{dom } g)$ and some $\boldsymbol{\mu} \in \text{ri}(\text{dom } g^*)$ such that $A^\mathsf{T} \boldsymbol{\mu} \in \text{ri}(\text{dom } f^*)$. Then*

$$\inf_{\boldsymbol{u} \in \mathbb{R}^d} [f(-\boldsymbol{u}) + g(A\boldsymbol{u})] = \sup_{\boldsymbol{\mu} \in \mathbb{R}^K} [-f^*(A^\mathsf{T} \boldsymbol{\mu}) - g^*(\boldsymbol{\mu})]$$

*and both the supremum and the infimum are attained. Vectors $\hat{\boldsymbol{u}}$ and $\hat{\boldsymbol{\mu}}$ are their respective solutions if and only if $A^\mathsf{T} \hat{\boldsymbol{\mu}} \in \partial f(-\hat{\boldsymbol{u}})$ and $\hat{\boldsymbol{\mu}} \in \partial g(A\hat{\boldsymbol{u}})$.*

*Proof.* The results follows from Corollary 31.2.1 and Theorem 31.3 of Rockafellar [15]. $\square$

## A.3 Convex$^+$ Functions

Throughout the paper we work with functions that satisfy additional assumptions beyond convexity. We refer to them as *convex$^+$ functions*. They have a close relationship to functions of *Legendre type* (see Sec. 26 of Rockafellar [15]). Before we define convex$^+$ functions, we introduce the *gradient space*, which plays a role in their structure.

**Definition A.5.** Let $f : \mathbb{R}^d \to \mathbb{R} \cup \{\infty\}$ be differentiable on the interior of its domain $D := \text{int } \text{dom } f$. Its *gradient space*, denoted $\mathcal{G}(f)$, is the linear space parallel to the affine hull of the set of its gradients,

$$\mathcal{G}(f) := \text{span}\{\nabla f(\boldsymbol{u}) - \nabla f(\boldsymbol{u}') : \boldsymbol{u}, \boldsymbol{u}' \in D\} \ .$$

**Definition A.6.** Let $f : \mathbb{R}^d \to \mathbb{R} \cup \{\infty\}$ and $D := \operatorname{int} \operatorname{dom} f$. We say that $f$ is *convex$^+$* if it satisfies the following conditions:

1. $f$ is closed and convex.
2. $D$ is non-empty.
3. $f$ has continuous third derivatives on $D$.
4. $\operatorname{range}(\nabla^2 f(\boldsymbol{u})) = \mathcal{G}(f)$ for all $\boldsymbol{u} \in D$.
5. $\lim_{t\to\infty} \|\nabla f(\boldsymbol{u}_t)\| = \infty$ whenever $\boldsymbol{u}_1, \boldsymbol{u}_2, \dots$ is a sequence in $D$ converging to a boundary point of $D$.

**Proposition A.7.** *Let $f : \mathbb{R}^d \to \mathbb{R} \cup \{\infty\}$ be convex$^+$, $D := \operatorname{int} \operatorname{dom} f$. Let $P$ be the projection on $\mathcal{G}(f)$ and $\boldsymbol{a}$ the unique point in $\mathcal{A} \cap \mathcal{G}(f)^\perp$, where $\mathcal{A}$ is the affine hull of the set of gradients of $f$. Then the following statements hold:*

1. $f(\boldsymbol{u}) = f(P\boldsymbol{u}) + \boldsymbol{a}^\mathsf{T}\boldsymbol{u}$.
2. $\nabla f(\boldsymbol{u}) = \nabla f(P\boldsymbol{u})$.
3. $\nabla^2 f(\boldsymbol{u}) = \nabla^2 f(P\boldsymbol{u})$.
4. $f$ *is strictly convex on* $D \cap \mathcal{G}(f)$.
5. $\nabla f$ *is one-to-one on* $D \cap \mathcal{G}(f)$.

*Proof.* We prove the first statement by the Mean Value Theorem. First, since $f$ is differentiable on an open convex $D$, it must be actually continuously differentiable on $D$ (by Corollary 25.5.1 of Rockafellar [15]), so the Mean Value Theorem can be applied on $D$. Let $\boldsymbol{u} \in D$ and $\boldsymbol{v} \in \mathcal{G}(f)^\perp$. Then for any $\boldsymbol{u}' = \boldsymbol{u} + t\boldsymbol{v} \in D$, we have, for some $\bar{\boldsymbol{u}}$ on the line segment connecting $\boldsymbol{u}$ and $\boldsymbol{u}'$,

$$f(\boldsymbol{u}') = f(\boldsymbol{u}) + [\nabla f(\bar{\boldsymbol{u}})]^\mathsf{T} t\boldsymbol{v} = f(\boldsymbol{u}) + t\boldsymbol{a}^\mathsf{T}\boldsymbol{v} \tag{8}$$

where the second equality follows because $\boldsymbol{v} \perp \mathcal{G}(f)$. We argue that the entire line $\{\boldsymbol{u} + t\boldsymbol{v} : t \in \mathbb{R}\}$ must be contained in $D$. For contradiction, assume it intersects the boundary of $D$ at $\boldsymbol{u}^\star = \boldsymbol{u} + t^\star \boldsymbol{v}$, and say $t^\star > 0$. Consider an increasing sequence $0 = t_1, t_2, \dots$ converging to $t^\star$. Eq. (8) holds for $\boldsymbol{u}'$ replaced by $\boldsymbol{u}_i = \boldsymbol{u} + t_i\boldsymbol{v}$ as well as points $\tilde{\boldsymbol{u}}_i = \tilde{\boldsymbol{u}} + t_i\boldsymbol{v}$, where $\tilde{\boldsymbol{u}}$ is in a small enough neighborhood of $\boldsymbol{u}$ along directions in $\mathcal{G}(f)$. This means that $\nabla f(\boldsymbol{u}_i) = P\nabla f(\boldsymbol{u}) + \boldsymbol{a} = \nabla f(\boldsymbol{u})$. However, this is not possible for convex$^+$ functions, because the norms of their gradients go to $\infty$ towards the boundary. Similar argument holds for $t^\star < 0$. This means that the entire line $\{\boldsymbol{u} + t\boldsymbol{v} : t \in \mathbb{R}\}$ must be in $D$. This holds for arbitrary $\boldsymbol{u} \in D$, so $D$ can be written as $D = D_0 + \mathcal{G}(f)^\perp$ where $D_0 \subseteq \mathcal{G}(f)$. Eq. (8) now implies that statement (1) holds over $D$. Since $f$ is closed, the statement also holds over $\operatorname{dom} f$, which then necessarily has form $\operatorname{dom} f = S_0 + \mathcal{G}(f)^\perp$ where $S_0 \subseteq \mathcal{G}(f)$. Therefore, statement (1) also holds for $\boldsymbol{u} \notin \operatorname{dom} f$.

The remaining statements are more straightforward. To prove the second statement, note that for any $\boldsymbol{u}' \in D$, its gradient can be decomposed as $\nabla f(\boldsymbol{u}') = P\nabla f(\boldsymbol{u}') + \boldsymbol{a}$. Thus, $\nabla f(\boldsymbol{u}) = P\nabla f(P\boldsymbol{u}) + \boldsymbol{a} = \nabla f(P\boldsymbol{u})$. For the third statement, we have $\nabla^2 f(\boldsymbol{u}) = P[\nabla^2 f(P\boldsymbol{u})]P = \nabla^2 f(P\boldsymbol{u})$, because $\operatorname{range}(\nabla^2 f(P\boldsymbol{u})) = \mathcal{G}(f)$. The fourth statement is equivalent to the fifth statement and they follow because $\operatorname{range}(\nabla^2 f(P\boldsymbol{u})) = \mathcal{G}(f)$. $\qquad\square$

This proposition immediately implies that the Hessian of $f$ can be expressed as a function of the gradient of $f$. We will denote such a function $H_f$:

**Proposition A.8.** *Let $f : \mathbb{R}^d \to \mathbb{R} \cup \{\infty\}$ be convex$^+$ and $D := \operatorname{int} \operatorname{dom} f$. Let $D' := \{\nabla f(\boldsymbol{u}) : \boldsymbol{u} \in D\}$ be the set of its gradients. Then there exists a map $H_f : D' \to \mathbb{R}^{d\times d}$ such that $\nabla^2 f(\boldsymbol{u}) = H_f(\nabla f(\boldsymbol{u}))$ for all $\boldsymbol{u} \in D$.*

*Proof.* Let $D_0 = D \cap \mathcal{G}(f)$. Proposition A.7 implies that $\nabla f$ is a bijection from $D_0$ to $D'$. Denoting its inverse from $D'$ to $D_0$ as $\boldsymbol{h}$, we can then define the map $H_f$ via $H_f(\boldsymbol{\mu}) = \nabla^2 f(\boldsymbol{h}(\boldsymbol{\mu}))$. Now, for any $\boldsymbol{u} \in D$, we have

$$\nabla^2 f(\boldsymbol{u}) = \nabla^2 f(P\boldsymbol{u}) = \nabla^2 f(\boldsymbol{h}(\nabla f(P\boldsymbol{u}))) = H_f(\nabla f(P\boldsymbol{u})) = H_f(\nabla f(\boldsymbol{u})) \ . \qquad\square$$

**Proposition A.9.** *Let $f : \mathbb{R}^d \to \mathbb{R} \cup \{\infty\}$ be a convex$^+$ function and let $\mathcal{A} = \operatorname{aff} \operatorname{dom} f^*$. Then there exists a convex$^+$ function $g : \mathbb{R}^d \to \mathbb{R} \cup \{\infty\}$ such that the following hold:*

1. $g$ *agrees with* $f^*$ *on* $\mathcal{A}$.

2. $\mathcal{G}(g)$ is parallel to $\mathcal{A}$, or equivalently $\mathcal{G}(g) = \mathcal{G}(f)$.
3. For $\boldsymbol{\mu} \in \operatorname{ri} \operatorname{dom} f^*$: $\partial f^*(\boldsymbol{\mu}) = \nabla g(\boldsymbol{\mu}) + \mathcal{G}(f)^\perp$.
4. For $\boldsymbol{\mu} \in \operatorname{ri} \operatorname{dom} f^*$: $\nabla^2 g(\boldsymbol{\mu}) = H_f^+(\boldsymbol{\mu})$.

*Proof.* We begin by representing $f$ via a function of Legendre type and then rely on the properties of such functions to obtain $g$. In particular, we note that by Proposition A.7, the function $f$ is defined by its values on $\mathcal{G}(f)$, except for a linear term, and so we construct a function $f_0$ that is a transformation of $f$ on $\mathcal{G}(f)$ and is of Legendre type.

To start, let $D = \operatorname{int} \operatorname{dom} f$, and $D_0 = D \cap \mathcal{G}(f)$. Assume that $\mathcal{G}(f)$ is $d_0$ dimensional $d_0 \leq d$, and let $A \in \mathbb{R}^{d \times d_0}$ be a matrix whose columns form an orthonormal basis of $\mathcal{G}(f)$. Then $A^\mathsf{T} A = I_{d_0}$, i.e., $A^\mathsf{T}$ is the left inverse of $A$. The matrix $A$ is an injective linear map from $\mathbb{R}^{d_0} \to \mathbb{R}^d$, but it is also a bijection from $\mathbb{R}^{d_0}$ to $\mathcal{G}(f)$. We also have $AA^\mathsf{T} = P$, where $P$ is the projection on $\mathcal{G}(f)$. Define the function $f_0 : \mathbb{R}^{d_0} \to \mathbb{R} \cup \{\infty\}$ as

$$f_0(\boldsymbol{v}) \coloneqq f(A\boldsymbol{v}) \ .$$

Let $S = \operatorname{int} \operatorname{dom} f_0$, so $S = A^\mathsf{T} D_0$. We have

$$\nabla f_0(\boldsymbol{v}) = A^\mathsf{T} \nabla f(A\boldsymbol{v}), \quad \nabla^2 f_0(\boldsymbol{v}) = A^\mathsf{T} \nabla^2 f(A\boldsymbol{v}) A, \quad \nabla^3 f_0(\boldsymbol{v})[\cdot, \cdot, \cdot] = \nabla^3 f(A\boldsymbol{v})[A(\cdot), A(\cdot), A(\cdot)],$$

so in particular $f_0$ has continuous third derivatives over $S$. We next argue that $f_0$ is of *Legendre type* in the sense of Rockafellar [15], page 258. For that we need to check that it satisfies the following conditions:

(a) *S is non-empty.*
This follows, because $D_0$ is non-empty and $D_0 \subseteq \mathcal{G}(f)$. Now, $A^\mathsf{T}$, as a linear map, is a bijection from $\mathcal{G}(f)$ to $\mathbb{R}^{d_0}$, so the set $S = A^\mathsf{T} D_0$ is also non-empty.

(b) $f_0$ *is differentiable throughout $S$.*
Similarly to the previous property, this holds, because $f$ is differentiable throughout $D_0$.

(c) $\lim_{t \to \infty} \|\nabla f_0(\boldsymbol{v}_t)\| = \infty$ *whenever* $\boldsymbol{v}_1, \boldsymbol{v}_2, \ldots$ *is a sequence in $S$ converging to a boundary point of $S$.*
If $\boldsymbol{u}_1, \boldsymbol{u}_2, \ldots$ is any sequence in $D_0$ converging to the relative boundary of $D_0$, then this point is on the boundary of $D$ and therefore $\|\nabla f(\boldsymbol{u}_t)\| \to \infty$, because $f$ is convex$^+$. Now, suppose we are given a sequence $\boldsymbol{v}_1, \boldsymbol{v}_2, \ldots$ in $S$ converging to a boundary point of $S$. Then $\boldsymbol{u}_t = A\boldsymbol{v}_t$ is exactly a sequence in $D_0$ converging to the relative boundary of $D_0$, so $\|\nabla f(A\boldsymbol{v}_t)\| \to \infty$. Since $\nabla f(A\boldsymbol{v}_t) \in \mathcal{G}(f)$ and the row space of $A^\mathsf{T}$ coincides with $\mathcal{G}(f)$, we also have $\|\nabla f_0(\boldsymbol{v}_t)\| = \|A^\mathsf{T} \nabla f(A\boldsymbol{v}_t)\| \to \infty$.

(d) $f_0$ *is strictly convex on $S$.*
Since $f_0$ has a continuous Hessian on $S$, it suffices to show that its Hessian is full rank, i.e., its range is $\mathbb{R}^{d_0}$. For any $\boldsymbol{v} \in S$, we have $\operatorname{range}(\nabla^2 f_0(\boldsymbol{v})) = \operatorname{range}(A^\mathsf{T} \nabla^2 f(A\boldsymbol{v}) A)$, which must be $\mathbb{R}^{d_0}$, because $\operatorname{range}(\nabla^2 f(A\boldsymbol{v})) = \mathcal{G}(f)$ and $A$ is a bijection from $\mathbb{R}^{d_0}$ to $\mathcal{G}(f)$.

We now express $f$ in terms of $f_0$. Note that by conjugacy, the affine hull of gradients of $f$ coincides with $\mathcal{A} = \operatorname{aff} \operatorname{dom} f^*$. By Proposition A.7, the function $f$ is defined by its values on $\mathcal{G}(f)$, except for the linear term, described by the unique $\boldsymbol{a} \in \mathcal{A} \cap \mathcal{G}^\perp$. We defined $f_0$ to exactly represent $f$ on $\mathcal{G}(f)$, so for any $\boldsymbol{u}' \in \mathcal{G}(f)$, we have $f_0(A^\mathsf{T} \boldsymbol{u}') = f(AA^\mathsf{T} \boldsymbol{u}') = f(\boldsymbol{u}')$, because $AA^\mathsf{T} = P$. Thus, for any $\boldsymbol{u} \in \mathbb{R}^d$ we have, by Proposition A.7,

$$f(\boldsymbol{u}) = f(P\boldsymbol{u}) + \boldsymbol{a}^\mathsf{T} \boldsymbol{u} = f_0(A^\mathsf{T} P\boldsymbol{u}) + \boldsymbol{a}^\mathsf{T} \boldsymbol{u} = f_0(A^\mathsf{T} \boldsymbol{u}) + \boldsymbol{a}^\mathsf{T} \boldsymbol{u} \ , \tag{9}$$

because the row space of $A^\mathsf{T}$ coincides with $\mathcal{G}(f)$. This relationship between $f$ and $f_0$ implies the following relationship for their conjugates (by Theorems 12.3 and 16.3 of Rockafellar [15]):

$$f^*(\boldsymbol{\mu}) = \inf_{\boldsymbol{y} \in \mathbb{R}^d : A\boldsymbol{y} = \boldsymbol{\mu} - \boldsymbol{a}} f_0^*(\boldsymbol{y}) \ , \tag{10}$$

where the infimum of an empty set is $\infty$. The linear map $A$ is injective and $\operatorname{range}(A) = \mathcal{G}(f)$, so the linear system $A\boldsymbol{v} = \boldsymbol{\mu} - \boldsymbol{a}$ has a single solution $\boldsymbol{v} = A^{\mathsf{T}}(\boldsymbol{\mu} - \boldsymbol{a})$ when $(\boldsymbol{\mu} - \boldsymbol{a}) \in \mathcal{G}(f)$, and no solutions when $(\boldsymbol{\mu} - \boldsymbol{a}) \notin \mathcal{G}(f)$. Therefore,

$$f^*(\boldsymbol{\mu}) = \begin{cases} f_0^*\big(A^{\mathsf{T}}(\boldsymbol{\mu} - \boldsymbol{a})\big) & \text{if } \boldsymbol{\mu} \in \mathcal{A}, \\ \infty & \text{if } \boldsymbol{\mu} \notin \mathcal{A}. \end{cases}$$

Since $f_0$ is of Legendre type, so is its conjugate $f_0^*$ (see Theorem 26.5 of Rockafellar [15]), which means that $f_0^*$ with $S^* := \operatorname{int} \operatorname{dom} f_0^*$ satisfies the properties (a)–(d). The function $g$ is constructed as follows:

$$g(\boldsymbol{\mu}) := f_0^*\big(A^{\mathsf{T}}(\boldsymbol{\mu} - \boldsymbol{a})\big) = f_0^*(A^{\mathsf{T}}\boldsymbol{\mu}) \ ,$$

where the equality follows because $\boldsymbol{a} \in \mathcal{G}(f)^{\perp}$. This differs from the expression for $f^*$ in that it does not equal to $\infty$ outside $\mathcal{A}$. Before we argue that $g$ is convex$^+$, we analyze the gradient, Hessian and third derivatives of $f_0^*$. From the properties of the conjugates, we know that $\nabla f_0^*$ is the inverse of $\nabla f_0$, so for all $\boldsymbol{y} \in S^*$,

$$\nabla f_0^*(\boldsymbol{y}) = [\nabla f_0]^{-1}(\boldsymbol{y}) \ .$$

Since $\nabla f_0$ is continuously differentiable and its derivative (i.e., the Hessian of $f_0$) is an invertible matrix, the inverse of $\nabla f_0$ is also continuously differentiable and its derivative (i.e., the Hessian of $f_0^*$) is

$$\nabla^2 f_0^*(\boldsymbol{y}) = \big[\nabla^2 f_0\big([\nabla f_0]^{-1}(\boldsymbol{y})\big)\big]^{-1} = \big[\nabla^2 f_0\big(\nabla f_0^*(\boldsymbol{y})\big)\big]^{-1} \ . \tag{11}$$

In particular note that $\nabla^2 f_0^*(\boldsymbol{y})$ is also an invertible matrix. We next show that $f_0^*$ has a continuous third derivative, by using the chain rule to argue that $\nabla^2 f_0^*$ is continuously differentiable. This follows, because $\nabla^2 f_0^*$ is the composition of (i) the matrix inversion $[\cdot]^{-1}$, (ii) the Hessian map $\nabla^2 f_0$, and (iii) the conjugate gradient $\nabla f_0^*$, and all of them are continuously differentiable at points where the respective derivatives are taken; specifically, the matrix inversion is taken at an invertible matrix $\nabla^2 f_0\big(\nabla f_0^*(\boldsymbol{y})\big)$, the Hessian at $\nabla f_0^*(\boldsymbol{y}) \in S$, and the conjugate gradient at $\boldsymbol{y} \in S^*$.

Now we can show that $g$ is convex$^+$. From the definition of $g$, we have

$$\nabla g(\boldsymbol{\mu}) = A \nabla f_0^*(A^{\mathsf{T}}\boldsymbol{\mu}), \quad \nabla^2 g(\boldsymbol{\mu}) = A \nabla f_0^*(A^{\mathsf{T}}\boldsymbol{\mu}) A^{\mathsf{T}}, \tag{12}$$

and

$$\nabla^3 g(\boldsymbol{\mu})[\cdot, \cdot, \cdot] = \nabla^3 f_0^*(A^{\mathsf{T}}\boldsymbol{\mu})[A^{\mathsf{T}}(\cdot), A^{\mathsf{T}}(\cdot), A^{\mathsf{T}}(\cdot)]. \tag{13}$$

Let $D^* := \operatorname{int} \operatorname{dom} g$. From the definition of $g$, we have $D^* = AS^* + \mathcal{G}(f)^{\perp}$. Note that $\mathcal{G}(f_0^*) = \mathbb{R}^{n_0}$, because $\operatorname{dom} f_0$ has a non-empty interior. Therefore, by Eq. (12), $\mathcal{G}(g) = \operatorname{range}(A) = \mathcal{G}(f)$. We next verify that $g$ satisfies properties (1)–(5) of convexity$^+$, relying on the properties (a)–(d) satisfied by $f_0^*$ and $S^*$:

1. *g is closed and convex.*
   This follows, because $f_0^*$ is closed and convex.
2. *$D^*$ is non-empty.*
   This follows, since $f_0^*$ satisfies (a), so $S^*$ is non-empty, and so is $D^*$.
3. *g has continuous third derivatives on $D^*$.*
   This follows by Eq. (13), because $f_0^*$ has continuous third derivatives on $S^*$.
4. $\operatorname{range}(\nabla^2 g(\boldsymbol{\mu})) = \mathcal{G}(g)$ *for all $\boldsymbol{\mu} \in D^*$.*
   Since the Hessians of $f_0^*$ are full rank, Eq. (12) implies that $\operatorname{range}(\nabla^2 g(\boldsymbol{\mu})) = \operatorname{range}(A) = \mathcal{G}(g)$.
5. $\lim_{t \to \infty} \|\nabla g(\boldsymbol{\mu}_t)\| = \infty$ *whenever $\boldsymbol{\mu}_1, \boldsymbol{\mu}_2, \ldots$ is a sequence in $D^*$ converging to a boundary point of $D^*$.*
   If $\boldsymbol{\mu}_t$ converges to a point on the border of $D^* = AS^* + \mathcal{G}(f)^{\perp}$, then the sequence of points $\boldsymbol{y}_t = A^{\mathsf{T}}\boldsymbol{\mu}_t$ converges to the border of $S^*$. Since $f_0^*$ satisfies property (c), this means that $\|\nabla f_0^*(\boldsymbol{y}_t)\| \to \infty$. And since $A$ is injective, also $\|\nabla g(\boldsymbol{\mu}_t)\| = \|A \nabla f_0^*(\boldsymbol{y}_t)\| \to \infty$.

We now prove that $g$ has the properties stated in the theorem:

1. *g agrees with $f^*$ on $\mathcal{A}$.*
   Immediate from the definition of $g$.

2. $\mathcal{G}(g)$ *is parallel to* $\mathcal{A}$, *or equivalently* $\mathcal{G}(g) = \mathcal{G}(f)$.
   As we already argued, Eq. (12) and the fact that $\mathcal{G}(f_0^*) = \mathbb{R}^{n_0}$ imply that $\mathcal{G}(g) = \text{range}(A) = \mathcal{G}(f)$.

3. *For* $\boldsymbol{\mu} \in \text{ri dom } f^*$: $\partial f^*(\boldsymbol{\mu}) = \nabla g(\boldsymbol{\mu}) + \mathcal{G}(f)^{\perp}$.
   Note that $\boldsymbol{\mu} \in \mathcal{A}$, so $(\boldsymbol{\mu} - \boldsymbol{a}) \in \mathcal{G}(f)$. The statement follows by the following chain of equivalences

$$
\begin{aligned}
\boldsymbol{u} \in \partial f^*(\boldsymbol{\mu}) \ \text{iff} \ &\nabla f(\boldsymbol{u}) = \boldsymbol{\mu} \\
\text{iff} \ &A \nabla f_0(A^\mathsf{T} \boldsymbol{u}) + \boldsymbol{a} = \boldsymbol{\mu} && \text{(by Eq. 9)} \\
\text{iff} \ &\nabla f_0(A^\mathsf{T} \boldsymbol{u}) = A^\mathsf{T}(\boldsymbol{\mu} - \boldsymbol{a}) = A^\mathsf{T} \boldsymbol{\mu} \\
& && \text{(because } (\boldsymbol{\mu} - \boldsymbol{a}) \in \mathcal{G}(f) = \text{range}(A)) \\
\text{iff} \ &\nabla f_0^*(A^\mathsf{T} \boldsymbol{\mu}) = A^\mathsf{T} \boldsymbol{u} \\
\text{iff} \ &A \nabla f_0^*(A^\mathsf{T} \boldsymbol{\mu}) = A A^\mathsf{T} \boldsymbol{u} = P \boldsymbol{u} && \text{(because } A \text{ is injective)} \\
\text{iff} \ &\nabla g(\boldsymbol{\mu}) = P \boldsymbol{u} && \text{(by Eq. 12)} \\
\text{iff} \ &\boldsymbol{u} \in \nabla g(\boldsymbol{\mu}) + \mathcal{G}(f)^{\perp} && \text{(because } \nabla g(\boldsymbol{\mu}) \in \text{range}(A) = \mathcal{G}(f))
\end{aligned}
$$

4. *For* $\boldsymbol{\mu} \in \text{ri dom } f^*$: $\nabla^2 g(\boldsymbol{\mu}) = H_f^+(\boldsymbol{\mu})$.
   From Eqs. (12) and (11), we have

$$
\begin{aligned}
\nabla^2 g(\boldsymbol{\mu}) &= A[\nabla^2 f_0^*(A^\mathsf{T} \boldsymbol{\mu})]A^\mathsf{T} \\
&= A\big[\nabla^2 f_0\big(\nabla f_0^*(A^\mathsf{T} \boldsymbol{\mu})\big)\big]^{-1} A^\mathsf{T} \ .
\end{aligned}
$$

By Proposition A.8, $H_f(\boldsymbol{\mu}) = \nabla^2 f(\boldsymbol{u})$ for any $\boldsymbol{u}$ such that $\nabla f(\boldsymbol{u}) = \boldsymbol{\mu}$, which is equivalent to $\boldsymbol{u} \in \partial f^*(\boldsymbol{\mu})$. Above, we have shown that $\nabla g(\boldsymbol{\mu}) \in \partial f^*(\boldsymbol{\mu})$, so $H_f(\boldsymbol{\mu}) = \nabla^2 f(\nabla g(\boldsymbol{\mu}))$. We continue the derivation of $H_f(\boldsymbol{\mu})$ using Eqs. (9) and (12):

$$
\begin{aligned}
H_f(\boldsymbol{\mu}) &= \nabla^2 f(\nabla g(\boldsymbol{\mu})) \\
&= A\big[\nabla^2 f_0\big(A^\mathsf{T} \nabla g(\boldsymbol{\mu})\big)\big]A^\mathsf{T} \\
&= A\big[\nabla^2 f_0\big(A^\mathsf{T} A \nabla f_0^*(A^\mathsf{T} \boldsymbol{\mu})\big)\big]A^\mathsf{T} \\
&= A\big[\nabla^2 f_0\big(\nabla f_0^*(A^\mathsf{T} \boldsymbol{\mu})\big)\big]A^\mathsf{T} \ ,
\end{aligned}
$$

where the last equation follows, because $A^\mathsf{T} A = I_{d_0}$. Since $\mathcal{G}(g) = \mathcal{G}(f)$, the ranges of $\nabla^2 g(\boldsymbol{\mu})$ and $H_f(\boldsymbol{\mu})$ coincide with $\mathcal{G}(f)$. From the above derivations of $\nabla^2 g(\boldsymbol{\mu})$ and $H_f(\boldsymbol{\mu})$, we also have

$$
[\nabla^2 g(\boldsymbol{\mu})]H_f(\boldsymbol{\mu}) = H_f(\boldsymbol{\mu})[\nabla^2 g(\boldsymbol{\mu})] = A A^\mathsf{T} = P \ ,
$$

so indeed $\nabla^2 g(\boldsymbol{\mu}) = H_f^+(\boldsymbol{\mu})$. $\qquad\square$

Thanks to the continuity of third derivatives of $f$ and the continuity of second derivatives of $g$ from Proposition A.9, we can easily prove a local Lipschitz property for $H_f$:

**Proposition A.10.** *Let* $f : \mathbb{R}^d \to \mathbb{R} \cup \{\infty\}$ *be convex*$^+$ *and* $D'$ *be the set of its gradients (necessarily open within* $\text{aff } D'$*). Let* $\boldsymbol{\mu} \in D'$ *and let* $B$ *be a closed ball (in* $\text{aff } D'$*) centered at* $\boldsymbol{\mu}$ *and fully contained in* $D'$*. Then there exists a constant* $c$ *such that for all* $\boldsymbol{\mu}' \in B$

$$
H_f(\boldsymbol{\mu}') \simeq \big(1 \pm c\|\boldsymbol{\mu}' - \boldsymbol{\mu}\|\big) H_f(\boldsymbol{\mu}) \ .
$$

*Proof.* Let $g$ be the function from Proposition A.9. Similarly as we argued in the proof of Proposition A.9, we can write

$$
H_f(\boldsymbol{\mu}) = \nabla^2 f(\nabla g(\boldsymbol{\mu})) \ ,
$$

because $\nabla g(\boldsymbol{\mu}) \in \partial f^*(\boldsymbol{\mu})$, and thus $\nabla f(\nabla g(\boldsymbol{\mu})) = \boldsymbol{\mu}$. Since, $g$ has continuous second derivatives and $f$ has continuous third derivatives, the Mean Value Theorem implies that $\nabla g$ is Lipschitz continuous on any compact subset of $D'$, and $\nabla^2 f$ is Lipschitz continuous on any compact subset of $\text{int dom } f$. Since $B$ is compact, and so is its image under $\nabla g$ by continuity of $\nabla g$, we obtain that both $\nabla g$ and $\nabla^2 f$ are Lipschitz on required sets, and so $H_f$ is also Lipschitz within $B$, with some constant $L$, i.e.,

$$
\|H_f(\boldsymbol{\mu}') - H_f(\boldsymbol{\mu})\| \le L\|\boldsymbol{\mu}' - \boldsymbol{\mu}\| \ .
$$

Since $\mathrm{range}(H_f(\boldsymbol{\mu}')) = \mathrm{range}(H_f(\boldsymbol{\mu})) = \mathcal{G}(f)$, this implies that

$$H_f(\boldsymbol{\mu}') \simeq H_f(\boldsymbol{\mu}) \pm L\|\boldsymbol{\mu}' - \boldsymbol{\mu}\|P \ ,$$

where $P$ is the projection on $\mathcal{G}(f)$. Since $\mathrm{range}(H_f(\boldsymbol{\mu})) = \mathcal{G}(f)$, we have $P \preceq \sigma^{-1}H_f(\boldsymbol{\mu})$ where $\sigma = \lambda_{\min}(H_f(\boldsymbol{\mu}))$ is the smallest positive eigenvalue of $H_f(\boldsymbol{\mu})$. Thus, we have

$$H_f(\boldsymbol{\mu}') \simeq \left(1 \pm L\sigma^{-1}\|\boldsymbol{\mu}' - \boldsymbol{\mu}\|\right)H_f(\boldsymbol{\mu}) \ . \qquad \square$$

### A.4 Lipschitz Gradients and Strong Convexity

In addition to (or instead of) convexity$^+$, some of our results require Lipschitz gradients or, dually, strong convexity. To be precise, we say that a differentiable function $f : \mathbb{R}^d \to \mathbb{R}$ has a Lipschitz gradient if there exists a constant $L$ such that $\|\nabla f(\boldsymbol{u}) - \nabla f(\boldsymbol{v})\| \le L\|\boldsymbol{u} - \boldsymbol{v}\|$ for all $\boldsymbol{u}, \boldsymbol{v} \in \mathbb{R}^d$. If $f$ is twice differentiable, it suffices to check that $\nabla^2 f(\boldsymbol{u}) \preceq LI_d$ for all $\boldsymbol{u}$, where $I_d \in \mathbb{R}^{d \times d}$ is the identity.

We say that $f$ is strongly convex with the strong convexity constant $\sigma$ if

$$f(\boldsymbol{v}) \ge f(\boldsymbol{u}) + \boldsymbol{g}^{\mathsf{T}}(\boldsymbol{v} - \boldsymbol{u}) + \frac{1}{2}\sigma\|\boldsymbol{v} - \boldsymbol{u}\|^2 \ ,$$

for all $\boldsymbol{v}, \boldsymbol{u} \in \mathbb{R}^d$ and $\boldsymbol{g} \in \partial f(\boldsymbol{u})$. A standard convex analysis result states that if $f : \mathbb{R}^d \to \mathbb{R}$ has a gradient with Lipschitz constant $L$ then $f^*$ is strongly convex with the strong convexity constant $\sigma = 1/L$ (see Prop. 12.60 of Rockafellar and Wets [16]).

## B  Proofs and Additional Results for Section 4

### B.1  Proof of Theorem 4.1

We prove a more explicit version of the theorem:

**Theorem B.1.** *Under the exponential trader model, $(\bar{\boldsymbol{r}}, \bar{\boldsymbol{c}}, \bar{\boldsymbol{\mu}})$ is a market-clearing equilibrium if and only if*

$$\bar{\boldsymbol{r}} \in \underset{\boldsymbol{r}: \sum_{i=1}^N \boldsymbol{r}_i = \boldsymbol{0}}{\operatorname{argmin}} \sum_{i=1}^N F_i(-\boldsymbol{r}_i), \qquad \sum_{i=1}^N \bar{c}_i = 0, \qquad and \qquad \bar{\boldsymbol{\mu}} = \nabla T(\tilde{\boldsymbol{\theta}}_i - a_i\bar{\boldsymbol{r}}_i) \ \forall i \in [N].$$

*A market-clearing equilibrium always exists. Furthermore, for any market-clearing equilibrium, the equilibrium prices are unique solutions of the following dual problem:*

$$\bar{\boldsymbol{\mu}} = \underset{\boldsymbol{\mu} \in \mathbb{R}^K}{\operatorname{argmin}} \left[\sum_i F_i^*(\boldsymbol{\mu})\right] \ .$$

*Proof.* We first express the market-clearing equilibrium definition using the trader potential functions $F_i$ instead of trader utilities. Since $[-F_i(-\boldsymbol{r}_i) + c_i]$ is a monotone one-to-one transformation of the utility $U_i(\boldsymbol{r}_i, c_i)$, we get the following equivalences

$$\boldsymbol{0} \in \underset{\boldsymbol{\delta} \in \mathbb{R}^K}{\operatorname{argmax}} U_i(\bar{\boldsymbol{r}}_i + \boldsymbol{\delta}, \bar{c}_i - \boldsymbol{\delta} \cdot \bar{\boldsymbol{\mu}}) \text{ iff } \boldsymbol{0} \in \underset{\boldsymbol{\delta} \in \mathbb{R}^K}{\operatorname{argmin}}\left[F_i(-\bar{\boldsymbol{r}}_i - \boldsymbol{\delta}) - \bar{c}_i + \boldsymbol{\delta} \cdot \bar{\boldsymbol{\mu}}\right]$$

$$\text{iff } \nabla F_i(-\bar{\boldsymbol{r}}_i) = \bar{\boldsymbol{\mu}} \ ,$$

where the last step follows by setting the gradient of the objective to zero at $\boldsymbol{\delta} = \boldsymbol{0}$. Thus, we have that $(\bar{\boldsymbol{r}}, \bar{\boldsymbol{c}}, \bar{\boldsymbol{\mu}})$ is a market-clearing equilibrium iff

$$\sum_{i=1}^N \bar{\boldsymbol{r}}_i = \boldsymbol{0}, \quad \sum_{i=1}^N \bar{c}_i = 0, \quad \nabla F_i(-\bar{\boldsymbol{r}}_i) = \bar{\boldsymbol{\mu}} \text{ for all } i \in [N]. \tag{14}$$

We now analyze the minimization of the potential $\sum_i F_i(-\boldsymbol{r}_i)$ subject to the market clearing constraint $\sum_i \boldsymbol{r}_i = \boldsymbol{0}$. We express this constraint using the convex indicator function $\mathbb{I}\{\cdot\}$, which equals zero if its argument is true and $\infty$ when its false. We also introduce the matrix $A \in \mathbb{R}^{K \times NK}$ with the block structure $A := (I_K \ I_K \ \cdots \ I_K)$ where $I_K$ is the $K \times K$ identity matrix. Thus, $A$ implements

the summation over the blocks $r_i$, since $Ar = \sum_i r_i$. With this notation, the potential minimization problem can be written as

$$\min_{r \in \mathbb{R}^{NK}} \Big[ \underbrace{\sum_{i=1}^{N} F_i(-r_i)}_{f(-r)} + \underbrace{\mathbb{I}\{Ar = 0\}}_{g(Ar)} \Big] \ , \tag{15}$$

where we introduced the functions $f(r) = \sum_i F_i(r_i)$ and $g(s) = \mathbb{I}\{s = 0\}$ for $s \in \mathbb{R}^K$. Now, if certain conditions are satisfied, we can apply Fenchel's duality (Theorem A.4) and obtain that the value of the primal (15) equals the value of the following dual problem

$$\max_{\mu \in \mathbb{R}^K} \Big[ -f^*(A^\mathsf{T}\mu) - g^*(\mu) \Big] \ ,$$

which is equivalent to

$$\max_{\mu \in \mathbb{R}^K} \Big[ -\sum_{i=1}^{N} F_i^*(\mu) \Big] \ , \tag{16}$$

because $g^*(\mu) = 0$ and $f^*(y) = \sum_i F_i^*(y_i)$, for $y \in \mathbb{R}^{NK}$, so $f^*(A^\mathsf{T}\mu) = \sum_i F_i^*(\mu)$. It remains to verify that the preconditions of Theorem A.4 are satisfied. First, we need to check that there exists $r$ such that $Ar \in \mathrm{ri}(\mathrm{dom}\, g)$. Since $\mathrm{ri}(\mathrm{dom}\, g) = \{0\}$, the vector $r = 0$ satisfies this. We also need to check that there exist $\mu$ such that $A^\mathsf{T}\mu \in \mathrm{ri}(\mathrm{dom}\, f^*)$, which for our choices of $A$ and $f$ is equivalent to $\mu \in \mathrm{ri}(\mathrm{dom}\, F_i^*)$ for all $i \in [N]$. Since $\mathrm{dom}\, F_i^* = \mathrm{dom}\, T^* = \mathcal{M}$, any $\mu \in \mathcal{M}$ satisfies this. Thus, conclusions of Theorem A.4 hold.

The conclusions state that both the primal and the dual are attained, and $\hat{r}$ and $\hat{\mu}$ are their solutions if and only if $A^\mathsf{T}\hat{\mu} = \nabla f(-\hat{r})$ and $\hat{\mu} \in \partial g(A\hat{r})$. From the definitions of $A$ and $f$, the first condition is equivalent to

$$\hat{\mu} = \nabla F(-\hat{r}_i) = \nabla T(\tilde{\theta}_i - a_i\hat{r}_i) \ .$$

The second condition is by conjugacy equivalent to $A\hat{r} = \nabla g(\hat{\mu}) = 0$, i.e.,

$$\sum_{i=1}^{N} \hat{r}_i = 0 \ .$$

This establishes that $\hat{r}$ and $\hat{\mu}$ are solutions to the primal (15) and dual (16), if and only if they satisfy the conditions in (14), i.e., if and only if they form a market-clearing equilibrium. This proves the theorem except for the uniqueness of the equilibrium prices $\bar{\mu}$. The uniqueness follows from the fact that $\bar{\mu}$ minimizes $\sum_i F_i^*(\mu)$, and the functions $F_i^*$ are strongly convex on their domain $\mathcal{M}$, which in turn follows because the functions $F_i$ have Lipschitz gradients (a property they inherit from the log partition function $T$). $\qquad\square$

## B.2 Proof of Theorem 4.2

We prove a more explicit version of the theorem:

**Theorem B.2.** *Under the exponential trader model, $(r^\star, c^\star, \mu^\star)$ is a market-maker equilibrium for cost function $C_b$ if and only if, for the market state $s^\star = \sum_{i=1}^{N} r_i^\star$,*

$$r^\star \in \underset{r}{\operatorname{argmin}}\, F(r), \quad \sum_{i=1}^{N} c_i^\star = C_b(0) - C_b(s^\star), \quad \text{and} \quad \mu^\star = \nabla C_b(s^\star) = \nabla T(\tilde{\theta}_i - a_i r_i^\star)\, \forall i \in [N].$$

*A market-maker equilibrium always exists. Furthermore, for any market-maker equilibrium, the equilibrium prices are unique solutions of the following dual problem:*

$$\mu^\star = \underset{\mu \in \mathbb{R}^K}{\operatorname{argmin}} \Big[ \sum_i F_i^*(\mu) + bC^*(\mu) \Big] \ .$$

*Proof.* We proceed similarly to the proof of Theorem B.1. We first express the market-maker equilibrium definition using trader potentials instead of trader utilities:

$$0 \in \underset{\delta \in \mathbb{R}^K}{\operatorname{argmax}}\, U_i\Big(r_i^\star + \delta,\, c_i^\star - C_b(s^\star + \delta) + C_b(s^\star)\Big)$$

$$\text{iff } \mathbf{0} \in \operatorname*{argmin}_{\boldsymbol{\delta} \in \mathbb{R}^K} \big[ F_i(-\boldsymbol{r}_i^\star - \boldsymbol{\delta}) - c_i^\star + C_b(\boldsymbol{s}^\star + \boldsymbol{\delta}) - C_b(\boldsymbol{s}^\star) \big]$$

$$\text{iff } \nabla F_i(-\boldsymbol{r}_i^\star) = \nabla C_b(\boldsymbol{s}^\star) \ .$$

Thus, we have that $(\boldsymbol{r}^\star, \boldsymbol{c}^\star, \boldsymbol{\mu}^\star)$ is a market-maker equilibrium iff, for the market state $\boldsymbol{s}^\star = \sum_{i=1}^N \boldsymbol{r}_i^\star$,

$$\sum_{i=1}^N c_i^\star + C_b(\boldsymbol{s}^\star) - C_b(\mathbf{0}) = 0, \quad \boldsymbol{\mu}^\star = \nabla C_b(\boldsymbol{s}^\star), \quad \nabla F_i(-\bar{\boldsymbol{r}}_i) = \nabla C_b(\boldsymbol{s}^\star) \text{ for all } i \in [N]. \quad (17)$$

We next use Fenchel's duality to analyze minimization of the potential $F(\boldsymbol{r}) = \sum_i F_i(-\boldsymbol{r}_i) + C_b(\sum_i \boldsymbol{r}_i)$. We again define $A := (I_K \ I_K \ \cdots \ I_K)$ and $f(\boldsymbol{r}) := \sum_i F_i(\boldsymbol{r}_i)$, but in this case set $g(\boldsymbol{s}) = C_b(\boldsymbol{s})$. Therefore, by Theorem A.4, we obtain the following correspondence between the primal and the dual:

$$\min_{\boldsymbol{r} \in \mathbb{R}^{NK}} \left[ \sum_{i=1}^N F_i(-\boldsymbol{r}_i) + C_b(A\boldsymbol{r}) \right] = \min_{\boldsymbol{r} \in \mathbb{R}^{NK}} \left[ f(-\boldsymbol{r}) + g(A\boldsymbol{r}) \right] \quad (18)$$

$$= \max_{\boldsymbol{\mu} \in \mathbb{R}^K} \left[ -f^*(A^\mathsf{T} \boldsymbol{\mu}) - g^*(\boldsymbol{\mu}) \right]$$

$$= \max_{\boldsymbol{\mu} \in \mathbb{R}^K} \left[ -\sum_{i=1}^N F_i^*(\boldsymbol{\mu}) - bC^*(\boldsymbol{\mu}) \right] \ , \quad (19)$$

where we used the fact that $g(\boldsymbol{r}) = bC(\boldsymbol{r}/b)$ and therefore $g^*(\boldsymbol{\mu}) = bC^*(\boldsymbol{r})$ (this is immediate from the definition of conjugate). It remains to check the preconditions of Theorem A.4. First, we need to check that there exists $\boldsymbol{r}$ such that $A\boldsymbol{r} \in \mathrm{ri}(\mathrm{dom}\, g)$. Since $\mathrm{ri}(\mathrm{dom}\, C_b) = \mathbb{R}^K$, this is vacuous and any vector $\boldsymbol{r}$ satisfies this. We also need to check that there exist $\boldsymbol{\mu}$ such that $A^\mathsf{T} \boldsymbol{\mu} \in \mathrm{ri}(\mathrm{dom}\, f^*)$, which is equivalent to $\boldsymbol{\mu} \in \mathrm{ri}(\mathrm{dom}\, F_i^*)$ for all $i \in [N]$. Since $\mathrm{dom}\, F_i^* = \mathrm{dom}\, T^* = \mathcal{M}$, any $\boldsymbol{\mu} \in \mathcal{M}$ satisfies this. Thus, conclusions of Theorem A.4 hold.

The conclusions state that both the primal and the dual are attained, and $\hat{\boldsymbol{r}}$ and $\hat{\boldsymbol{\mu}}$ are their solutions if and only if $A^\mathsf{T} \hat{\boldsymbol{\mu}} = \nabla f(-\hat{\boldsymbol{r}})$ and $\hat{\boldsymbol{\mu}} = \nabla g(A\hat{\boldsymbol{r}})$. As in the proof of Theorem B.1, for our $A$ and $f$, the first condition is equivalent to

$$\hat{\boldsymbol{\mu}} = \nabla F(-\hat{\boldsymbol{r}}_i) = \nabla T(\tilde{\boldsymbol{\theta}}_i - a_i \hat{\boldsymbol{r}}_i) \ .$$

The second condition, $g(\boldsymbol{s}) = C_b(\boldsymbol{s})$, is

$$\hat{\boldsymbol{\mu}} = \nabla C_b\left( \sum_{i=1}^N \hat{\boldsymbol{r}}_i \right) \ .$$

This establishes that $\hat{\boldsymbol{r}}$ and $\hat{\boldsymbol{\mu}}$ are solutions to the primal (18) and dual (19), if and only if they satisfy the conditions in (17), i.e., if and only if they form a market-maker equilibrium. It remains to show that $\boldsymbol{\mu}^\star$ is unique. As before this follows by strong convexity of $F_i^*$ and the fact that $\boldsymbol{\mu}^\star$ minimizes $\sum_i F_i^*(\boldsymbol{\mu}) + bC^*(\boldsymbol{\mu})$. $\qquad\square$

### B.3 Proof of Theorem 4.3

By Theorem 4.1, $\bar{\boldsymbol{\mu}} = \operatorname{argmin}_{\boldsymbol{\mu} \in \mathbb{R}^K} [\sum_i F_i^*(\boldsymbol{\mu})]$ and from the first-order optimality

$$\mathbf{0} \in \partial \left[ \sum_i F_i^*(\bar{\boldsymbol{\mu}}) \right] \ .$$

Since $F_i(\boldsymbol{s}) = \frac{1}{a_i} T(\tilde{\boldsymbol{\theta}}_i + a_i \boldsymbol{s})$, the properties of the conjugates (Theorems 12.3 and 16.1 of Rockafellar [15]) yield

$$F_i^*(\boldsymbol{\mu}) = \frac{1}{a_i} \big( T^*(\boldsymbol{\mu}) - \tilde{\boldsymbol{\theta}}_i \cdot \boldsymbol{\mu} \big) \ .$$

Thus,

$$\sum_i F_i^*(\bar{\boldsymbol{\mu}}) = \left[ \sum_i 1/a_i \right] T^*(\boldsymbol{\mu}) - \left[ \sum_i \tilde{\boldsymbol{\theta}}_i / a_i \right] \cdot \boldsymbol{\mu}$$

$$\implies \partial\left[\sum_i F_i^*(\bar{\boldsymbol{\mu}})\right] = \left[\sum_i 1/a_i\right]\partial T^*(\boldsymbol{\mu}) - \left[\sum_i \tilde{\boldsymbol{\theta}}_i/a_i\right] \ .$$

Therefore, $\mathbf{0} \in \partial[\sum_i F_i^*(\bar{\boldsymbol{\mu}})]$ iff

$$\frac{\sum_i \tilde{\boldsymbol{\theta}}_i/a_i}{\sum_i 1/a_i} \in \partial T^*(\bar{\boldsymbol{\mu}}) \ ,$$

which is equivalent to

$$\bar{\boldsymbol{\mu}} = \nabla T\left(\frac{\sum_i \tilde{\boldsymbol{\theta}}_i/a_i}{\sum_i 1/a_i}\right) = \nabla T(\bar{\boldsymbol{\theta}}) = \mathbb{E}_{\bar{\boldsymbol{\theta}}}\left[\boldsymbol{\phi}(\omega)\right] \ ,$$

where $\bar{\boldsymbol{\theta}} := (\sum_i \tilde{\boldsymbol{\theta}}_i/a_i)/(\sum_i 1/a_i)$ and the last equality follows from the properties of the log partition function.

## B.4 Sampling Error

The market's forecasting ability is fundamentally limited by the information present among the population of traders, and the traders' risk attitudes in communicating their information via trades. In this appendix, we quantify these sources of error by analyzing the discrepancy $\|\boldsymbol{\mu}^{\text{true}} - \bar{\boldsymbol{\mu}}\|$ between the true expected security values and the market-clearing equilibrium prices.

The characterization of $\bar{\boldsymbol{\mu}}$ in Theorem 4.3 reflects two possible sources of error. First, the beliefs $\tilde{\boldsymbol{\theta}}_i$ are typically noisy signals of the ground truth. Second, beliefs are weighted according to risk aversions $a_i$, which can skew the prices. To formalize the latter concept, we write

$$N_{\text{eff}} = \frac{\left(\sum_i a_i^{-1}\right)^2}{\left(\sum_i a_i^{-2}\right)}$$

to denote the *effective sample size* of the weighted average. When risk aversion coefficients are equal across agents, we have $N_{\text{eff}} = N$, and when one agent has much smaller risk aversion than the others, $N_{\text{eff}} \to 1$. As the next result shows, the magnitude of the sampling error depends on the effective sample size as it relates to the number of securities and the variance in trader beliefs.

**Theorem B.3.** *Under the exponential trader model, assume that the beliefs $\tilde{\boldsymbol{\theta}}_i$ are drawn independently for each trader $i \in [N]$ with mean $\mathbb{E}[\tilde{\boldsymbol{\theta}}_i] = \boldsymbol{\theta}^{\text{true}}$ and covariance $\mathbb{V}(\tilde{\boldsymbol{\theta}}_i) \preceq \sigma^2 I_K$ for some $\sigma^2 \geq 0$. For any $\delta \in (0,1)$, the market-clearing prices $\bar{\boldsymbol{\mu}}$ satisfy, with probability at least $1 - \delta$, $\|\bar{\boldsymbol{\mu}} - \boldsymbol{\mu}^{\text{true}}\| \leq O\left(\sigma\sqrt{K/(N_{\text{eff}}\,\delta)}\right)$. Furthermore, assuming that each $a_i$ lies in a bounded range $[a_{\min}, a_{\max}]$ where $a_{\min}, a_{\max} > 0$, we have that $N_{\text{eff}} \to \infty$ as $N \to \infty$.*

*Proof.* We write $w_i = a_i^{-1}/(\sum_j a_j^{-1})$ for the weights in the average. Note that $N_{\text{eff}} = (\sum_i w_i^2)^{-1}$. By the fact that beliefs are independent, we have:

$$\mathbb{E}\left[\sum_{i=1}^N w_i \tilde{\boldsymbol{\theta}}_i\right] = \boldsymbol{\theta}^{\text{true}} \text{ and } \mathbb{V}\left(\sum_{i=1}^N w_i \tilde{\boldsymbol{\theta}}_i\right) \preceq N_{\text{eff}}^{-1}\sigma^2 I_K.$$

By applying the multidimensional version of Chebyshev's inequality, we therefore have

$$\Pr\left(\left\|\sum_{i=1}^N w_i \tilde{\boldsymbol{\theta}}_i - \boldsymbol{\theta}^{\text{true}}\right\| > t\right) \leq \frac{K\sigma^2}{N_{\text{eff}}\,t^2}.$$

The result then follows from the fact that $\bar{\boldsymbol{\mu}} = \nabla T(\sum_i w_i \tilde{\boldsymbol{\theta}}_i)$ by Theorem 4.3, the fact that $\boldsymbol{\mu}^{\text{true}} = \nabla T(\boldsymbol{\theta}^{\text{true}})$, and the Lipschitz continuity of $\nabla T$.

For the final claim, we have

$$N_{\text{eff}} = \frac{\left(\sum_i a_i^{-1}\right)^2}{\left(\sum_i a_i^{-2}\right)} \geq \frac{\left(\sum_i a_{\max}^{-1}\right)^2}{\left(\sum_i a_{\min}^{-2}\right)} = N(a_{\min}/a_{\max})^2. \qquad \square$$

Theorem B.3 implies that as the number of traders grows large, the market prices $\bar{\boldsymbol{\mu}}$ converge to the ground truth $\boldsymbol{\mu}^{\text{true}}$ in probability. It is important to note that this relies on Theorem 4.3, which is an artifact of exponential utility—for general utilities, market-clearing prices may not be consistent in the statistical sense, and this extra discrepancy would need to be quantified in the error decomposition.

For a finite number of traders, the bound in Theorem B.3 increases with the belief variance and the number of securities, as one would expect. It decreases with the effective sample size: the information incorporated into market-clearing prices improves when risk aversions are more uniform, and when the number of agents increases.

## C  Proofs and Additional Results for Section 5

### C.1  Proof of Theorem 5.1

Since gradients $\nabla F_i$ are Lipschitz, the functions $F_i^*$ are *strongly convex* over their domain, which is $\mathcal{M}$. Therefore, their sum $G(\boldsymbol{\mu}) := \sum_i F_i^*(\boldsymbol{\mu})$ is also strongly convex on $\mathcal{M}$ with some strong convexity constant $\sigma$. Since $\bar{\boldsymbol{\mu}} \in \text{ri}\,\mathcal{M}$, $G$ is subdifferentiable at $\bar{\boldsymbol{\mu}}$, and since $\bar{\boldsymbol{\mu}}$ minimizes $G$, any element $\boldsymbol{g} \in \partial G(\bar{\boldsymbol{\mu}})$ satisfies $\boldsymbol{g}^{\mathsf{T}}(\boldsymbol{\mu} - \bar{\boldsymbol{\mu}}) = 0$ for all $\boldsymbol{\mu} \in \mathcal{M}$. This together with strong convexity yields the lower bound $\sum_i F_i^*(\boldsymbol{\mu}) \geq \sum_i F_i^*(\bar{\boldsymbol{\mu}}) + \frac{1}{2}\sigma\|\boldsymbol{\mu} - \bar{\boldsymbol{\mu}}\|^2$. At the same time, $C^*(\boldsymbol{\mu})$ is bounded below by a linear function of the form $C^*(\bar{\boldsymbol{\mu}}) + \boldsymbol{u}^{\mathsf{T}}(\boldsymbol{\mu} - \bar{\boldsymbol{\mu}})$, because $C^*$ is subdifferentiable at $\bar{\boldsymbol{\mu}}$ since $\bar{\boldsymbol{\mu}} \in \text{ri}\,\mathcal{M} \subseteq \text{ri}\,\text{dom}\,C^*$.

Now from the optimality of $\boldsymbol{\mu}^{\star}$ and the lower bounds on $\sum_i F_i^*(\boldsymbol{\mu})$ and $C^*(\boldsymbol{\mu})$, we have

$$\sum_i F_i^*(\bar{\boldsymbol{\mu}}) + bC^*(\bar{\boldsymbol{\mu}}) \geq \sum_i F_i^*(\boldsymbol{\mu}^{\star}) + bC^*(\boldsymbol{\mu}^{\star})$$

$$\geq \left(\sum_i F_i^*(\bar{\boldsymbol{\mu}}) + \frac{1}{2}\sigma\|\boldsymbol{\mu}^{\star} - \bar{\boldsymbol{\mu}}\|^2\right) + b\left(C^*(\bar{\boldsymbol{\mu}}) + \boldsymbol{u}^{\mathsf{T}}(\boldsymbol{\mu}^{\star} - \bar{\boldsymbol{\mu}})\right)$$

$$\implies \|\boldsymbol{\mu}^{\star} - \bar{\boldsymbol{\mu}}\|^2 \leq -\frac{2b}{\sigma}\boldsymbol{u}^{\mathsf{T}}(\boldsymbol{\mu}^{\star} - \bar{\boldsymbol{\mu}})$$

$$\implies \|\boldsymbol{\mu}^{\star} - \bar{\boldsymbol{\mu}}\| \leq \frac{2b}{\sigma}\|\boldsymbol{u}\| \ .$$

### C.2  A Remark on Partial and Incoherent Beliefs

The proof of Theorem 5.1 crucially relies on the fact that $\text{dom}(\sum_i F_i^*) = \mathcal{M} \subseteq \text{dom}\,C^*$, i.e., that the cost function $C$ does not force any additional constraints on $\boldsymbol{\mu}$ beyond those already represented by the trader potentials $F_i$. This is natural in our setting, because trader utilities restrict the equilibrium prices to lie in the smallest set including all coherent price vectors, $\mathcal{M}$. This property of the trader utilities means that the traders would be always willing to trade if the prices were outside the set $\mathcal{M}$. If the trader utilities did not have this property, for instance, if each trader was interested in only a few securities, or their beliefs were incoherent, then this result might not hold. In such a setting, we might end up with $\bar{\boldsymbol{\mu}} \notin \text{dom}\,C^*$. At best, we could then show that

$$\lim_{b \to 0} \boldsymbol{\mu}^{\star}(b; C) = \underset{\boldsymbol{\mu} \in \text{dom}\,C^*}{\text{argmin}} \left[\sum_{i=1}^N F_i^*(\boldsymbol{\mu})\right] .$$

This, of course, agrees with Theorem 5.1 for our specific setting when the restriction to $\text{dom}\,C^*$ creates no additional constraints, because $\text{dom}(\sum_i F_i^*) \subseteq \text{dom}\,C^*$.

### C.3  Proof of Theorem 5.6

The proof will proceed by analyzing the Taylor expansion of the dual objective characterizing $\boldsymbol{\mu}^{\star}$. However, the functions $F_i^*$ and $C^*$ might not be differentiable in the standard sense, because their domains might have empty interiors (and only non-empty relative interiors). Fortunately, $F_i$ and $C$ are convex$^+$, so by Proposition A.9 there exist convex$^+$ functions $G_i$ and $R$ that coincide with $F_i^*$ and $C^*$ on $\text{aff}\,\text{dom}\,F_i^*$ and $\text{aff}\,\text{dom}\,C^*$. These functions are three times continuously differentiable, which is what we need to obtain the third order Taylor expansion.

Note that $\operatorname{dom} F_i^* = \mathcal{M}$ for all $i \in [N]$. Let $\mathcal{A}$ denote the affine hull of $\mathcal{M} = \operatorname{dom} F_i^*$. Since $\mathcal{M} \subseteq \operatorname{dom} C^*$, the dual in Eq. (6) is equivalent to

$$\boldsymbol{\mu}^\star = \operatorname*{argmin}_{\boldsymbol{\mu} \in \mathcal{A}} \left[ \sum_{i=1}^N G_i(\boldsymbol{\mu}) + b R(\boldsymbol{\mu}) \right] \quad . \tag{20}$$

Note that $\bar{\boldsymbol{\mu}} \in \operatorname{ri} \mathcal{M}$, because by the definition $\bar{\boldsymbol{\mu}} = \nabla T(\bar{\boldsymbol{s}})$ for some $\bar{\boldsymbol{s}}$ and the gradients of $T$ are in $\operatorname{ri} \mathcal{M}$. Thus, functions $G_i$ and $R$ are differentiable and have Hessians at $\bar{\boldsymbol{\mu}}$ (by convexity$^+$). We apply the Taylor expansion at $\bar{\boldsymbol{\mu}}$ to analyze the value of the objective at $\boldsymbol{\mu}^\star$. Let $G(\boldsymbol{\mu}) := \sum_i G_i(\boldsymbol{\mu})$. By Mean Value Theorem, we have

$$\nabla G(\boldsymbol{\mu}^\star) = \nabla G(\bar{\boldsymbol{\mu}}) + \nabla^2 G(\boldsymbol{\mu}_G)(\bar{\boldsymbol{\mu}} - \boldsymbol{\mu}^\star)$$
$$\nabla R(\boldsymbol{\mu}^\star) = \nabla R(\bar{\boldsymbol{\mu}}) + \nabla^2 R(\boldsymbol{\mu}_R)(\bar{\boldsymbol{\mu}} - \boldsymbol{\mu}^\star)$$

for some $\boldsymbol{\mu}_G$ and $\boldsymbol{\mu}_R$ on the line segment connecting $\boldsymbol{\mu}^\star$ with $\bar{\boldsymbol{\mu}}$. By Theorem 5.1, $\|\boldsymbol{\mu}^\star - \bar{\boldsymbol{\mu}}\| = O(b)$ as $b \to 0$, and thus also $\|\boldsymbol{\mu}_G - \bar{\boldsymbol{\mu}}\| = O(b)$ and $\|\boldsymbol{\mu}_R - \bar{\boldsymbol{\mu}}\| = O(b)$. By the continuity of third derivatives of $G$ and $R$ in the neighborhood of $\bar{\boldsymbol{\mu}}$, the Hessians of $G$ and $R$ are Lipschitz in some neighborhood of $\bar{\boldsymbol{\mu}}$, which means that

$$\nabla^2 G(\boldsymbol{\mu}_G) = \nabla^2 G(\bar{\boldsymbol{\mu}}) + \Delta_G$$
$$\nabla^2 R(\boldsymbol{\mu}_R) = \nabla^2 R(\bar{\boldsymbol{\mu}}) + \Delta_R$$

where $\Delta_G$ and $\Delta_R$ are matrices with $\|\Delta_G\| = O(\|\boldsymbol{\mu}_G - \bar{\boldsymbol{\mu}}\|) = O(b)$ and $\|\Delta_R\| = O(\|\boldsymbol{\mu}_R - \bar{\boldsymbol{\mu}}\|) = O(b)$.

We next calculate Hessians $\nabla^2 G(\bar{\boldsymbol{\mu}})$ and $\nabla^2 R(\bar{\boldsymbol{\mu}})$. First, note that

$$\nabla^2 G(\boldsymbol{\mu}) = \sum_i \nabla^2 G_i(\boldsymbol{\mu}) \quad ,$$

and by Proposition A.9, we have

$$\nabla^2 G_i(\boldsymbol{\mu}) = H_{F_i}^+(\boldsymbol{\mu}) = (1/a_i) H_T^+(\boldsymbol{\mu}) \quad ,$$

where the last equality follows, because $H_{F_i}(\boldsymbol{\mu}) = a_i H_T(\boldsymbol{\mu})$ from the definition of $F_i$. Thus,

$$\nabla^2 G(\boldsymbol{\mu}) = \sum_i (1/a_i) H_T^+(\boldsymbol{\mu}) = (N/\bar{a}) H_T^+(\boldsymbol{\mu}) \quad .$$

We also have

$$\nabla^2 R(\boldsymbol{\mu}) = H_C^+(\boldsymbol{\mu}) \quad .$$

By Proposition A.9, the affine space $\mathcal{A}$ is parallel to $\mathcal{G}(T)$. From the optimality of $\boldsymbol{\mu}^\star$ in (20), we have $\big( \nabla G(\boldsymbol{\mu}^\star) + b \nabla R(\boldsymbol{\mu}^\star) \big) \perp \mathcal{G}(T)$. Thus, writing $P$ for the projection on $\mathcal{G}(T)$, we obtain

$$\mathbf{0} = P \Big( \nabla G(\boldsymbol{\mu}^\star) + b \nabla R(\boldsymbol{\mu}^\star) \Big)$$
$$= P \nabla G(\bar{\boldsymbol{\mu}}) + P \nabla^2 G(\bar{\boldsymbol{\mu}})(\boldsymbol{\mu}^\star - \bar{\boldsymbol{\mu}}) + \underbrace{P \Delta_G (\boldsymbol{\mu}^\star - \bar{\boldsymbol{\mu}})}_{\boldsymbol{\varepsilon}_G} + b P \nabla R(\bar{\boldsymbol{\mu}}) + \underbrace{b P \big[ \nabla^2 R(\bar{\boldsymbol{\mu}}) + \Delta_R \big](\boldsymbol{\mu}^\star - \bar{\boldsymbol{\mu}})}_{\boldsymbol{\varepsilon}_R}$$

$$\tag{21}$$

$$= \nabla^2 G(\bar{\boldsymbol{\mu}})(\boldsymbol{\mu}^\star - \bar{\boldsymbol{\mu}}) + b P \nabla R(\bar{\boldsymbol{\mu}}) + \boldsymbol{\varepsilon}_G + \boldsymbol{\varepsilon}_R \quad . \tag{22}$$

In Eq. (21), the terms $\boldsymbol{\varepsilon}_G$ and $\boldsymbol{\varepsilon}_R$ have norms $O(b^2)$, because $\|\boldsymbol{\mu}^\star - \bar{\boldsymbol{\mu}}\|$, $\|\Delta_G\|$ and $\|\Delta_R\|$ are all at most $O(b)$. In Eq. (22), we use that $P \nabla G(\bar{\boldsymbol{\mu}}) = \mathbf{0}$ by optimality of $\bar{\boldsymbol{\mu}}$. We also use that $P \nabla^2 G(\bar{\boldsymbol{\mu}}) = \nabla^2 G(\bar{\boldsymbol{\mu}})$, because $\operatorname{range}(\nabla^2 G(\bar{\boldsymbol{\mu}})) = \mathcal{G}(T)$.

Since $(\boldsymbol{\mu}^\star - \bar{\boldsymbol{\mu}}) \in \mathcal{G}(T)$, multiplying Eq. (22) by $[\nabla^2 G(\bar{\boldsymbol{\mu}})]^+$, we obtain

$$\boldsymbol{\mu}^\star - \bar{\boldsymbol{\mu}} + \underbrace{[\nabla^2 G(\bar{\boldsymbol{\mu}})]^+ (\boldsymbol{\varepsilon}_G + \boldsymbol{\varepsilon}_R)}_{\boldsymbol{\varepsilon}} = -b [\nabla^2 G(\bar{\boldsymbol{\mu}})]^+ P \nabla R(\bar{\boldsymbol{\mu}})$$
$$= -b(\bar{a}/N) H_T(\bar{\boldsymbol{\mu}}) P \nabla R(\bar{\boldsymbol{\mu}})$$
$$= -b(\bar{a}/N) H_T(\bar{\boldsymbol{\mu}}) \partial C^*(\bar{\boldsymbol{\mu}}) \quad ,$$

where the last step follows, because $H_T(\bar{\boldsymbol{\mu}}) P = H_T(\bar{\boldsymbol{\mu}})$, and by Proposition A.9, $\partial C^*(\bar{\boldsymbol{\mu}}) = \nabla R(\bar{\boldsymbol{\mu}}) + \mathcal{G}^\perp(C)$, and $\mathcal{G}^\perp(C) \subseteq \mathcal{G}^\perp(T)$. The theorem now follows by noting that $\|\boldsymbol{\varepsilon}\| = O(b^2)$.

## C.4 Proof of Theorem 5.7

We prove a slightly stronger statement that holds not only for the bias measured under the Euclidean norm, but also when it is measured by the KL divergence. We use a more compact notation $\boldsymbol{\mu}^{\texttt{IND}}(b)$ and $\boldsymbol{\mu}^{\texttt{LMSR}}(b)$ for $\boldsymbol{\mu}^{\star}(b; \texttt{IND})$ and $\boldsymbol{\mu}^{\star}(b; \texttt{LMSR})$.

**Theorem C.1.** *For any $\bar{\boldsymbol{\mu}}$ there exist $\eta \in [1, 2]$ and $\eta_{\text{KL}} \in [1, 2]$ such that for all $b$*

$$\left\| \boldsymbol{\mu}^{\texttt{IND}}(b/\eta) - \bar{\boldsymbol{\mu}} \right\| = \left\| \boldsymbol{\mu}^{\texttt{LMSR}}(b) - \bar{\boldsymbol{\mu}} \right\| + O(b^2) \ , \tag{23}$$

$$\text{KL}\big( \boldsymbol{\mu}^{\texttt{IND}}(b/\eta_{\text{KL}}) \,\big\|\, \bar{\boldsymbol{\mu}} \big) = \text{KL}\big( \boldsymbol{\mu}^{\texttt{LMSR}}(b) \,\big\|\, \bar{\boldsymbol{\mu}} \big) + O(b^3) \ , \tag{24}$$

$$\text{KL}\big( \bar{\boldsymbol{\mu}} \,\big\|\, \boldsymbol{\mu}^{\texttt{IND}}(b/\eta_{\text{KL}}) \big) = \text{KL}\big( \bar{\boldsymbol{\mu}} \,\big\|\, \boldsymbol{\mu}^{\texttt{LMSR}}(b) \big) + O(b^3) \ . \tag{25}$$

*For these same $\eta$ and $\eta_{\text{KL}}$, we also have, for all $b$,*

$$\left\| \boldsymbol{\mu}^{\texttt{IND}}(b) - \bar{\boldsymbol{\mu}} \right\| = \eta \left\| \boldsymbol{\mu}^{\texttt{LMSR}}(b) - \bar{\boldsymbol{\mu}} \right\| \pm O(b^2) \ , \tag{26}$$

$$\text{KL}\big( \boldsymbol{\mu}^{\texttt{IND}}(b) \,\big\|\, \bar{\boldsymbol{\mu}} \big) = \eta_{\text{KL}}^2 \text{KL}\big( \boldsymbol{\mu}^{\texttt{LMSR}}(b) \,\big\|\, \bar{\boldsymbol{\mu}} \big) + O(b^3) \ , \tag{27}$$

$$\text{KL}\big( \bar{\boldsymbol{\mu}} \,\big\|\, \boldsymbol{\mu}^{\texttt{IND}}(b) \big) = \eta_{\text{KL}}^2 \text{KL}\big( \bar{\boldsymbol{\mu}} \,\big\|\, \boldsymbol{\mu}^{\texttt{LMSR}}(b) \big) + O(b^3) \ . \tag{28}$$

*Proof.* Without loss of generality, we assume that the coordinates of $\bar{\boldsymbol{\mu}}$ are sorted in the non-increasing order, i.e., $\bar{\mu}_1 \geq \bar{\mu}_2 \geq \cdots \geq \bar{\mu}_K$.

Our proof is based on Theorem 5.6, which states that

$$\boldsymbol{\mu}^{\star}(b; C) - \bar{\boldsymbol{\mu}} = b\left( -\frac{\bar{a}}{N} H_T(\bar{\boldsymbol{\mu}}) \partial C^*(\bar{\boldsymbol{\mu}}) \right) + \boldsymbol{\varepsilon}_b \ , \quad \text{where } \|\boldsymbol{\varepsilon}_b\| = O(b^2). \tag{29}$$

Let $H := H_T(\bar{\boldsymbol{\mu}}) = (\text{diag}_{k \in [K]} \bar{\mu}_k) - \bar{\boldsymbol{\mu}}\bar{\boldsymbol{\mu}}^{\intercal}$, and let $\boldsymbol{s}^{\texttt{LMSR}}$ and $\boldsymbol{s}^{\texttt{IND}}$ denote arbitrary elements of $\partial C^*(\bar{\boldsymbol{\mu}})$ for the costs LMSR and IND, respectively. Then Eq. (29) yields

$$\left\| \boldsymbol{\mu}^{\texttt{IND}}(b) - \bar{\boldsymbol{\mu}} \right\| = b\frac{\bar{a}}{N} \cdot \|H\boldsymbol{s}^{\texttt{IND}}\| + O(b^2) \ , \quad \left\| \boldsymbol{\mu}^{\texttt{LMSR}}(b) - \bar{\boldsymbol{\mu}} \right\| = b\frac{\bar{a}}{N} \cdot \|H\boldsymbol{s}^{\texttt{LMSR}}\| + O(b^2) \ ,$$

so Eqs. (23) and (26) follow by setting

$$\eta := \frac{\|H\boldsymbol{s}^{\texttt{IND}}\|}{\|H\boldsymbol{s}^{\texttt{LMSR}}\|} = \left( \frac{(\boldsymbol{s}^{\texttt{IND}})^{\intercal} H^2 \boldsymbol{s}^{\texttt{IND}}}{(\boldsymbol{s}^{\texttt{LMSR}})^{\intercal} H^2 \boldsymbol{s}^{\texttt{LMSR}}} \right)^{1/2} \tag{30}$$

and it remains to prove that $\eta \in [1, 2]$. (We do so below.)

For KL divergence results, we begin by using the fact that all entries of $\bar{\boldsymbol{\mu}}$ are positive so both $f_1(\boldsymbol{\mu}) := \text{KL}(\boldsymbol{\mu}\|\bar{\boldsymbol{\mu}})$ and $f_2(\boldsymbol{\mu}) := \text{KL}(\bar{\boldsymbol{\mu}}\|\boldsymbol{\mu})$ have bounded and continuous third derivatives in a sufficiently small neighborhood of $\bar{\boldsymbol{\mu}}$. Therefore, by Taylor's theorem, we obtain in this neighborhood

$$\text{KL}(\boldsymbol{\mu}\|\bar{\boldsymbol{\mu}}) = f_1(\boldsymbol{\mu}) = \underbrace{f_1(\bar{\boldsymbol{\mu}}) + \nabla f_1(\bar{\boldsymbol{\mu}})^{\intercal}(\boldsymbol{\mu} - \bar{\boldsymbol{\mu}})}_{=0} + (\boldsymbol{\mu} - \bar{\boldsymbol{\mu}})^{\intercal} \nabla^2 f_1(\bar{\boldsymbol{\mu}})^{\intercal}(\boldsymbol{\mu} - \bar{\boldsymbol{\mu}}) + O(\|\boldsymbol{\mu} - \bar{\boldsymbol{\mu}}\|^3)$$

$$\text{KL}(\bar{\boldsymbol{\mu}}\|\boldsymbol{\mu}) = f_2(\boldsymbol{\mu}) = \underbrace{f_2(\bar{\boldsymbol{\mu}}) + \nabla f_2(\bar{\boldsymbol{\mu}})^{\intercal}(\boldsymbol{\mu} - \bar{\boldsymbol{\mu}})}_{=0} + (\boldsymbol{\mu} - \bar{\boldsymbol{\mu}})^{\intercal} \nabla^2 f_2(\bar{\boldsymbol{\mu}})^{\intercal}(\boldsymbol{\mu} - \bar{\boldsymbol{\mu}}) + O(\|\boldsymbol{\mu} - \bar{\boldsymbol{\mu}}\|^3) \ .$$

By direct calculation, $\nabla^2 f_1(\bar{\boldsymbol{\mu}}) = \nabla^2 f_2(\bar{\boldsymbol{\mu}}) = \text{diag}_{k \in [K]}(\bar{\mu}_k)^{-1} =: M$. Now, by Theorem 5.1, we have $\|\boldsymbol{\mu}^{\star}(b; C) - \bar{\boldsymbol{\mu}}\| = O(b)$, and so we obtain

$$\text{KL}\big( \boldsymbol{\mu}^{\star}(b; C) \,\big\|\, \bar{\boldsymbol{\mu}} \big) = \big( \boldsymbol{\mu}^{\star}(b; C) - \bar{\boldsymbol{\mu}} \big)^{\intercal} M \big( \boldsymbol{\mu}^{\star}(b; C) - \bar{\boldsymbol{\mu}} \big) + O(b^3) \tag{31}$$

$$\text{KL}\big( \bar{\boldsymbol{\mu}} \,\big\|\, \boldsymbol{\mu}^{\star}(b; C) \big) = \big( \boldsymbol{\mu}^{\star}(b; C) - \bar{\boldsymbol{\mu}} \big)^{\intercal} M \big( \boldsymbol{\mu}^{\star}(b; C) - \bar{\boldsymbol{\mu}} \big) + O(b^3) \ . \tag{32}$$

We next invoke Eq. (29), but before we do so, note that since $H = (\text{diag}_{k \in [K]} \bar{\mu}_k) - \bar{\boldsymbol{\mu}}\bar{\boldsymbol{\mu}}^{\intercal}$ and $M = \text{diag}_{k \in [K]}(\bar{\mu}_k)^{-1}$, we have $HMH = H$. Now, invoking Eq. (29) and plugging it into Eq. (31), we obtain

$$\text{KL}\big( \boldsymbol{\mu}^{\texttt{IND}}(b) \,\big\|\, \bar{\boldsymbol{\mu}} \big) = b^2 \frac{\bar{a}^2}{N^2} \cdot (\boldsymbol{s}^{\texttt{IND}})^{\intercal} H\boldsymbol{s}^{\texttt{IND}} + O(b^3)$$

$$\text{KL}\big( \boldsymbol{\mu}^{\texttt{LMSR}}(b) \,\big\|\, \bar{\boldsymbol{\mu}} \big) = b^2 \frac{\bar{a}^2}{N^2} \cdot (\boldsymbol{s}^{\texttt{LMSR}})^{\intercal} H\boldsymbol{s}^{\texttt{LMSR}} + O(b^3)$$

and similarly for KL $(\bar{\boldsymbol{\mu}}\|\cdot)$. Therefore Eqs. (24), (25), (27) and (28) follow by setting

$$
\eta_{\text{KL}} := \left( \frac{(\boldsymbol{s}^{\text{IND}})^{\intercal} H \boldsymbol{s}^{\text{IND}}}{(\boldsymbol{s}^{\text{LMSR}})^{\intercal} H \boldsymbol{s}^{\text{LMSR}}} \right)^{1/2} \tag{33}
$$

and it remains to prove that $\eta_{\text{KL}} \in [1, 2]$.

In the remainder of the proof, we show that $\eta$ and $\eta_{\text{KL}}$ defined in Eqs. (30) and (33) are in $[1, 2]$. We proceed by Lemma C.2 (see below), which shows that $1 \leq (\boldsymbol{v}^{\intercal} H \boldsymbol{v})/(\boldsymbol{s}^{\intercal} H \boldsymbol{s}) \leq 4$ and $1 \leq (\boldsymbol{v} H^2 \boldsymbol{v})/(\boldsymbol{s} H^2 \boldsymbol{s}) \leq 4$ for any sorted vectors $\boldsymbol{s}$ and $\boldsymbol{v}$, whose differences between consecutive coordinates are within a factor-of-two of each other. We only need to show that $\boldsymbol{s} = \boldsymbol{s}^{\text{LMSR}}$ and $\boldsymbol{v} = \boldsymbol{s}^{\text{IND}}$ satisfy this condition.

Recall that $\boldsymbol{s}^{\text{LMSR}}$ and $\boldsymbol{s}^{\text{IND}}$ can be chosen as arbitrary elements of $\partial C^*(\bar{\boldsymbol{\mu}})$ for the costs LMSR and IND. From the properties of conjugates, $\boldsymbol{s} \in \partial C^*(\bar{\boldsymbol{\mu}})$ iff $\nabla C(\boldsymbol{s}) = \bar{\boldsymbol{\mu}}$, so we can obtain $\boldsymbol{s}^{\text{LMSR}}$ and $\boldsymbol{s}^{\text{IND}}$ by inverting the gradients of LMSR and IND:

$$
s_k^{\text{LMSR}} = \log \bar{\mu}_k \ , \quad s_k^{\text{IND}} = \log \left( \frac{\bar{\mu}_k}{1 - \bar{\mu}_k} \right) \quad \text{for all } k \in [K].
$$

Note that both $s_k^{\text{LMSR}}$ and $s_k^{\text{IND}}$ are monotone transformations of $\bar{\mu}_k$, and since $\bar{\boldsymbol{\mu}}$ is sorted, so must be $\boldsymbol{s}^{\text{LMSR}}$ and $\boldsymbol{s}^{\text{IND}}$. We next show that the differences between the consecutive coordinates of $\boldsymbol{s}^{\text{LMSR}}$ and $\boldsymbol{s}^{\text{IND}}$ are within a factor two of each other. For any $k \in [K - 1]$, we have

$$
s_k^{\text{LMSR}} - s_{k+1}^{\text{LMSR}} = \log \left( \frac{\bar{\mu}_k}{\bar{\mu}_{k+1}} \right)
$$

and we also have

$$
\begin{aligned}
s_k^{\text{IND}} - s_{k+1}^{\text{IND}} &= \log \left( \frac{\bar{\mu}_k}{\bar{\mu}_{k+1}} \cdot \frac{1 - \bar{\mu}_{k+1}}{1 - \bar{\mu}_k} \right) \\
&= \log \left( \frac{\bar{\mu}_k}{\bar{\mu}_{k+1}} \cdot \frac{c_k + \bar{\mu}_k}{c_k + \bar{\mu}_{k+1}} \right) = \log \left( \frac{\bar{\mu}_k}{\bar{\mu}_{k+1}} \right) + \log \left( \frac{c_k + \bar{\mu}_k}{c_k + \bar{\mu}_{k+1}} \right) \ ,
\end{aligned}
$$

where $c_k := 1 - \bar{\mu}_k - \bar{\mu}_{k+1} \geq 0$. Since $\bar{\mu}_k \geq \bar{\mu}_{k+1}$, we therefore have

$$
0 \leq \log \left( \frac{c_k + \bar{\mu}_k}{c_k + \bar{\mu}_{k+1}} \right) \leq \log \left( \frac{\bar{\mu}_k}{\bar{\mu}_{k+1}} \right)
$$

and therefore

$$
s_k^{\text{LMSR}} - s_{k+1}^{\text{LMSR}} \leq s_k^{\text{IND}} - s_{k+1}^{\text{IND}} \leq 2 (s_k^{\text{LMSR}} - s_{k+1}^{\text{LMSR}}) \ .
$$

Thus, Lemma C.2 with $\boldsymbol{s} = \boldsymbol{s}^{\text{LMSR}}$ and $\boldsymbol{v} = \boldsymbol{s}^{\text{IND}}$ and $H = H_C(\bar{\boldsymbol{\mu}})$ proves that indeed $\eta \in [1, 2]$ and $\eta_{\text{KL}} \in [1, 2]$. □

**Lemma C.2.** *Let $\boldsymbol{\mu} \in \mathbb{R}^K$ be a sorted probability vector, i.e., $\mu_1 \geq \mu_2 \geq \cdots \geq \mu_K$, and let $H = (\text{diag}_{k \in [K]} \mu_k) - \boldsymbol{\mu}\boldsymbol{\mu}^{\intercal}$ be the covariance matrix of the associated multinomial distribution. Let $\boldsymbol{s}, \boldsymbol{v}$ be sorted vectors in $\mathbb{R}^K$, i.e., $s_1 \geq \cdots \geq s_K$ and $v_1 \geq \cdots \geq v_K$, such that $s_k - s_{k+1} \leq v_k - v_{k+1} \leq 2(s_k - s_{k+1})$. Then the following two statements hold*

$$
\boldsymbol{s}^{\intercal} H \boldsymbol{s} \leq \boldsymbol{v}^{\intercal} H \boldsymbol{v} \leq 4 \cdot \boldsymbol{s}^{\intercal} H \boldsymbol{s} \ , \quad \boldsymbol{s}^{\intercal} H^2 \boldsymbol{s} \leq \boldsymbol{v}^{\intercal} H^2 \boldsymbol{v} \leq 4 \cdot \boldsymbol{s}^{\intercal} H^2 \boldsymbol{s} \ .
$$

*Proof.* The proof proceeds in several steps.

**Step 1: Decomposition of $\boldsymbol{s}$ and $\boldsymbol{v}$ into an alternate basis.** We begin by rewriting $\boldsymbol{s}$ and $\boldsymbol{v}$ in the basis consisting of vectors $\boldsymbol{z}_k \in \mathbb{R}^K$, for $k = 1, \dots, K$, where each $\boldsymbol{z}_k$ has ones on positions 1 through $k$, and zeros on the remaining positions, i.e., $z_{kj} = \mathbf{1}\{j \leq k\}$. Let

$$
a_k := s_k - s_{k+1} \quad \text{and} \quad b_k := v_k - v_{k+1} \quad \text{for } k = 1, \dots, K - 1.
$$

The vectors $\boldsymbol{s}$ and $\boldsymbol{v}$ can then be written as

$$
\boldsymbol{s} = s_K \boldsymbol{z}_K + \sum_{k=1}^{K-1} a_k \boldsymbol{z}_k \ , \quad \boldsymbol{v} = v_K \boldsymbol{z}_K + \sum_{k=1}^{K-1} b_k \boldsymbol{z}_k \tag{34}
$$

where

$$0 \le a_k \le b_k \le 2a_k \ . \tag{35}$$

From the definition of $H$, we have $H\boldsymbol{z}_K = 0$ and so

$$\boldsymbol{s}^\mathsf{T} H\boldsymbol{s} = (\boldsymbol{s}')^\mathsf{T} H\boldsymbol{s}' \ , \quad \boldsymbol{v}^\mathsf{T} H\boldsymbol{v} = (\boldsymbol{v}')^\mathsf{T} H\boldsymbol{v}' \ , \quad \boldsymbol{s}^\mathsf{T} H^2 \boldsymbol{s} = (\boldsymbol{s}')^\mathsf{T} H^2 \boldsymbol{s}' \ , \quad \boldsymbol{v}^\mathsf{T} H^2 \boldsymbol{v} = (\boldsymbol{v}')^\mathsf{T} H^2 \boldsymbol{v}' \ , \tag{36}$$

where $\boldsymbol{s}'$ and $\boldsymbol{v}'$ exclude the basis element $\boldsymbol{z}_K$, i.e.,

$$\boldsymbol{s}' = \sum_{k=1}^{K-1} a_k \boldsymbol{z}_k \ , \quad \boldsymbol{v}' = \sum_{k=1}^{K-1} b_k \boldsymbol{z}_k \ . \tag{37}$$

Therefore, we have the decomposition

$$\boldsymbol{s}^\mathsf{T} H\boldsymbol{s} = (\boldsymbol{s}')^\mathsf{T} H\boldsymbol{s}' = \left( \sum_{k=1}^{K-1} a_k \boldsymbol{z}_k \right)^\mathsf{T} H \left( \sum_{\ell=1}^{K-1} a_\ell \boldsymbol{z}_\ell \right) = \sum_{k,\ell \in [K-1]} a_k a_\ell (\boldsymbol{z}_k^\mathsf{T} H \boldsymbol{z}_\ell) \ , \tag{38}$$

and similar decompositions for $\boldsymbol{v}^\mathsf{T} H\boldsymbol{v}$, $\boldsymbol{s}^\mathsf{T} H^2 \boldsymbol{s}$, and $\boldsymbol{v}^\mathsf{T} H^2 \boldsymbol{v}$.

In the remainder of the proof we show that for all $k, \ell \in [K-1]$, we have $\boldsymbol{z}_k^\mathsf{T} H \boldsymbol{z}_\ell \ge 0$ and $\boldsymbol{z}_k^\mathsf{T} H^2 \boldsymbol{z}_\ell \ge 0$, which together with the decomposition in Eq. (38) and similar decompositions implied by Eq. (36) will imply the theorem, thanks to the fact that $0 \le a_k \le b_k \le 2a_k$.

**Step 2: $\boldsymbol{z}_k^\mathsf{T} H \boldsymbol{z}_\ell \ge 0$.** Fix $k, \ell \in [K]$ and assume $k \le \ell$. Then

$$\begin{aligned}
\boldsymbol{z}_k^\mathsf{T} H \boldsymbol{z}_\ell &= \sum_{j \in [K]} \mu_j z_{kj} z_{\ell j} - (\boldsymbol{z}_k^\mathsf{T} \boldsymbol{\mu})(\boldsymbol{\mu}^\mathsf{T} \boldsymbol{z}_\ell) \\
&= \underbrace{\left( \sum_{j \le k} \mu_j \right)}_{\mu_k^{[1]}} - \underbrace{\left( \sum_{j \le k} \mu_j \right)}_{\mu_k^{[1]}} \underbrace{\left( \sum_{j \le \ell} \mu_j \right)}_{\mu_\ell^{[1]}} \\
&= \mu_k^{[1]} \left( 1 - \mu_\ell^{[1]} \right) \ge 0 \ . 
\end{aligned} \tag{39}$$

Above, we introduced notation for partial sums

$$\mu_k^{[d]} := \sum_{j=1}^{k} \mu_j^d \ ,$$

and used the fact that $\mu_\ell^{[1]} \le 1$, because entries of $\boldsymbol{\mu}$ are non-negative and sum to one. This proves that $\boldsymbol{z}_k^\mathsf{T} H \boldsymbol{z}_\ell \ge 0$ when $k \le \ell$. The case $k \ge \ell$ follows by symmetry of $H$.

**Step 3: $\boldsymbol{z}_k^\mathsf{T} H^2 \boldsymbol{z}_\ell \ge 0$.** Again, let $k, \ell \in [K]$ and $k \le \ell$. First note that

$$H^2 = (\operatorname{diag}_{j \in [K]} \mu_j^2) - \sum_{j \in [K]} \mu_j^2 \boldsymbol{e}_j \boldsymbol{\mu}^\mathsf{T} - \sum_{j \in [K]} \mu_j^2 \boldsymbol{\mu} \boldsymbol{e}_j^\mathsf{T} + \mu_K^{[2]} \boldsymbol{\mu} \boldsymbol{\mu}^\mathsf{T}$$

where $\boldsymbol{e}_j$ is the $j$th vector of the standard basis and we used our partial sum notation to substitute $\mu_K^{[2]}$ for $\|\boldsymbol{\mu}\|^2$. Thus, we can write

$$\begin{aligned}
\boldsymbol{z}_k^\mathsf{T} H^2 \boldsymbol{z}_\ell &= \mu_k^{[2]} - \mu_k^{[2]} \mu_\ell^{[1]} - \mu_k^{[1]} \mu_\ell^{[2]} + \mu_K^{[2]} \mu_k^{[1]} \mu_\ell^{[1]} \\
&\ge \mu_k^{[2]} - \mu_k^{[2]} \mu_\ell^{[1]} - \mu_k^{[1]} \mu_\ell^{[2]} + \mu_\ell^{[2]} \mu_k^{[1]} \mu_\ell^{[1]} \tag{40} \\
&= \left( \mu_k^{[2]} - \mu_\ell^{[2]} \mu_k^{[1]} \right) \left( 1 - \mu_\ell^{[1]} \right) \\
&\ge \left( \mu_k^{[2]} \mu_\ell^{[1]} - \mu_\ell^{[2]} \mu_k^{[1]} \right) \left( 1 - \mu_\ell^{[1]} \right) \tag{41} \\
&= \mu_k^{[1]} \mu_\ell^{[1]} \left( \frac{\mu_k^{[2]}}{\mu_k^{[1]}} - \frac{\mu_\ell^{[2]}}{\mu_\ell^{[1]}} \right) \left( 1 - \mu_\ell^{[1]} \right) \ge 0 \ . \tag{42}
\end{aligned}$$

In Eq. (40), we used the bound $\mu_K^{[2]} \geq \mu_\ell^{[2]}$. In Eq. (41), we used that $0 \leq \mu_\ell^{[1]} \leq 1$, so $\mu_k^{[2]}(1-\mu_\ell^{[1]}) \geq \mu_k^{[2]}\mu_\ell^{[1]}(1-\mu_\ell^{[1]})$. The final inequality uses the fact that $\mu_\ell^{[1]} \leq 1$ and $\mu_k^{[2]}/\mu_k^{[1]} \geq \mu_\ell^{[2]}/\mu_\ell^{[1]}$. The latter clearly holds if $\ell = k$ or $\mu_\ell^{[1]} = \mu_k^{[1]}$ (in which case $\mu_j = 0$ for $k < j \leq \ell$ and thus $\mu_\ell^{[2]} = \mu_k^{[2]}$). We now argue that $\mu_k^{[2]}/\mu_k^{[1]} \geq \mu_\ell^{[2]}/\mu_\ell^{[1]}$ also holds when $k < \ell$ and $\mu_{k+1} > 0$. We introduce the interval sum notation

$$\mu_{k+1:\ell}^{[d]} := \sum_{j=k+1}^{\ell} \mu_j^d = \mu_\ell^{[d]} - \mu_k^{[d]} \ .$$

We begin by writing $\mu_\ell^{[2]}/\mu_\ell^{[1]}$ as the following convex combination

$$\frac{\mu_\ell^{[2]}}{\mu_\ell^{[1]}} = \frac{\mu_k^{[2]} + \mu_{k+1:\ell}^{[2]}}{\mu_\ell^{[1]}} = \frac{\mu_k^{[1]}}{\mu_\ell^{[1]}} \cdot \frac{\mu_k^{[2]}}{\mu_k^{[1]}} + \frac{\mu_{k+1:\ell}^{[1]}}{\mu_\ell^{[1]}} \cdot \frac{\mu_{k+1:\ell}^{[2]}}{\mu_{k+1:\ell}^{[1]}}$$

$$= \lambda \cdot \frac{\mu_k^{[2]}}{\mu_k^{[1]}} + (1-\lambda) \cdot \frac{\mu_{k+1:\ell}^{[2]}}{\mu_{k+1:\ell}^{[1]}} \ , \tag{43}$$

where we write $\lambda := \mu_k^{[1]}/\mu_\ell^{[1]}$. The expressions weighted in Eq. (43) by $\lambda$ and $1 - \lambda$ can be viewed as weighted averages of $\mu_j$, with the weights also equal to $\mu_j$. Since $\boldsymbol{\mu}$ is sorted, we have

$$\frac{\mu_k^{[2]}}{\mu_k^{[1]}} = \sum_{j=1}^{k} \frac{\mu_j}{\mu_k^{[1]}} \cdot \mu_j \geq \mu_k \qquad \text{and} \qquad \frac{\mu_{k+1:\ell}^{[2]}}{\mu_{k+1:\ell}^{[1]}} = \sum_{j=k+1}^{\ell} \frac{\mu_j}{\mu_{k+1:\ell}^{[1]}} \cdot \mu_j \leq \mu_{k+1} \ ,$$

so

$$\frac{\mu_k^{[2]}}{\mu_k^{[1]}} \geq \mu_k \geq \mu_{k+1} \geq \frac{\mu_{k+1:\ell}^{[2]}}{\mu_{k+1:\ell}^{[1]}} \ .$$

Plugging this back into Eq. (43), we obtain

$$\frac{\mu_\ell^{[2]}}{\mu_\ell^{[1]}} = \lambda \cdot \frac{\mu_k^{[2]}}{\mu_k^{[1]}} + (1-\lambda) \cdot \frac{\mu_{k+1:\ell}^{[2]}}{\mu_{k+1:\ell}^{[1]}} \leq \frac{\mu_k^{[2]}}{\mu_k^{[1]}} \ ,$$

finishing the proof of Eq. (42), showing that $\boldsymbol{z}_k^\intercal H^2 \boldsymbol{z}_\ell \geq 0$ when $k \leq \ell$. The case $k \geq \ell$ again follows by symmetry of $H^2$.

**Step 4: Putting it all together.** Let $M$ be either the matrix $H$ or $H^2$. Since in both cases $\boldsymbol{z}_k^\intercal M \boldsymbol{z}_\ell \geq 0$, the inequalities $0 \leq a_k \leq b_k \leq 2a_k$ imply that

$$\sum_{k,\ell \in [K-1]} a_k a_\ell (\boldsymbol{z}_k^\intercal M \boldsymbol{z}_\ell) \leq \sum_{k,\ell \in [K-1]} b_k b_\ell (\boldsymbol{z}_k^\intercal M \boldsymbol{z}_\ell) \leq 4 \cdot \sum_{k,\ell \in [K-1]} a_k a_\ell (\boldsymbol{z}_k^\intercal M \boldsymbol{z}_\ell) \ .$$

This is by the decomposition in Eq. (38) and analogous decompositions for $\boldsymbol{v}^\intercal H \boldsymbol{v}$, $\boldsymbol{s}^\intercal H^2 \boldsymbol{s}$, and $\boldsymbol{v}^\intercal H^2 \boldsymbol{v}$ equivalent to

$$\boldsymbol{s}^\intercal M \boldsymbol{s} \leq \boldsymbol{v}^\intercal M \boldsymbol{v} \leq 4 \cdot \boldsymbol{s}^\intercal M \boldsymbol{s} \ ,$$

proving the lemma. $\qquad \square$

## D  Proofs and Additional Results for Section 6

### D.1  Trader Dynamics

To study convergence properties of the market, we need to posit a model of how the traders arrive in the market and which securities they buy or sell, as a function of their current holdings of securities $\boldsymbol{r}_i$ and cash $c_i$, and the current market state $\boldsymbol{s}$. We refer to such a model as *trader dynamics*. We consider two simple trader dynamics:

- *All-securities dynamics* (ASD). In each round, a trader $i \in [N]$ is chosen uniformly at random. This trader then buys a bundle $\boldsymbol{\delta} \in \mathbb{R}^K$ which optimizes her utility, i.e., if the current state of the market is $\boldsymbol{s}$ and the current cash and security allocations of the trader are $c_i$ and $\boldsymbol{r}_i$, then the trader picks $\boldsymbol{\delta}$ maximizing $U_i(\boldsymbol{r}_i + \boldsymbol{\delta}, c_i - C_b(\boldsymbol{s} + \boldsymbol{\delta}) + C_b(\boldsymbol{s}))$.

- *Single-security dynamics* (SSD). In each round, a trader $i \in [N]$ is chosen uniformly at random and this trader picks a security $k \in [K]$ uniformly at random. The trader then buys a quantity $\delta \in \mathbb{R}$ of the $k$th security to optimize her utility. Let $\boldsymbol{e}_k$ be the $k$th vector of the standard basis. Then the trader picks $\delta$ maximizing $U_i(\boldsymbol{r}_i + \delta \boldsymbol{e}_k, \ c_i - C_b(\boldsymbol{s} + \delta \boldsymbol{e}_k) + C_b(\boldsymbol{s}))$.

The all-securities dynamics has been recently studied by Frongillo and Reid [9]. This model assumes that traders are able to calculate the optimal bundle over all securities, which may not be a realistic assumption when the number of securities is large. The single-security dynamics, in which we only assume that traders can optimize over a single security at a time, is more appropriate for computationally limited traders.

We formalize both dynamics in a unified analysis via *blocks*. Specifically, we assume that the coordinates $[N] \times [K]$ are partitioned into blocks $\alpha \in \mathcal{A}$, where $\alpha \subseteq [N] \times [K]$, such that $\biguplus_{\alpha \in \mathcal{A}} \alpha = [N] \times [K]$. Blocks are disjoint subsets of coordinates of the overall allocation vector $\boldsymbol{r} \in \mathbb{R}^{NK}$. We further assume that each block $\alpha \in \mathcal{A}$ is fully contained within coordinates corresponding to some trader $i$. Thus each block $\alpha$ can be written as $\{i\} \times \beta$ for some $i \in [N]$ and $\beta \subseteq [K]$. For ASD, we have $N$ blocks $\mathcal{A} = \big\{\{i\} \times [K] : i \in [N]\big\}$. For SSD, we have $NK$ blocks $\mathcal{A} = \big\{\{i\} \times \{k\} : i \in [N], k \in [K]\big\}$.

Let $E_\alpha \in \mathbb{R}^{NK \times |\alpha|}$ be the *embedding matrix* for the block $\alpha$. It maps $|\alpha|$-dimensional vectors $\boldsymbol{u} \in \mathbb{R}^{|\alpha|}$ to vectors $\boldsymbol{v} \in \mathbb{R}^{NK}$ that are zero everywhere except for the block $\alpha$, where they coincide with $\boldsymbol{u}$. The range of $E_\alpha$ is exactly the set of vectors that are zero outside the block $\alpha$. Its transpose $E_\alpha^\mathsf{T}$ projects vectors from $\mathbb{R}^{NK}$ to $\mathbb{R}^{|\alpha|}$ by removing all coordinates outside the block $\alpha$. For any subset $\beta \subseteq [K]$, we similarly define the embedding matrices $E_\beta \in \mathbb{R}^{K \times |\beta|}$.

The next theorem shows that optimizing utility, as in ASD or SSD, corresponds to the greedy optimization of the potential function $F$ along the coordinates of the corresponding block:

**Theorem D.1.** *Let $\alpha = \{i\} \times \beta$, where $\beta \subseteq [K]$, be a block of coordinates controlled by trader $i$. Assume that trader $i$ has security bundle $\boldsymbol{r}_i$ and $c_i$ units of cash, and the current market state is $\boldsymbol{s}$. Then*

$$\operatorname*{argmax}_{\boldsymbol{\delta} \in \mathrm{range}(E_\beta)} U_i(\boldsymbol{r}_i + \boldsymbol{\delta}, \ c_i - C_b(\boldsymbol{s} + \boldsymbol{\delta}) + C_b(\boldsymbol{s})) = \operatorname*{argmin}_{\boldsymbol{\delta} \in \mathrm{range}(E_\beta)} F(\boldsymbol{r}_{-i}, \ \boldsymbol{r}_i + \boldsymbol{\delta}) \ ,$$

*where $\boldsymbol{r}_{-i}$ denotes the concatenation of $\boldsymbol{r}_j$ across $j \neq i$.*

*Proof.* The proof is immediate from the definition of the potential $F$. $\qquad\square$

### D.2 Relationship between the Suboptimality of Potential and the Convergence Error

In this appendix, we relate the suboptimality of the objective to several other quantities used in analysis of convergence error. We begin by defining these quantities.

Given an allocation vector $\boldsymbol{r} \in R^{NK}$, the associated market price (the gradient of the cost) will be denoted $\boldsymbol{\mu}_0(\boldsymbol{r})$ and the gradients of trader potentials will be denoted $\boldsymbol{\mu}_i(\boldsymbol{r})$:

$$\begin{aligned}
\boldsymbol{\mu}_0(\boldsymbol{r}) &:= \nabla C_b\big(\textstyle\sum_{i=1}^N \boldsymbol{r}_i\big) = \nabla C\big(\textstyle\sum_{i=1}^N \boldsymbol{r}_i / b\big) \\
\boldsymbol{\mu}_i(\boldsymbol{r}) &:= \nabla F_i(-\boldsymbol{r}_i) = \nabla T(\tilde{\boldsymbol{\theta}}_i - a_i \boldsymbol{r}_i) \qquad \text{for } i \in [N]
\end{aligned}$$

where $T$ is the log partition function. As in the body of the paper, let $\boldsymbol{r}^\star$ denote an arbitrary minimizer of $F$ and let $F^\star$ denote the minimum value of $F$. From Theorem B.2, at any equilibrium allocation $\boldsymbol{r}^\star$,

$$\boldsymbol{\mu}_0(\boldsymbol{r}^\star) = \boldsymbol{\mu}^\star \qquad \text{and} \qquad \boldsymbol{\mu}_i(\boldsymbol{r}^\star) = \boldsymbol{\mu}^\star \text{ for all } i \in [N].$$

Let $\boldsymbol{r}^t$ denote the allocation vector after the $t$th trade, and $\boldsymbol{\mu}^t := \boldsymbol{\mu}_0(\boldsymbol{r}^t)$ be the corresponding market price. We next show that to bound the convergence error $\|\boldsymbol{\mu}_0(\boldsymbol{r}^t) - \boldsymbol{\mu}^\star\|$ it suffices to bound the suboptimality of the current objective value, $F(\boldsymbol{r}^t) - F^\star$. In fact, we show that the suboptimality $F(\boldsymbol{r}^t) - F^\star$ simultaneously also bounds $\|\boldsymbol{\mu}_i(\boldsymbol{r}^t) - \boldsymbol{\mu}^\star\|$, which can be viewed as a measure of suboptimality of individual traders and will be used in our later analysis. We first prove this result when $C$ has a Lipschitz-continuous gradient and then for the case when $C$ is convex$^+$.

**Theorem D.2.** *If $\nabla C$ has the Lipschitz constant $L_C$ then for any $\boldsymbol{r} \in \mathbb{R}^{NK}$*

$$F(\boldsymbol{r}) - F^\star \geq \frac{b}{2L_C} \cdot \|\boldsymbol{\mu}_0(\boldsymbol{r}) - \boldsymbol{\mu}^\star\|^2 + \frac{1}{2L_T} \cdot \sum_{i=1}^N \frac{1}{a_i} \|\boldsymbol{\mu}_i(\boldsymbol{r}) - \boldsymbol{\mu}^\star\|^2 \ ,$$

*where $L_T$ is the Lipschitz constant of $\nabla T$.*

*Proof.* First note that since $\nabla C$ and $\nabla T$ have Lipschitz constants $L_C$ and $L_T$, their conjugates are strongly convex with constants $1/L_C$ and $1/L_T$. Further, by the properties of conjugates (see Theorems 12.3 and 16.1 of Rockafellar [15]), the definitions of $F_i$ and $C_b$ yield

$$F_i^*(\boldsymbol{\mu}) = \frac{1}{a_i} \big( T^*(\boldsymbol{\mu}) - \tilde{\boldsymbol{\theta}}_i \cdot \boldsymbol{\mu} \big) \ , \quad C_b^*(\boldsymbol{\mu}) = bC^*(\boldsymbol{\mu}) \ ,$$

and so $F_i^*$ and $C_b^*$ are strongly convex, respectively, with constants $1/(a_i L_T)$ and $b/L_C$.

We now invoke the duality result of Theorem B.2 to prove our theorem. Specifically, from Eqs. (18) and (19), we have

$$F(\boldsymbol{r}^\star) = -\sum_{i=1}^N F_i^*(\boldsymbol{\mu}^\star) - C_b^*(\boldsymbol{\mu}^\star) \ .$$

Therefore, for any $\boldsymbol{r}$, we have

$$F(\boldsymbol{r}) - F^\star = \sum_{i=1}^N F_i(-\boldsymbol{r}_i) + C_b\big(\sum_i \boldsymbol{r}_i\big) + \sum_{i=1}^N F_i^*(\boldsymbol{\mu}^\star) + C_b^*(\boldsymbol{\mu}^\star)$$

$$= \sum_{i=1}^N \Big[ F_i(-\boldsymbol{r}_i) + F_i^*(\boldsymbol{\mu}^\star) + \boldsymbol{r}_i^\mathsf{T} \boldsymbol{\mu}^\star \Big] + \Big[ C_b\big(\sum_i \boldsymbol{r}_i\big) + C_b^*(\boldsymbol{\mu}^\star) - \big(\sum_i \boldsymbol{r}_i\big)^\mathsf{T} \boldsymbol{\mu}^\star \Big].$$

$$(44)$$

Using conjugacy and strong convexity, we next show that the terms in the brackets can be lower-bounded by quadratic functions.

Let $\boldsymbol{s} := \sum_i \boldsymbol{r}_i$. Since $\boldsymbol{\mu}_i(\boldsymbol{r}) = \nabla F_i(-\boldsymbol{r}_i)$ and $\boldsymbol{\mu}_0(\boldsymbol{r}) = \nabla C_b(\boldsymbol{s})$,

$$(-\boldsymbol{r}_i) \in \partial F_i^*(\boldsymbol{\mu}_i(\boldsymbol{r})) \quad \text{and} \quad F_i(-\boldsymbol{r}_i) = -\boldsymbol{r}_i^\mathsf{T} \boldsymbol{\mu}_i(\boldsymbol{r}) - F_i^*(\boldsymbol{\mu}_i(\boldsymbol{r}))$$
$$\boldsymbol{s} \in \partial C_b^*(\boldsymbol{\mu}_0(\boldsymbol{r})) \quad \text{and} \quad C_b(\boldsymbol{\mu}_0(\boldsymbol{r})) = \boldsymbol{s}^\mathsf{T} \boldsymbol{\mu}_0(\boldsymbol{r}) - C_b^*(\boldsymbol{\mu}_0(\boldsymbol{r})) \ .$$

Using these identities and invoking the strong convexity of $F_i$ and $C_b$, we thus obtain

$$F_i(-\boldsymbol{r}_i) + F_i^*(\boldsymbol{\mu}^\star) + \boldsymbol{r}_i^\mathsf{T} \boldsymbol{\mu}^\star = F_i^*(\boldsymbol{\mu}^\star) - (-\boldsymbol{r}_i)^\mathsf{T} \big( \boldsymbol{\mu}^\star - \boldsymbol{\mu}_i(\boldsymbol{r}) \big) - F_i^*(\boldsymbol{\mu}_i(\boldsymbol{r}))$$

$$\geq \frac{1}{2a_i L_T} \|\boldsymbol{\mu}_i(\boldsymbol{r}^t) - \boldsymbol{\mu}^\star\|^2$$

$$C_b(\boldsymbol{s}) + C_b^*(\boldsymbol{\mu}^\star) - \boldsymbol{s}^\mathsf{T} \boldsymbol{\mu}^\star = C_b^*(\boldsymbol{\mu}^\star) - \boldsymbol{s}^\mathsf{T} \big( \boldsymbol{\mu}^\star - \boldsymbol{\mu}_0(\boldsymbol{r}) \big) - C_b^*(\boldsymbol{\mu}_0(\boldsymbol{r}))$$

$$\geq \frac{b}{2L_C} \|\boldsymbol{\mu}_0(\boldsymbol{r}) - \boldsymbol{\mu}^\star\|^2 \ .$$

The theorem now follows by applying these lower bounds in Eq. (44). $\qquad \square$

**Theorem D.3.** *If $C$ is convex$^+$ then there exist $\varepsilon > 0$ and $c > 0$ such that if $F(\boldsymbol{r}) - F^\star \leq \varepsilon$ then*

$$F(\boldsymbol{r}) - F^\star \geq c \left[ b\|\boldsymbol{\mu}_0(\boldsymbol{r}) - \boldsymbol{\mu}^\star\|^2 + \sum_{i=1}^N \frac{1}{a_i} \|\boldsymbol{\mu}_i(\boldsymbol{r}) - \boldsymbol{\mu}^\star\|^2 \right] \ .$$

*Proof.* The proof begins similarly to proof of Theorem D.2, by establishing the identity

$$F(\boldsymbol{r}) - F^\star = \sum_{i=1}^N \Big[ F_i^*(\boldsymbol{\mu}^\star) - (-\boldsymbol{r}_i)^\mathsf{T} \big( \boldsymbol{\mu}^\star - \boldsymbol{\mu}_i(\boldsymbol{r}) \big) - F_i^*(\boldsymbol{\mu}_i(\boldsymbol{r})) \Big]$$

$$+ \Big[ C_b^*(\boldsymbol{\mu}^\star) - \boldsymbol{s}^\mathsf{T} \big( \boldsymbol{\mu}^\star - \boldsymbol{\mu}_0(\boldsymbol{r}) \big) - C_b^*(\boldsymbol{\mu}_0(\boldsymbol{r})) \Big] \ . \tag{45}$$

Let $G_i$ be the convex$^+$ functions that match $F_i^*$ on $\operatorname{dom} F_i^*$ and $R$ be the convex$^+$ function that matches $C^*$ on $\operatorname{dom} C^*$, and that have the additional properties outlined in Proposition A.9. Thus,

$$F(\boldsymbol{r}) - F^\star = \sum_{i=1}^{N}\Big[G_i(\boldsymbol{\mu}^\star) - [\nabla G_i(\boldsymbol{\mu}_i(\boldsymbol{r}))]^\mathsf{T}\big(\boldsymbol{\mu}^\star - \boldsymbol{\mu}_i(\boldsymbol{r})\big) - G_i(\boldsymbol{\mu}_i(\boldsymbol{r}))\Big]$$

$$+ b\Big[R(\boldsymbol{\mu}^\star) - [\nabla R(\boldsymbol{\mu}_0(\boldsymbol{r}))]^\mathsf{T}\big(\boldsymbol{\mu}^\star - \boldsymbol{\mu}_0(\boldsymbol{r})\big) - R(\boldsymbol{\mu}_0(\boldsymbol{r}))\Big] \ . \qquad (46)$$

By convexity$^+$, functions $G_i$ are strictly convex on $\mathcal{M} = \operatorname{dom} F_i^*$ and $R$ is also strictly convex on $\mathcal{M} \subseteq \operatorname{dom} C^*$. Thus, $G_i$ and $R$ are strictly convex in a neighborhood of $\boldsymbol{\mu}^\star$ (in aff $\mathcal{M}$), which implies that the terms on the right-hand side of Eq. (46) are strictly convex in $\boldsymbol{\mu}_i(\boldsymbol{r})$ and $\boldsymbol{\mu}_0(\boldsymbol{r})$. Since they are minimized at $\boldsymbol{\mu}^\star$, we obtain $\boldsymbol{\mu}_i(\boldsymbol{r}) \to \boldsymbol{\mu}^\star$ and $\boldsymbol{\mu}_0(\boldsymbol{r}) \to \boldsymbol{\mu}^\star$ as $F(\boldsymbol{r}) - F^\star \to 0$. Thus, by picking a sufficiently small $\varepsilon$, we can guarantee that $\boldsymbol{\mu}_i(\boldsymbol{r})$ and $\boldsymbol{\mu}_0(\boldsymbol{r})$ are arbitrarily close to $\boldsymbol{\mu}^\star$ whenever $F(\boldsymbol{r}) - F^\star \le \varepsilon$. From Taylor's theorem, and the fact that $\nabla^2 G_i \equiv H_{F_i}^+ \equiv (1/a_i)H_T^+$ and $\nabla^2 R \equiv H_C^+$, we obtain

$$G_i(\boldsymbol{\mu}^\star) - [\nabla G_i(\boldsymbol{\mu}_i(\boldsymbol{r}))]^\mathsf{T}\big(\boldsymbol{\mu}^\star - \boldsymbol{\mu}_i(\boldsymbol{r})\big) - G_i(\boldsymbol{\mu}_i(\boldsymbol{r})) = \frac{1}{2a_i}\big(\boldsymbol{\mu}^\star - \boldsymbol{\mu}_i(\boldsymbol{r})\big)^\mathsf{T} H_T^+(\bar{\boldsymbol{\mu}}_i)\big(\boldsymbol{\mu}^\star - \boldsymbol{\mu}_i(\boldsymbol{r})\big)$$

$$R(\boldsymbol{\mu}^\star) - [\nabla R(\boldsymbol{\mu}_0(\boldsymbol{r}))]^\mathsf{T}\big(\boldsymbol{\mu}^\star - \boldsymbol{\mu}_0(\boldsymbol{r})\big) - R(\boldsymbol{\mu}_0(\boldsymbol{r})) = \frac{1}{2}\big(\boldsymbol{\mu}^\star - \boldsymbol{\mu}_0(\boldsymbol{r})\big)^\mathsf{T} H_C^+(\bar{\boldsymbol{\mu}}_0)\big(\boldsymbol{\mu}^\star - \boldsymbol{\mu}_0(\boldsymbol{r})\big).$$

Now, envoking Proposition A.10 for convex$^+$ functions $T$ and $C$, we obtain that for a sufficiently small $\varepsilon$ we have that if $F(\boldsymbol{r}) - F^\star \le \varepsilon$, then

$$H_T^+(\bar{\boldsymbol{\mu}}_i) \simeq \left(1 \pm \frac{1}{2}\right) H_T^+(\boldsymbol{\mu}^\star) \ , \quad H_C^+(\bar{\boldsymbol{\mu}}_0) \simeq \left(1 \pm \frac{1}{2}\right) H_C^+(\boldsymbol{\mu}^\star) \ .$$

Plugging this back into Eq. (46), we obtain

$$F(\boldsymbol{r}) - F^\star \simeq \left(1 \pm \frac{1}{2}\right)\sum_{i=1}^{N}\frac{1}{2a_i}\big(\boldsymbol{\mu}^\star - \boldsymbol{\mu}_i(\boldsymbol{r})\big)^\mathsf{T} H_T^+(\boldsymbol{\mu}^\star)\big(\boldsymbol{\mu}^\star - \boldsymbol{\mu}_i(\boldsymbol{r})\big)$$

$$+ b\left(1 \pm \frac{1}{2}\right)\frac{1}{2}\big(\boldsymbol{\mu}^\star - \boldsymbol{\mu}_0(\boldsymbol{r})\big)^\mathsf{T} H_C^+(\boldsymbol{\mu}^\star)\big(\boldsymbol{\mu}^\star - \boldsymbol{\mu}_0(\boldsymbol{r})\big) \ . \qquad (47)$$

The theorem now follows by noting that the ranges of matrices $H_T(\boldsymbol{\mu}^\star)$ and $H_C(\boldsymbol{\mu}^\star)$ include all directions $\boldsymbol{\mu} - \boldsymbol{\mu}'$ where $\boldsymbol{\mu}, \boldsymbol{\mu}' \in \mathcal{M}$, because $\mathcal{M} = \operatorname{dom} T^*$ and $\mathcal{M} \subseteq \operatorname{dom} C^*$. □

### D.3 Local Convergence Rate of Block-Coordinate Descent

In this section, we consider general unconstrained convex minimization, but use the same notation as in the rest of the paper. The key difference from the standard analysis of Nesterov [11] is the focus on the local convergence in the neighborhood of the solution, rather than global convergence. This analysis is not specific to our setting, and may be of independent interest.

We consider the optimization problem

$$\min_{\boldsymbol{r}\in\mathbb{R}^{NK}} F(\boldsymbol{r}) \ ,$$

where $F : \mathbb{R}^{NK} \to \mathbb{R}$ is a differentiable convex function bounded below. We are given a set of blocks $\alpha \in \mathcal{A}$, which partition the coordinates $[N] \times [K]$. The *block-coordinate descent* algorithm sets the initial iterate $\boldsymbol{r}^0 = \boldsymbol{0}$, and in each iteration chooses an index $\alpha \in \mathcal{A}$ uniformly at random and fully optimizes the objective over the coordinates in block $\alpha$: given the current iterate $\boldsymbol{r}^t$, the new iterate is $\boldsymbol{r}^{t+1} = \Psi_\alpha(\boldsymbol{r}^t)$ where

$$\Psi_\alpha(\boldsymbol{r}) := \operatorname*{argmin}_{\boldsymbol{r}' \in \boldsymbol{r}+\operatorname{range}(E_\alpha)} F(\boldsymbol{r}') \qquad (48)$$

and $E_\alpha$ is the embedding matrix for the block $\alpha$ as introduced in Appendix D.1.

Nesterov [11] shows that when the optimization objective is strongly convex, it is possible to achieve the linear convergence rate of the form $\mathbb{E}\left[F(\boldsymbol{r}^t)\right] - F^\star \le c\gamma^t$ for some constants $c > 0$ and $\gamma < 1$. While the objective in our setting is not globally strongly convex, it is strongly convex locally, so the

optimization eventually stays within a region where the strong convexity constant is bounded away from zero, yielding a linear convergence rate. Recall from Section 6 that $\gamma_{\text{high}}$ is an upper bound on the local convergence rate of an algorithm if the algorithm with probability 1 reaches an iteration $t_0$ such that for some $c > 0$ and all $t \geq t_0$,

$$\mathbb{E}\left[F(\boldsymbol{r}^t) \mid \boldsymbol{r}^{t_0}\right] - F^\star \leq c\gamma_{\text{high}}^{t-t_0} \ .$$

Also, $\gamma_{\text{low}}$ is a lower bound on the local convergence rate of an algorithm if $\gamma_{\text{high}} \geq \gamma_{\text{low}}$ holds for all upper bounds $\gamma_{\text{high}}$.

Our local convergence-rate results are based on various local properties of $F$, by which we mean properties that hold on some *proper level set* of $F$, in the sense of the following definition:

**Definition D.4** (Level Set). For a given $\lambda \in \mathbb{R}$, the level set $S(F, \lambda)$ of a function $F$ is defined as the set of points $\boldsymbol{r}$ where $F$ is at most $\lambda$:

$$S(F, \lambda) \coloneqq \{\boldsymbol{r} : F(\boldsymbol{r}) \leq \lambda\} \ .$$

The level set $S(F, \lambda)$ is called *proper* if $\lambda > F^\star$.

Our first result requires that the objective be locally strongly convex and smooth in the sense of the following definition:

**Definition D.5** (Strong Convexity and Smoothness for Matrix Seminorms). Let $A$ and $B$ be symmetric positive-semidefinite matrices. We say that a differentiable function $F$ is strongly convex on a set $S$ with respect to $A$ and smooth on $S$ with respect to $B$ if

$$\frac{1}{2}\boldsymbol{\delta}^\intercal A\boldsymbol{\delta} \leq F(\boldsymbol{r} + \boldsymbol{\delta}) - F(\boldsymbol{r}) - \boldsymbol{\delta}^\intercal \nabla F(\boldsymbol{r}) \leq \frac{1}{2}\boldsymbol{\delta}^\intercal B\boldsymbol{\delta}$$

whenever $\boldsymbol{r} \in S$ and $\boldsymbol{r} + \boldsymbol{\delta} \in S$.

To state the theorem, we introduce additional notation. For a matrix $M \in \mathbb{R}^{NK \times NK}$, let $M_{\alpha,\alpha}$ denote the block consisting of rows and columns in $\alpha$, i.e., $M_{\alpha,\alpha} \coloneqq E_\alpha^\intercal M E_\alpha$. And let $\mathcal{D} : \mathbb{R}^{NK \times NK} \to \mathbb{R}^{NK \times NK}$ be the operation of retaining only the block diagonal of a matrix, i.e., $\mathcal{D}(M) \coloneqq \text{diag}_{\alpha \in \mathcal{A}} M_{\alpha,\alpha}$. Recall that $M^+$ denotes the pseudoinverse of $M$. Finally, let $\boldsymbol{g}^t \coloneqq \nabla F(\boldsymbol{r}^t)$ denote the gradient of the objective in the $t$th iteration.

**Theorem D.6.** *Assume that $F$ attains a minimum and let $S \coloneqq S(F, \lambda)$ be a proper level set which satisfies the following conditions:*

1. *$F$ is strongly convex and smooth on $S$ with respect to some positive-semidefinite matrices $A$ and $B$ such that $\mathcal{G}(F) \subseteq \text{range}(A)$, $\mathcal{G}(F) \subseteq \text{range}(B)$.*
2. *There exist non-negative constants $\sigma_{\text{low}} \leq \sigma_{\text{high}} \leq \infty$ and $\ell < \infty$ such that whenever some iterate $\boldsymbol{r}^{t_0}$ lies in $S$, then all the consecutive iterates with $t \geq t_0 + \ell$ satisfy*

$$\sigma_{\text{low}}\mathbb{E}\left[(\boldsymbol{g}^t)^\intercal A^+ \boldsymbol{g}^t \,\Big|\, \boldsymbol{r}^{t_0}\right] \leq \mathbb{E}\left[(\boldsymbol{g}^t)^\intercal \mathcal{D}(B)^+ \boldsymbol{g}^t \,\Big|\, \boldsymbol{r}^{t_0}\right] \ ,$$

$$\mathbb{E}\left[(\boldsymbol{g}^t)^\intercal \mathcal{D}(A)^+ \boldsymbol{g}^t \,\Big|\, \boldsymbol{r}^{t_0}\right] \leq \sigma_{\text{high}}\mathbb{E}\left[(\boldsymbol{g}^t)^\intercal B^+ \boldsymbol{g}^t \,\Big|\, \boldsymbol{r}^{t_0}\right] \ ,$$

*where the expectation is over the random choice of updates by the algorithm.*

*Then if $F^\star < F(\boldsymbol{r}^{t_0}) < \lambda$ and $t \geq t_0 + \ell$, we have*

$$\max\left\{1 - \frac{\sigma_{\text{high}}}{|\mathcal{A}|}, 0\right\} \leq \frac{\mathbb{E}\left[F(\boldsymbol{r}^{t+1}) \mid \boldsymbol{r}^{t_0}\right] - F^\star}{\mathbb{E}\left[F(\boldsymbol{r}^t) \mid \boldsymbol{r}^{t_0}\right] - F^\star} \leq 1 - \frac{\sigma_{\text{low}}}{|\mathcal{A}|} \ .$$

The proof is deferred to a separate subsection below (Appendix D.4). We next show how to apply Theorem D.6 to obtain bounds on local convergence rate. For the lower bounds, we assume that the optimization problem is *non-degenerate* in the sense that the probability of reaching an iterate $\boldsymbol{r}^t$ which attains a minimum is zero; in other words, only approximate solutions are reached in a finite number of steps. This condition holds for the potential function from the main paper under all-security dynamics (ASD), whenever there are at least three agents with distinct beliefs.

**Proposition D.7.** *Let $S_{\text{in}} = S(F, \lambda_{\text{in}})$ and $S_{\text{out}} = S(F, \lambda_{\text{out}})$ be level sets with $F^\star < \lambda_{\text{in}} < \lambda_{\text{out}}$, and assume that the conditions of Theorem D.6 hold for the level set $S_{\text{out}}$ with matrices $A$ and $B$, and non-negative scalars $\sigma_{\text{low}} \le \sigma_{\text{high}} \le \infty$ and $\ell < \infty$. Then for every $\boldsymbol{r}^{t_0} \in S_{\text{in}}$ there exists $c > 0$ such that for all $t \ge t_0$*

$$\mathbb{E}\left[ F(\boldsymbol{r}^t) \mid \boldsymbol{r}^{t_0} \right] - F^\star \le c\gamma_{\text{high}}^{t-t_0} \ ,$$

*where $\gamma_{\text{high}} = 1 - \sigma_{\text{low}}/|\mathcal{A}|$. Furthermore, if the optimization problem is non-degenerate then for every $\boldsymbol{r}^{t_0} \in S_{\text{in}}$, which has a non-zero probability of occurring, and all $t_1 \ge t_0$, there exists $c > 0$ such that for all $t \ge t_1$*

$$\mathbb{E}\left[ F(\boldsymbol{r}^t) \mid \boldsymbol{r}^{t_1} \right] - F^\star \ge c\gamma_{\text{low}}^{t-t_1} \ ,$$

*where $\gamma_{\text{low}} = \max\{1 - \sigma_{\text{high}}/|\mathcal{A}|, 0\}$.*

*Proof.* For the upper bound, if $F(\boldsymbol{r}^{t_0}) = F^\star$, the bound holds for any $c > 0$. Otherwise, set $c := \gamma_{\text{high}}^{-\ell}\left(F(\boldsymbol{r}^{t_0}) - F^\star\right)$. Since the optimization does not increase the objective, this guarantees that the bound holds for $t_0 \le t \le t_0 + \ell$. For $t \ge t_0 + \ell$, Theorem D.6 gives

$$\mathbb{E}\left[ F(\boldsymbol{r}^{t+1}) \mid \boldsymbol{r}^{t_0} \right] - F^\star \le \gamma_{\text{high}} \cdot \left(\mathbb{E}\left[ F(\boldsymbol{r}^t) \mid \boldsymbol{r}^{t_0} \right] - F^\star\right) \ ,$$

and the upper bound now follows by induction.

For the lower bound, note that the non-degeneracy guarantees that $F(\boldsymbol{r}^t) > F^\star$ for all $t \ge t_0$. If $\gamma_{\text{low}} = 0$, the bound holds for any $c$. Next consider the case when $\gamma_{\text{low}} > 0$. Pick $t_1 \ge t_0$ and set

$$c := \min_{t_1 \le t \le t_1 + \ell} \gamma_{\text{low}}^{-\ell}\left(\mathbb{E}\left[ F(\boldsymbol{r}^t) \mid \boldsymbol{r}^{t_1} \right] - F^\star\right) \ ,$$

which is non-zero, because $\gamma_{\text{low}} \in (0, 1]$ and $F(\boldsymbol{r}^t) > F^\star$. This guarantees that the bound holds up to $t = t_1 + \ell$. For larger $t$, it follows by Theorem D.6 and induction. □

When the objective is convex$^+$, we can use the Hessian at the minimum of $F$ to obtain both a quadratic lower and upper bound on $F$, yielding the following bounds on the local convergence rate:

**Theorem D.8.** *Let $F$ be a convex$^+$ function attaining a minimum at $\boldsymbol{r}^\star$. Let $H^\star := \nabla^2 F(\boldsymbol{r}^\star)$ and assume there exists $\lambda > F^\star$ and non-negative constants $\sigma_{\text{low}}$, $\sigma_{\text{high}}$, and $\ell$ such that*

$$\sigma_{\text{low}} \le \frac{\mathbb{E}\left[ (\boldsymbol{g}^t)^\intercal \mathcal{D}(H^\star)^+ \boldsymbol{g}^t \mid \boldsymbol{r}^{t-\ell} \right]}{\mathbb{E}\left[ (\boldsymbol{g}^t)^\intercal (H^\star)^+ \boldsymbol{g}^t \mid \boldsymbol{r}^{t-\ell} \right]} \le \sigma_{\text{high}} \tag{49}$$

*whenever $F(\boldsymbol{r}^{t-\ell}) \le \lambda$, where the expectation is over the choice of updates by the algorithm. Then, for all $\varepsilon > 0$, the local convergence rate is bounded above by $\gamma_{\text{high}} = 1 - \sigma_{\text{low}}/|\mathcal{A}| + \varepsilon$. If the optimization problem is non-degenerate, the local convergence is also bounded below by $\gamma_{\text{low}} = 1 - \sigma_{\text{high}}/|\mathcal{A}|$.*

We defer the proof of Theorem D.8 to a separate subsection (Appendix D.5). The ratio bounded in Eq. (49) can be interpreted as a curvature of the quadratic form $\mathcal{D}(H^\star)^+$ under the norm described by the quadratic form $(H^\star)^+$. Larger values of the ratio (larger curvature) mean faster convergence. We will refer to the bounds $\sigma_{\text{low}}$ and $\sigma_{\text{high}}$ as *lower and upper bounds on local strong convexity* of $F$ (under randomized block-coordinate descent updates). Any non-trivial lower bound, i.e., $\sigma_{\text{low}} > 0$, yields local linear convergence rate since it implies $\gamma_{\text{high}} < 1$, since $\varepsilon$ can be chosen arbitrarily small.

If we know the Hessian $H^\star$, we can obtain a simple lower bound $\sigma_{\text{low}}$ and a linear convergence by considering all directions in the span of gradients (and setting $\ell = 0$). The span of gradients coincides with $\mathcal{G}(F)$, because $\boldsymbol{0}$ is a valid gradient (since $F$ attains a minimum). Thus, we can obtain $\sigma_{\text{low}}$ by the following generalized eigenvalue calculation:

$$\sigma_{\text{low}} = \min_{\boldsymbol{u} \in \mathcal{G}(F)\setminus\{\boldsymbol{0}\}} \frac{\boldsymbol{u}^\intercal \mathcal{D}(H^\star)^+ \boldsymbol{u}}{\boldsymbol{u}^\intercal (H^\star)^+ \boldsymbol{u}} = \lambda_{\min}\left((H^\star)^{1/2}\mathcal{D}(H^\star)^+(H^\star)^{1/2}\right) \ , \tag{50}$$

where $\lambda_{\min}(\cdot)$ is the smallest positive eigenvalue. This is a valid setting of $\sigma_{\text{low}}$, because for all $\boldsymbol{u} \in \mathcal{G}(F)$ we then have $\sigma_{\text{low}}\boldsymbol{u}^\intercal (H^\star)^+ \boldsymbol{u} \le \boldsymbol{u}^\intercal \mathcal{D}(H^\star)^+ \boldsymbol{u}$ and therefore, when $F(\boldsymbol{r}^{t-\ell}) \le \lambda$, also

$$\sigma_{\text{low}}\mathbb{E}\left[ (\boldsymbol{g}^t)^\intercal (H^\star)^+ \boldsymbol{g}^t \mid \boldsymbol{r}^{t-\ell} \right] \le \mathbb{E}\left[ (\boldsymbol{g}^t)^\intercal \mathcal{D}(H^\star)^+ \boldsymbol{g}^t \mid \boldsymbol{r}^{t-\ell} \right].$$

This value of $\sigma_{\text{low}}$ is non-zero, because $\mathcal{G}(F) = \text{range}(H^\star)^+ \subseteq \text{range}\,\mathcal{D}(H^\star)^+$. We pursue this style of analysis for single-securities dynamics (SSD), where we derive $\sigma_{\text{low}}$ using Eq. (50), but do so directly in terms of Hessians of functions $C$ and $T$ rather than the potential $F$. For all-securities dynamics, we consider $\ell > 0$ and take advantage of the averaging effect of expectations in Eq. (49), which yields a tighter lower bound $\sigma_{\text{low}}$ and a non-trivial upper bound $\sigma_{\text{high}}$.

### D.4 Proof of Theorem D.6

*Proof of Theorem D.6.* The vector of partial derivatives of $F$ within block $\alpha$ will be called *partial gradient* and denoted $\nabla_\alpha F(r) \coloneqq E_\alpha^\intercal \nabla F(r)$. Since $F$ has a minimizer, its set of gradients contains a zero, and therefore its gradient space $\mathcal{G}(F)$ coincides with the span of its gradients (see Definition A.5). We also define the set $\mathcal{G}_\alpha(F) \coloneqq E_\alpha^\intercal \mathcal{G}(F)$, which then coincides with the span of partial gradients of $F$. Note that by Proposition D.9, proved below, any positive-semidefinite matrix $M$ satisfies $\operatorname{range}(M) \subseteq \operatorname{range}(\mathcal{D}(M))$. Since $\mathcal{G}(F) \subseteq \operatorname{range}(A)$, we therefore obtain $\mathcal{G}_\alpha(F) \subseteq \operatorname{range}(B_{\alpha,\alpha})$ by the following reasoning:

$$\mathcal{G}_\alpha(F) = E_\alpha^\intercal \mathcal{G}(F) \subseteq E_\alpha^\intercal \operatorname{range}(A) \subseteq E_\alpha^\intercal \operatorname{range}(\mathcal{D}(A)) = \operatorname{range}(A_{\alpha,\alpha}) \ ,$$

and similarly obtain $\mathcal{G}_\alpha(F) \subseteq \operatorname{range}(B_{\alpha,\alpha})$.

Assume $F^\star < F(r^{t_0}) < \lambda$ and let $t \geq t_0 + \ell$. Consider the bundle $r \coloneqq r^t$ and analyze the value of the objective at the next iterate $r^{t+1} = \Psi_\alpha(r)$. First note that the objective does not increase during optimization, so $F(r) < \lambda$ and in particular $r \in S$. From Eq. (48), we then have

$$F(\Psi_\alpha(r)) = \min_{r' \in (r + \operatorname{range}(E_\alpha)) \cap S} F(r') \tag{51}$$

$$\leq \min_{r' \in (r + \operatorname{range}(E_\alpha)) \cap S} \left( F(r) + (r' - r)^\intercal \nabla F(r) + \frac{1}{2}(r' - r)^\intercal B(r' - r) \right) \tag{52}$$

$$= \min_{r' \in (r + \operatorname{range}(E_\alpha))} \left( F(r) + (r' - r)^\intercal \nabla F(r) + \frac{1}{2}(r' - r)^\intercal B(r' - r) \right) \tag{53}$$

$$= \min_{\delta \in \mathbb{R}^{|\alpha|}} \left( F(r) + \delta^\intercal \nabla_\alpha F(r) + \frac{1}{2}\delta^\intercal B_{\alpha,\alpha}\delta \right) \tag{54}$$

$$= F(r) - \frac{1}{2}\big(\nabla_\alpha F(r)\big)^\intercal B_{\alpha,\alpha}^+ \big(\nabla_\alpha F(r)\big) \ . \tag{55}$$

In Eq. (51) it suffices to consider minimization over $S$, because the objective does not increase during the optimization and $r \in S$. In Eq. (52) we use the fact that $F$ is smooth on $S$ with respect to $B$. Eq. (53) follows, because the minimum in Eq. (53) is actually attained in $S$. We show this by contradiction. Let $F'(r')$ denote the function minimized in Eq. (53) and assume that the minimum of $F'$ over $r + \operatorname{range}(E_\alpha)$ is attained at $r' \notin S$. Since $r \in S$, the line connecting $r$ and $r'$ intersects the boundary of $S$ at some point $r''$, where $F(r'') = \lambda$ by continuity of $F$. From the foregoing, we then have

$$F'(r') \leq F'(r) = F(r) < \lambda = F(r'') \leq F'(r'') \ .$$

This however contradicts the convexity of $F'$ along the line connecting $r$ and $r'$, because $r''$ lies between $r$ and $r'$, and thus we should have $F'(r'') \leq \max\{F'(r), F'(r')\}$. This means that the minimum in Eq. (53) is indeed attained somewhere in $S$. In Eq. (54), we make the substitution $r' - r = E_\alpha \delta$, and Eq. (55) then follows by Proposition A.3, because $\nabla_\alpha F(r) \in \mathcal{G}_\alpha(F) \subseteq \operatorname{range}(B_{\alpha,\alpha})$.

Taking expectation over the uniformly random choice of the block $\alpha$, we have

$$F(r) - \mathbb{E}_\alpha\left[F(\Psi_\alpha(r))\right] \geq \frac{1}{2|\mathcal{A}|} \sum_{\alpha \in \mathcal{A}} \big(\nabla_\alpha F(r)\big)^\intercal B_{\alpha,\alpha}^+ \big(\nabla_\alpha F(r)\big)$$

$$= \frac{1}{2|\mathcal{A}|}\big(\nabla F(r)\big)^\intercal \mathcal{D}(B)^+ \big(\nabla F(r)\big) \ . \tag{56}$$

We can also apply the lower bound on $F$ and obtain

$$F(\Psi_\alpha(r)) \geq \min_{r' \in (r + \operatorname{range}(E_\alpha)) \cap S} \left( F(r) + (r' - r)^\intercal \nabla F(r) + \frac{1}{2}(r' - r)^\intercal A(r' - r) \right)$$

$$\geq \min_{r' \in (r + \operatorname{range}(E_\alpha))} \left( F(r) + (r' - r)^\intercal \nabla F(r) + \frac{1}{2}(r' - r)^\intercal A(r' - r) \right) \tag{57}$$

$$= \min_{\delta \in \mathbb{R}^{|\alpha|}} \left( F(r) + \delta^\intercal \nabla_\alpha F(r) + \frac{1}{2}\delta^\intercal A_{\alpha,\alpha}\delta \right)$$

$$= F(r) - \frac{1}{2}\big(\nabla_\alpha F(r)\big)^\intercal A_{\alpha,\alpha}^+ \big(\nabla_\alpha F(r)\big) \ . \tag{58}$$

The steps are analogous as in analyzing the upper bound except for Eq. (57), which is now more straightforward, since the minimum over a larger set cannot lie above a minimum over a smaller set.

Taking expectation over $\alpha$, we thus obtain

$$F(\boldsymbol{r}) - \mathbb{E}_\alpha\left[F(\Psi_\alpha(\boldsymbol{r}))\right] \leq \frac{1}{2|\mathcal{A}|}\big(\nabla F(\boldsymbol{r})\big)^\intercal \mathcal{D}(A)^+\big(\nabla F(\boldsymbol{r})\big) \ . \tag{59}$$

The same reasoning that gave us bounds on $F(\Psi_\alpha(\boldsymbol{r}))$ can be also used to bound the optimal value $F^\star$ by noting that $F^\star = F(\Psi_{\alpha^\star}(\boldsymbol{r}))$ where $\alpha^\star$ is the block containing all the coordinates, i.e., $\alpha^\star = [N] \times [K]$. (Note that $\alpha^\star$ is not necessarily in $\mathcal{A}$.) Eqs. (55) and (58) thus become

$$F^\star \leq F(\boldsymbol{r}) - \frac{1}{2}\big(\nabla F(\boldsymbol{r})\big)^\intercal B^+\big(\nabla F(\boldsymbol{r})\big) \ , \tag{60}$$

$$F^\star \geq F(\boldsymbol{r}) - \frac{1}{2}\big(\nabla F(\boldsymbol{r})\big)^\intercal A^+\big(\nabla F(\boldsymbol{r})\big) \ . \tag{61}$$

To finish the proof, we take the conditional expectations in Eq. (56), use the definition of $\sigma_{\text{low}}$ and take the conditional expectations in Eq. (61) to obtain

$$\mathbb{E}\left[F(\boldsymbol{r}^t) \mid \boldsymbol{r}^{t_0}\right] - \mathbb{E}\left[F(\boldsymbol{r}^{t+1}) \mid \boldsymbol{r}^{t_0}\right] \geq \frac{1}{2|\mathcal{A}|}\mathbb{E}\left[(\boldsymbol{g}^t)^\intercal \mathcal{D}(B)^+\boldsymbol{g}^t \,\Big|\, \boldsymbol{r}^{t_0}\right]$$

$$\geq \frac{1}{2|\mathcal{A}|}\cdot\sigma_{\text{low}}\mathbb{E}\left[(\boldsymbol{g}^t)^\intercal A^+\boldsymbol{g}^t \,\Big|\, \boldsymbol{r}^{t_0}\right]$$

$$\geq \frac{\sigma_{\text{low}}}{|\mathcal{A}|}\left(\mathbb{E}\left[F(\boldsymbol{r}^t) \mid \boldsymbol{r}^{t_0}\right] - F^\star\right) \ . \tag{62}$$

Similarly,

$$\mathbb{E}\left[F(\boldsymbol{r}^t) \mid \boldsymbol{r}^{t_0}\right] - \mathbb{E}\left[F(\boldsymbol{r}^{t+1}) \mid \boldsymbol{r}^{t_0}\right] \leq \frac{\sigma_{\text{high}}}{|\mathcal{A}|}\left(\mathbb{E}\left[F(\boldsymbol{r}^t) \mid \boldsymbol{r}^{t_0}\right] - F^\star\right) \ .$$

Since the objective never increases, we can actually write

$$\mathbb{E}\left[F(\boldsymbol{r}^t) \mid \boldsymbol{r}^{t_0}\right] - \mathbb{E}\left[F(\boldsymbol{r}^{t+1}) \mid \boldsymbol{r}^{t_0}\right] \leq \min\left\{\frac{\sigma_{\text{high}}}{|\mathcal{A}|}\,,\,1\right\}\cdot\left(\mathbb{E}\left[F(\boldsymbol{r}^t) \mid \boldsymbol{r}^{t_0}\right] - F^\star\right) \ . \tag{63}$$

The theorem now follows by rearranging terms in Eqs. (62) and (63). $\qquad\square$

It remains to prove the following proposition, which was used in the proof:

**Proposition D.9.** *For any positive-semidefinite matrix $M$, we have* $\text{range}(M) \subseteq \text{range}(\mathcal{D}(M))$.

*Proof.* For $\alpha \in \mathcal{A}$, let $P_\alpha := E_\alpha E_\alpha^T$ be the projection matrix into $\text{range}\,E_\alpha$, and note that

$$\mathcal{D}(\mathcal{M}) = \text{diag}_\alpha M_{\alpha,\alpha} = \sum_\alpha P_\alpha M P_\alpha \ .$$

Let $\boldsymbol{u} \in \mathbb{R}^{NK}$ and let $x_\alpha = \|M^{1/2}P_\alpha\boldsymbol{u}\|$. Then

$$\begin{aligned}
\boldsymbol{u}^\intercal M \boldsymbol{u} &= \|M^{1/2}\boldsymbol{u}\|^2 = \left\|M^{1/2}\big(\sum_{\alpha\in\mathcal{A}} P_\alpha\boldsymbol{u}\big)\right\|^2 \\
&\leq \big(\sum_{\alpha\in\mathcal{A}}\|M^{1/2}P_\alpha\boldsymbol{u}\|\big)^2 = \big(\sum_{\alpha\in\mathcal{A}} x_\alpha\big)^2 \\
&\leq |\mathcal{A}|\cdot\sum_{\alpha\in\mathcal{A}} x_\alpha^2 \\
&= |\mathcal{A}|\cdot\sum_{\alpha\in\mathcal{A}} \boldsymbol{u}^\intercal P_\alpha M P_\alpha\boldsymbol{u} = |\mathcal{A}|\cdot\big(\boldsymbol{u}^\intercal\mathcal{D}(M)\boldsymbol{u}\big) \ ,
\end{aligned} \tag{64}$$

where in Eq. (64) we used the inequality between the arithmetic and quadratic mean. Thus any $\boldsymbol{u} \in \text{range}(M)$ is also in $\text{range}(\mathcal{D}(M))$. $\qquad\square$

## D.5 Proof of Theorem D.8

This proof builds on several propositions proved in Appendix D.6 below. Let $\varepsilon_{\max} = \sqrt{\lambda - F^\star}$. For $n = 1, 2, \ldots$ define the level sets $S(n) := S(F, F^\star + \varepsilon_{\max}^2/n^2)$. Thus, by Proposition D.11, for some $c > 0$, we have

$$\nabla^2 F(\boldsymbol{r}) \simeq (1 \pm \underbrace{c\varepsilon_{\max}/n}_{c_1})H^\star \quad \text{for all } \boldsymbol{r} \in S(n) \ ,$$

where we have introduced the notation $c_1 := c\varepsilon_{\max}$. By Proposition D.12, if $n > c_1$, then the function $F$ is strongly convex and smooth on $S(n)$ with respect to $A(n) := (1 - c_1/n)H^\star$ and $B(n) := (1 + c_1/n)H^\star$. Thus, $S(n)$, $A(n)$, and $B(n)$ satisfy condition (1) of Theorem D.6 for $n > c_1$.

In the remainder, we only consider the level sets $S(n)$ for $n > c_1$. For each of them define

$$\sigma_{\text{low}}(n) := \frac{n - c_1}{n + c_1}\sigma_{\text{low}} \ , \quad \sigma_{\text{high}}(n) := \frac{n + c_1}{n - c_1}\sigma_{\text{high}} \ .$$

We next argue that they satisfy condition (2) of Theorem D.6. It suffices to verify that condition (2) of Theorem D.6 holds for $t = t_0 + \ell$; let's call this limited variant condition $(2')$. If $(2')$ is satisfied then also (2) is satisfied, because if $\boldsymbol{r}^{t_0} \in S(n)$, then also $\boldsymbol{r}^{t_0+k} \in S(n)$ for any $k \geq 1$, and so we can apply condition $(2')$ at $t = t_0 + k + \ell$ and by taking the conditional expectation we obtain the original condition (2).

To prove that condition (2) holds for $t = t_0 + \ell$, note that $S(n) \subseteq S(1) = S(F, \lambda)$, so Eq. (49) holds whenever $\boldsymbol{r}^{t_0} = \boldsymbol{r}^{t-\ell} \in S(n)$. So, assuming that $n > c_1$ and $\boldsymbol{r}^{t_0} \in S(n)$, we obtain

$$\sigma_{\text{low}}(n) = \frac{n - c_1}{n + c_1}\sigma_{\text{low}}$$

$$\leq \frac{n/(n + c_1)}{n/(n - c_1)} \cdot \frac{\mathbb{E}\left[ (\boldsymbol{g}^t)^\mathsf{T}\mathcal{D}(H^\star)^+\boldsymbol{g}^t \mid \boldsymbol{r}^{t_0} \right]}{\mathbb{E}\left[ (\boldsymbol{g}^t)^\mathsf{T}(H^\star)^+\boldsymbol{g}^t \mid \boldsymbol{r}^{t_0} \right]} = \frac{\mathbb{E}\left[ (\boldsymbol{g}^t)^\mathsf{T}\mathcal{D}(B(n))^+\boldsymbol{g}^t \mid \boldsymbol{r}^{t_0} \right]}{\mathbb{E}\left[ (\boldsymbol{g}^t)^\mathsf{T}A(n)^+\boldsymbol{g}^t \mid \boldsymbol{r}^{t_0} \right]}$$

$$\sigma_{\text{high}}(n) = \frac{n + c_1}{n - c_1}\sigma_{\text{high}}$$

$$\leq \frac{n/(n - c_1)}{n/(n + c_1)} \cdot \frac{\mathbb{E}\left[ (\boldsymbol{g}^t)^\mathsf{T}\mathcal{D}(H^\star)^+\boldsymbol{g}^t \mid \boldsymbol{r}^{t_0} \right]}{\mathbb{E}\left[ (\boldsymbol{g}^t)^\mathsf{T}(H^\star)^+\boldsymbol{g}^t \mid \boldsymbol{r}^{t_0} \right]} = \frac{\mathbb{E}\left[ (\boldsymbol{g}^t)^\mathsf{T}\mathcal{D}(A(n))^+\boldsymbol{g}^t \mid \boldsymbol{r}^{t_0} \right]}{\mathbb{E}\left[ (\boldsymbol{g}^t)^\mathsf{T}B(n)^+\boldsymbol{g}^t \mid \boldsymbol{r}^{t_0} \right]} \ .$$

Now invoking Proposition D.7 with $S_{\text{in}} = S(n+1)$ and $S_{\text{out}} = S(n)$, we obtain that if $\boldsymbol{r}^{t_0} \in S(n+1)$ then there exists $c > 0$ such that for all $t \geq t_0$

$$\mathbb{E}\left[F(\boldsymbol{r}^t) \mid \boldsymbol{r}^{t_0}\right] - F^\star \leq c\gamma_{\text{high}}(n)^{t-t_0} \ ,$$

where $\gamma_{\text{high}}(n) := 1 - \sigma_{\text{low}}(n)/|\mathcal{A}|$. Proposition D.13 now implies that $S(n + 1)$ is reached with probability 1, so $\gamma_{\text{high}}(n)$ is a valid upper bound on the local convergence rate.

Since the optimization starts at a deterministic point $\boldsymbol{r}^0 = \boldsymbol{0}$, and the randomization is among a finite set of choices, there is only a finite set of allocation vectors $\boldsymbol{r}^t$ that can be reached at any given iteration $t$. If the optimization problem is non-degenerate, then none of these allocations (for any $t$) are the actual minimizers of $F$. In that case, Proposition D.7 yields that if $\boldsymbol{r}^{t_0} \in S(n + 1)$, and it has a non-zero probability of occurring, then for all $t_1 \geq t_0$, there exists $c > 0$ such that for all $t \geq t_1$

$$\mathbb{E}\left[F(\boldsymbol{r}^t) \mid \boldsymbol{r}^{t_1}\right] - F^\star \geq c\gamma_{\text{low}}(n)^{t-t_1} \ ,$$

where $\gamma_{\text{low}}(n) := 1 - \sigma_{\text{high}}(n)/|\mathcal{A}|$. Since $S(n + 1)$ is reached with probability 1, this means that any valid upper bound must be greater than $\gamma_{\text{low}}(n)$.

The theorem now follows, because $\gamma_{\text{high}}(n) \to 1 - \sigma_{\text{low}}/|\mathcal{A}|$, and $\gamma_{\text{low}} \to 1 - \sigma_{\text{high}}/|\mathcal{A}|$, as $n \to \infty$.

## D.6 Supporting Propositions for Theorem D.8

Throughout the propositions below, let $F$ be a convex$^+$ function attaining a minimum. Let $P$ be a projection on $\mathcal{G}(F)$. Since $F$ attains a minimum, its gradient set includes zero, and therefore in Proposition A.7 we have $\boldsymbol{a} = \boldsymbol{0}$. This means that

$$F(\boldsymbol{r}) = F(P\boldsymbol{r}) \ ,$$

so we can assume, without loss of generality, that $r^\star \in \mathcal{G}(F)$. Let $H^\star := \nabla^2 F(r^\star)$. Finally, recall that $S(F, \lambda)$ denotes a level set (see Definition D.4).

**Proposition D.10.** *For any $\lambda > F^\star$, $S(F, \lambda) = S_0 + \mathcal{G}(F)^\perp$ where $S_0 \subseteq \mathcal{G}(F)$ is compact.*

*Proof.* Since $F(r) = F(Pr)$, any level set $S$ can be written as $S = S_0 + \mathcal{G}(F)^\perp$, where $S_0 = S \cap \mathcal{G}(F)$. Now if $\lambda > F^\star$ then $S$ is non-empty and closed and hence so is $S_0$. It remains to argue that it is bounded. By convexity$^+$, $F$ is strictly convex on $\mathcal{G}(F)$, so the minimum of $F$ on the sphere $\{r \in \mathcal{G}(F) : \|r - r^\star\| = 1\}$ must be some $\lambda_1 > F^\star$. By convexity, $F(r) - F^\star \geq \lambda_1 \|r - r^\star\|$ for all $r \in \mathcal{G}(F)$. Since $S_0 \subseteq \mathcal{G}(F)$ and $F(r) \leq \lambda$ for $r \in S_0$, the set $S_0$ must be bounded. $\qquad\square$

**Proposition D.11.** *Let $\varepsilon_{\max} > 0$. Then there exists a constant $c > 0$ such that for all $0 < \varepsilon \leq \varepsilon_{\max}$, and all $r \in S(F, F^\star + \varepsilon^2)$, we have*

$$\|Pr - r^\star\| \leq c\varepsilon \ , \nabla^2 F(r) \simeq (1 \pm c\varepsilon) H^\star \ .$$

*Proof.* Let $S := S(F, F^\star + \varepsilon_{\max}^2)$ and $S_0 := S \cap \mathcal{G}(F)$, which is compact by Proposition D.10. Let

$$\sigma_1 := \min_{r \in S_0} \lambda_{\min}(\nabla^2 F(r), P) \ .$$

Note that $\lambda_{\min}(\nabla^2 F(r), P) > 0$ on the compact set $S_0$, and $\nabla^2 F(\cdot)$ and $\lambda_{\min}(\cdot, P)$ are continuous, so $\sigma_1 > 0$. Therefore, $F$ is strictly convex on $S$ with the strict convexity constant $\sigma_1$. Using the fact that $\nabla F(r^\star) = 0$, we then have for any $r \in S$,

$$F(r) = F(Pr) \geq F^\star + \frac{1}{2}\sigma_1 \|Pr - r^\star\|^2 \ .$$

Therefore, if $r \in S(F, F^\star + \varepsilon^2) \subseteq S$ then

$$\|Pr - r^\star\| \leq \varepsilon\sqrt{2/\sigma_1} \ .$$

For the bound on the Hessian, note that since the third derivative of $F$ is continuous, it is upper bounded on $S_0$. Therefore, the Hessian is Lipschitz with some constant $L$ on $S_0$, and so $\|\nabla^2 F(r) - H^\star\| \leq L\|Pr - r^\star\|$. Thus, since $\text{range}(\nabla^2 F(r) - H^\star) \subseteq \mathcal{G}(F)$, we have

$$\nabla^2 F(r) \simeq H^\star \pm L\|Pr - r^\star\| P \ .$$

Since $\sigma_2 P \preceq H^\star$ for some $\sigma_2 > 0$, we obtain

$$\nabla^2 F(r) \simeq (1 \pm L\sigma_2^{-1}\|Pr - r^\star\|) H^\star \ .$$

Thus, for $r \in S(F, F^\star + \varepsilon^2) \subseteq S$ we have

$$\nabla^2 F(r) \simeq (1 \pm \varepsilon L\sigma_2^{-1}\sqrt{2/\sigma_1}) H^\star \ .$$

Setting $c := \max\{\sqrt{2/\sigma_1}, L\sigma_2^{-1}\sqrt{2/\sigma_1}\}$ then proves the proposition. $\qquad\square$

**Proposition D.12.** *Let $S$ be a convex set and $A$ and $B$ positive-semidefinite matrices such that $A \preceq \nabla^2 F(r) \preceq B$ for all $r \in S$. Then $F$ is strongly convex and smooth on $S$ with respect to $A$ and $B$.*

*Proof.* Let $r \in S$, $r + \delta \in S$. Then from the 2nd-order Taylor expansion, we have

$$F(r + \delta) - F(r) - \delta \nabla F(r) = \frac{1}{2}\delta^\top [\nabla^2 F(r')]\delta \ ,$$

where $r' \in S$. The proposition now follows, because $A \preceq [\nabla^2 F(r')] \preceq B$. $\qquad\square$

**Proposition D.13.** *For any $\lambda > F^\star$, with probability 1, the randomized block-coordinate descent algorithm with the objective $F$ reaches an iteration $t$ in which $r^t \in S(F, \lambda)$.*

*Proof.* Since the proposition holds for $\lambda \geq F(\boldsymbol{r}^0)$, consider the case $F^\star < \lambda < F(\boldsymbol{r}^0)$, and in particular assume $F^\star < F(\boldsymbol{r}^0)$. We prove the proposition by applying Theorem D.6.

Let $S := S(F, F(\boldsymbol{r}^0) + 1)$ and $S_0 := S \cap \mathcal{G}(F)$, which is compact by Proposition D.10. Let

$$c_{\min} := \min_{\boldsymbol{r} \in S_0} \lambda_{\min}(\nabla^2 F(\boldsymbol{r}), P) \ , \quad c_{\max} := \max_{\boldsymbol{r} \in S_0} \lambda_{\max}(\nabla^2 F(\boldsymbol{r}), P) \ .$$

Note that $\lambda_{\min}(\nabla^2 F(\boldsymbol{r}), P) > 0$ on the compact set $S_0$, and $\nabla^2 F(\cdot)$ and $\lambda_{\min}(\cdot, P)$ are continuous, so $c_{\min} > 0$. Similarly, since $\lambda_{\max}(\nabla^2 F(\boldsymbol{r}), P) < \infty$ on $S_0$, the continuity yields $c_{\max} < \infty$. Since $\nabla^2 F(\boldsymbol{r}) = \nabla^2 F(P\boldsymbol{r})$, we have that for all $\boldsymbol{r} \in S$

$$c_{\min} P \preceq \nabla^2 F(\boldsymbol{r}) \preceq c_{\max} P \ .$$

Therefore, by Proposition D.12, $F$ is strongly convex and smooth on $S$ with respect to $A := c_{\min} P$ and $B := c_{\max} P$. Let

$$\sigma_{\text{low}} := \lambda_{\min}\big(\mathcal{D}(B)^+, A^+\big) = \lambda_{\min}\big(c_{\max}^{-1}\mathcal{D}(P)^+, c_{\min}^{-1}P\big) = \frac{c_{\min}}{c_{\max}} \cdot \lambda_{\min}(\mathcal{D}(P)^+, P) \ ,$$

which is positive, because $\text{range}(P) \subseteq \text{range}(\mathcal{D}(P))$ by Proposition D.9.

Now, by Proposition D.7, with $S_{\text{in}} = S(F, F(\boldsymbol{r}^0))$, $S_{\text{out}} = S(F, F(\boldsymbol{r}^0) + 1) = S$, and the above matrices $A$ and $B$, the scalar $\sigma_{\text{low}}$, and $\ell = 0$, we obtain that for some constant $c$ and $\gamma := (1 - \sigma_{\text{low}}/|\mathcal{A}|) < 1$,

$$\mathbb{E}\big[F(\boldsymbol{r}^t)\big] - F^\star = \mathbb{E}\big[F(\boldsymbol{r}^t) \,|\, \boldsymbol{r}^0\big] - F^\star \leq c\gamma^t \ .$$

To finish the proof, we will appeal to Borel-Cantelli lemma and show that the algorithm must reach $S(F, \lambda)$ with probability 1. Specifically, note that by the Markov inequality

$$\sum_{t=1}^{\infty} \mathbb{P}\big\{F(\boldsymbol{r}^t) \geq \lambda\big\} = \sum_{t=1}^{\infty} \mathbb{P}\big\{F(\boldsymbol{r}^t) - F^\star \geq \lambda - F^\star\big\}$$

$$\leq \sum_{t=1}^{\infty} \frac{c\gamma^t}{\lambda - F^\star} = \frac{c\gamma}{(1 - \gamma)(\lambda - F^\star)} < \infty \ ,$$

so with probability 1, only a finite number of the events $\{F(\boldsymbol{r}^t) \geq \lambda\}$ will occur; in other words, the level set $S(F, \lambda)$ is reached with probability 1. $\quad\square$

## D.7 Local Convergence of the Market

Throughout this section, we assume that $C$ is convex$^+$, which implies that $F$ is convex$^+$ as well. Our key tool for the analysis of the convergence error of the market is Theorem D.8. Therefore, we need to analyze the gradient and Hessian of $F$. We begin the analysis by deriving explicit expressions for $\nabla F$ and $\nabla^2 F$ using the gradients and Hessians of $T$ and $C$. It will be convenient to do so for trader-level blocks $\nabla_i$ and $\nabla^2_{ij}$.

Given an allocation vector $\boldsymbol{r} \in R^{NK}$, the associated market price (the gradient of the cost) will be denoted $\boldsymbol{\mu}_0(\boldsymbol{r})$ and the gradients of trader potentials will be denoted $\boldsymbol{\mu}_i(\boldsymbol{r})$:

$$\boldsymbol{\mu}_0(\boldsymbol{r}) := \nabla C_b\big(\textstyle\sum_{i=1}^{N} \boldsymbol{r}_i\big) = \nabla C\big(\textstyle\sum_{i=1}^{N} \boldsymbol{r}_i/b\big)$$
$$\boldsymbol{\mu}_i(\boldsymbol{r}) := \nabla F_i(-\boldsymbol{r}_i) = \nabla T(\tilde{\boldsymbol{\theta}}_i - a_i \boldsymbol{r}_i) \qquad \text{for } i \in [N]$$

where $T$ is the log partition function.

The gradient of $F$ is composed of blocks

$$\nabla_i F(\boldsymbol{r}) = -\nabla F_i(-\boldsymbol{r}_i) + \nabla C_b\big(\textstyle\sum_i \boldsymbol{r}_i\big)$$
$$= -\boldsymbol{\mu}_i(\boldsymbol{r}) + \boldsymbol{\mu}_0(\boldsymbol{r}) \ . \tag{65}$$

For the Hessian, first consider $\nabla_{ii} F$:

$$\nabla^2_{ii} F(\boldsymbol{r}) = \nabla^2 F_i(-\boldsymbol{r}_i) + \nabla^2 C_b\big(\textstyle\sum_i \boldsymbol{r}_i\big)$$
$$= a_i \nabla^2 T(\tilde{\boldsymbol{\theta}}_i - a_i \boldsymbol{r}_i) + \frac{1}{b}\nabla^2 C\big((\textstyle\sum_i \boldsymbol{r}_i)/b\big)$$

$$= a_i H_T(\boldsymbol{\mu}_i(\boldsymbol{r})) + \frac{1}{b} H_C(\boldsymbol{\mu}_0(\boldsymbol{r})) \ . \tag{66}$$

Here, recall that for a convex$^+$ function $f$, its Hessian at any given point is only a function of the gradient at that point, which is denoted $H_f$. In Eq. (66), we express $\nabla^2 T$ and $\nabla^2 C$ using the respective functions $H_T$ and $H_C$ and the definitions of $\boldsymbol{\mu}_i$ and $\boldsymbol{\mu}_0$.

For $i \neq j$, the block $\nabla_{ij}^2 F$ is

$$\nabla_{ij}^2 F(\boldsymbol{r}) = \nabla^2 C_b\big(\textstyle\sum_i \boldsymbol{r}_i\big) = \frac{1}{b} \nabla^2 C\Big(\big(\textstyle\sum_i \boldsymbol{r}_i\big)/b\Big)$$

$$= \frac{1}{b} H_C(\boldsymbol{\mu}_0(\boldsymbol{r})) \ . \tag{67}$$

At any optimum $\boldsymbol{r}^\star$, we have $\boldsymbol{\mu}_i(\boldsymbol{r}^\star) = \boldsymbol{\mu}_0(\boldsymbol{r}^\star) = \boldsymbol{\mu}^\star$. Thus, using the Kronecker product notation, the Hessian of $F$ at $\boldsymbol{r}^\star$ can be expressed as

$$\nabla^2 F(\boldsymbol{r}^\star) = D \otimes H_T(\boldsymbol{\mu}^\star) + \frac{\mathbf{1}\mathbf{1}^\mathsf{T}}{b} \otimes H_C(\boldsymbol{\mu}^\star) \ , \tag{68}$$

where $D := \operatorname{diag}_{i \in [N]} a_i$ and $\mathbf{1}$ is the $N$-dimensional all-ones vector

Using local Lipschitz property of $H_T$ and $H_C$ (Proposition A.10) and the fact that $\|\boldsymbol{\mu}^\star - \bar{\boldsymbol{\mu}}\| = O(b)$ (Theorem 5.1), we immediately obtain the following asymptotic expression for $\nabla^2 F(\boldsymbol{r}^\star)$ as $b \to 0$:

**Proposition D.14.** *Let $H^\star := \nabla^2 F(\boldsymbol{r}^\star)$ and $D := \operatorname{diag}_i a_i$. Then*

$$H^\star \simeq (1 \pm O(b)) \left( D \otimes H_T(\bar{\boldsymbol{\mu}}) + (\mathbf{1}\mathbf{1}^\mathsf{T}) \otimes \frac{1}{b} H_C(\bar{\boldsymbol{\mu}}) \right) \ .$$

We next derive asymptotic formulas for matrices $\mathcal{D}(H^\star)^+$ and $(H^\star)^+$, from which we will immediately obtain a lower bound on strong convexity via Eq. (50).

Recall that $\mathcal{A}$ is the decomposition of the coordinates $[N] \times [K]$, but for our two dynamics (ASD and SSD), this decomposition has a special structure. This structure is described by a decomposition $\mathcal{B}$ of $[K]$, which is then applied to each trader, that is $\mathcal{A} = \big\{ \{i\} \times \beta : i \in [N], \beta \in \mathcal{B} \big\}$. For $M \in \mathbb{R}^{K \times K}$, we use the notation $\mathcal{D}_{\mathcal{B}}(M)$ to describe $\operatorname{diag}_{\beta \in \mathcal{B}} M_{\beta\beta}$. For $M \in \mathbb{R}^{NK \times NK}$, we continue writing $\mathcal{D}(M)$ instead of a more explicit $\mathcal{D}_{\mathcal{A}}(M)$.

In stating our results, we use the following shorthands, some of which have been already introduced:

$$H^\star := \nabla^2 F(\boldsymbol{r}^\star), \quad H_T := H_T(\bar{\boldsymbol{\mu}}), \quad H_C := H_C(\bar{\boldsymbol{\mu}}), \quad D := \operatorname{diag}_i a_i, \quad P = I_N - \mathbf{1}\mathbf{1}^\mathsf{T}/N.$$

The matrix $P$ is the projection matrix on the set of centered vectors, i.e., vectors $\boldsymbol{u}$ in $\mathbb{R}^N$ such that $\mathbf{1}^\mathsf{T} \boldsymbol{u} = 0$. With this notation, the pseudoinverses $\mathcal{D}(H^\star)^+$ and $(H^\star)^+$ are characterized in the following theorem:

**Theorem D.15.** *Let $M_1 := I_N \otimes \mathcal{D}_{\mathcal{B}}(H_C)^+$ and $M_2 := (PDP)^+ \otimes H_T^+$. Then, as $b \to 0$,*

$$\mathcal{D}(H^\star)^+ \simeq (1 \pm O(b)) \cdot b M_1 \ , \tag{69}$$

$$(H^\star)^+ \simeq M_2 \pm O(b) M_1 \ . \tag{70}$$

*Local strong convexity is bounded from below by*

$$\sigma_{\text{low}} = b \cdot \lambda_{\min} \left( (M_2^{1/2})^+ M_1 (M_2^{1/2})^+ \right) - O(b^2) \tag{71}$$

$$= b \cdot \lambda_{\min}(PDP) \cdot \lambda_{\min}\left( H_T^{1/2} \mathcal{D}_{\mathcal{B}}(H_C)^+ H_T^{1/2} \right) - O(b^2) \ , \tag{72}$$

*where $\lambda_{\min}(\cdot)$ denotes the smallest positive eigenvalue of a matrix.*

The matrices $M_1$ and $M_2$ in the statement of the theorem do not depend on the liquidity parameter $b$. The matrix $M_2$, which is the dominant part of the Hessian pseudoinverse $(H^\star)^+$, is also independent of the trader dynamics and the cost function. On the other hand, the matrix $M_1$ reflects the cost function and dynamics. The pseudoinverse $\mathcal{D}(H^\star)^+$ approximately equals $b M_1$ as $b \to 0$. The main implication is that $\sigma_{\text{low}} = \Omega(b)$. This yields linear convergence rate bound $\gamma_{\text{high}} = 1 - \Omega(b)$, which suggests worse convergence as $b \to 0$. However, in the absence of a matching lower bound, we cannot conclude that the actual convergence gets worse as $b \to 0$. In Appendix D.7.1, we derive a matching bound $\sigma_{\text{high}} = O(b)$ for ASD. Thus, for ASD, it is not possible to achieve a linear convergence rate better than $1 - \Theta(b)$. (We conjecture similar behavior for SSD.) This means there is a tradeoff between convergence, which slows down as $b \to 0$, and the market-maker bias, which gets smaller.

*Proof of Theorem D.15.* From Proposition D.14, we know that $H^\star \simeq (1 \pm O(b))M$ where

$$M := D \otimes H_T + (\mathbf{1}\mathbf{1}^\mathsf{T}) \otimes \frac{1}{b}H_C \ . \tag{73}$$

Proposition D.14 also implies $\mathcal{D}(H^\star) \simeq (1 \pm O(b))\mathcal{D}(M)$. Thus, by Proposition A.1, we obtain

$$\mathcal{D}(H^\star)^+ \simeq (1 \pm O(b))\mathcal{D}(M)^+ \ , \tag{74}$$

$$(H^\star)^+ \simeq (1 \pm O(b))M^+ \ . \tag{75}$$

The analysis will therefore focus on $M$ and convert to $H^\star$ only in the last step.

We begin with the analysis of $\mathcal{D}(M)$. Specifically, consider the block $M_{\alpha\alpha}$ where $\alpha = \{i\} \times \beta$ for some $\beta \in \mathcal{B}$. From the definition of $M$

$$M_{\alpha\alpha} = \frac{1}{b}H_{C,\beta\beta} + a_i H_{T,\beta\beta}.$$

Since $\mathrm{range}(H_T) = \mathcal{G}(T) \subseteq \mathcal{G}(C) = \mathrm{range}(H_C)$, there is some constant $c_1$ such that $H_T \preceq c_1 H_C$, so we can write

$$M_{\alpha\alpha} \simeq \left(\frac{1}{b} \pm a_i c_1\right) H_{C,\beta\beta}. \tag{76}$$

Setting $c_2 = (\max_i a_i)c_1$, and combining Eq. (76) across all blocks $\alpha = \{i\} \times \beta$, we thus obtain

$$\mathcal{D}(M) \simeq \left(\frac{1}{b} \pm c_2\right)\left(I_N \otimes \mathcal{D}_\mathcal{B}(H_C)\right) \ .$$

Therefore, by Proposition A.1,

$$\mathcal{D}(M)^+ \simeq \frac{b}{1 \pm bc_2}\left(I_N \otimes \mathcal{D}_\mathcal{B}(H_C)^+\right)$$
$$= b(1 \pm O(b))\left(I_N \otimes \mathcal{D}_\mathcal{B}(H_C)^+\right) \ .$$

The bound on $\mathcal{D}(H^\star)^+$ now follows by Eq. (75).

We next bound $(H^\star)^+$ by analyzing $M^+$. First, decompose the matrix $D$ into blocks corresponding to the ranges of the projection matrices $P$ and $I_N - P$. Let $A = PDP$, $B = PD(I_N - P)$ and $X = (I_N - P)D(I_N - P)$. Thus,

$$D = A + B + B^\mathsf{T} + X \ .$$

Using the decomposition of $D$, we can decompose $M$ in order to carry out blockwise inversion:

$$M = D \otimes H_T + \frac{N}{b}(I_N - P) \otimes H_C$$
$$= A \otimes H_T + (B + B^\mathsf{T}) \otimes H_T + \left(X \otimes H_T + \frac{N}{b}(I_N - P) \otimes H_C\right) \ . \tag{77}$$

We first analyze the Schur complement matrix, which appears in the blockwise inverse:

$$S := \left(X \otimes H_T + \frac{N}{b}(I_N - P) \otimes H_C\right) - (B^\mathsf{T} \otimes H_T)(A^+ \otimes H_T^+)(B^\mathsf{T} \otimes H_T)$$
$$= \frac{N}{b}(I_N - P) \otimes H_C + \left(X - B^\mathsf{T}A^+B\right) \otimes H_T \ .$$

As we argued before, $\mathrm{range}(H_T) \subseteq \mathrm{range}(H_C)$. Also $\mathrm{range}(X) \subseteq \mathrm{range}(I_N - P)$ and $\mathrm{range}(B^\mathsf{T}A^+B) \subseteq \mathrm{range}(I_N - P)$, so for some $c_3 > 0$, we have

$$\left(X - B^\mathsf{T}A^+B\right) \otimes H_T \simeq \pm c_3 (I_N - P) \otimes H_C \ ,$$

and therefore

$$S \simeq \left(\frac{N}{b} \pm c_3\right)\left((I_N - P) \otimes H_C\right) \ . \tag{78}$$

We now apply blockwise inversion (Proposition A.2) to Eq. (77), with the bounds of Eq. (78) on the Schur complement to obtain

$$M^+ \simeq A^+ \otimes H_T^+ + \frac{b}{N \pm bc_3}Y$$

where

$$Y := \Big( I_{NK} - (A^+ \otimes H_T^+)(B \otimes H_T) \Big) \big( (I_N - P) \otimes H_C^+ \big) \Big( I_{NK} - (A^+ \otimes H_T^+)(B \otimes H_T) \Big)^\mathsf{T}$$

is a positive-semidefinite matrix. Finally, invoking Eq. (75), we obtain

$$\begin{aligned}
(H^\star)^+ &\simeq (1 \pm O(b)) \big( A^+ \otimes H_T^+ \big) + b(1 \pm O(b)) Y \\
&\simeq \big( A^+ \otimes H_T^+ \big) \pm O(b) \big( A^+ \otimes H_T^+ + Y \big) \\
&\simeq \big( A^+ \otimes H_T^+ \big) \pm O(b) \big( I_N \otimes \mathcal{D}(H_C)^+ \big) \ ,
\end{aligned}$$

where in the last line we used that $\mathrm{range}(A^+ \otimes H_T^+ + Y) \subseteq \mathrm{range}(I_N \otimes H_C^+) \subseteq \mathrm{range}(I_N \otimes \mathcal{D}_\mathcal{B}(H_C)^+)$ because $\mathrm{range}(H_T) \subseteq \mathrm{range}(H_C) \subseteq \mathrm{range}(\mathcal{D}_\mathcal{B}(H_C))$.

Finally, to prove Eq. (72), we use Eq. (50). First, note that $(H^\star)^+ \simeq M_2 \pm O(b) M_1$, so

$$\mathrm{range}(M_2) \subseteq \mathrm{range}(H^\star) = \mathcal{G}(F) \subseteq \mathrm{range}(I_N \otimes H_C) \subseteq \mathrm{range}(M_1) \ .$$

Therefore, if $\boldsymbol{u} \in \mathcal{G}(F) \backslash \mathrm{range}(M_2)$ we have $\boldsymbol{u}^\mathsf{T} M_1 \boldsymbol{u} > 0$, but $\boldsymbol{u}^\mathsf{T} M_2 \boldsymbol{u} = 0$, so

$$\min_{\boldsymbol{u} \in \mathcal{G}(F) \backslash \{\boldsymbol{0}\}} \frac{\boldsymbol{u}^\mathsf{T} M_1 \boldsymbol{u}}{\boldsymbol{u}^\mathsf{T} M_2 \boldsymbol{u}} = \min_{\boldsymbol{u} \in \mathrm{range}(M_2) \backslash \{\boldsymbol{0}\}} \frac{\boldsymbol{u}^\mathsf{T} M_1 \boldsymbol{u}}{\boldsymbol{u}^\mathsf{T} M_2 \boldsymbol{u}} = \lambda_{\min}(M_1, M_2) > 0$$

and so

$$\frac{\boldsymbol{u}^\mathsf{T} M_2 \boldsymbol{u}}{\boldsymbol{u}^\mathsf{T} M_1 \boldsymbol{u}} \le \lambda_{\min}^{-1}(M_1, M_2)$$

for any $\boldsymbol{u} \in \mathcal{G}(F) \backslash \{\boldsymbol{0}\}$.

From the bounds in Eqs. (69) and (70), there exists a constant $c$ such that for $b$ sufficiently small, and for all $\boldsymbol{u} \in \mathcal{G}(F) \backslash \{\boldsymbol{0}\}$,

$$\begin{aligned}
\frac{\boldsymbol{u}^\mathsf{T} \mathcal{D}(H)^+ \boldsymbol{u}}{\boldsymbol{u}^\mathsf{T} H^+ \boldsymbol{u}} &\ge \frac{(1 - cb) \cdot \boldsymbol{u}^\mathsf{T} (b M_1) \boldsymbol{u}}{\boldsymbol{u}^\mathsf{T} M_2 \boldsymbol{u} + cb(\boldsymbol{u}^\mathsf{T} M_1 \boldsymbol{u})} \\
&= b \cdot \frac{1 - cb}{\frac{\boldsymbol{u}^\mathsf{T} M_2 \boldsymbol{u}}{\boldsymbol{u}^\mathsf{T} M_1 \boldsymbol{u}} + cb} \\
&\ge b \cdot \frac{1 - cb}{\lambda_{\min}^{-1}(M_1, M_2) + cb} \\
&\ge b \cdot (\lambda_{\min}(M_1, M_2) - O(b)) \ .
\end{aligned}$$

The bound on $\sigma_{\mathrm{low}}$ now follows by Eq. (50), after noting that

$$\lambda_{\min}(M_1, M_2) = \lambda_{\min} \left( (M_2^{1/2})^+ M_1 (M_2^{1/2})^+ \right) = \lambda_{\min}(PDP) \cdot \lambda_{\min} \left( H_T^{1/2} \mathcal{D}_\mathcal{B}(H_C)^+ H_T^{1/2} \right) \ .$$
$\qquad\qquad\qquad\qquad\qquad\qquad\qquad\qquad\qquad\qquad\qquad\qquad\qquad\qquad\qquad\qquad\qquad\square$

### D.7.1 Tighter Analysis of the All-securities Dynamics

In Theorem D.15 we derived a worst-case bound on the curvature, valid across all possible directions that a gradient can take. In our tighter analysis of ASD, we derive a tighter bound on the expected curvature, exploiting the fact that the updates are chosen uniformly at random. While our analysis only applies to ASD, we conjecture that a similar style of analysis can also work for SSD.

We will index blocks by $i$ rather than $\alpha$, since each block consists of all the coordinates controlled by the trader $i$.

We begin by a detailed analysis of how the block-coordinate updates affect the value of the gradient. Let $\boldsymbol{r}$ by the current iterate. Consider the update $\Psi_i$, which optimizes over the coordinates controlled by trader $i$ (see Eq. 48), and let $\boldsymbol{r}' = \Psi_i(\boldsymbol{r})$ be the new iterate. By the optimality of the update, we have

$$\boldsymbol{\mu}_i(\boldsymbol{r}') = \nabla F_i(-\boldsymbol{r}_i') = \nabla C_b \big( \textstyle\sum_j \boldsymbol{r}_j' \big) = \boldsymbol{\mu}_0(\boldsymbol{r}')$$

and therefore, by Eq. (65), for all $j \in [N]$,

$$\nabla_j F(\boldsymbol{r}') = -\boldsymbol{\mu}_j(\boldsymbol{r}') + \boldsymbol{\mu}_0(\boldsymbol{r}') = -\boldsymbol{\mu}_j(\boldsymbol{r}') + \boldsymbol{\mu}_i(\boldsymbol{r}') \ .$$

Thus, after the first update, the gradient $\nabla F(\boldsymbol{r})$ can be expressed using pairwise differences of $\boldsymbol{\mu}_j(\boldsymbol{r})$. When the update $\Psi_i$ is performed, the value of $\boldsymbol{\mu}_i$ changes, whereas $\boldsymbol{\mu}_j$ for $j \neq i$ is unchanged. The amount of change in $\boldsymbol{\mu}_i$ will be denoted as $\boldsymbol{\delta}_i$:

$$\boldsymbol{\delta}_i(\boldsymbol{r}) := \boldsymbol{\mu}_i(\boldsymbol{r}') - \boldsymbol{\mu}_i(\boldsymbol{r}) \text{ where } \boldsymbol{r}' = \Psi_i(\boldsymbol{r}) \ .$$

This is locally bounded as follows:

**Lemma D.16.** *There exists constants $b_0, c > 0$ such that for every $b \leq b_0$ there exists a proper level set $S$ such that if $\boldsymbol{r} \in S$ then*

$$\|\boldsymbol{\delta}_i(\boldsymbol{r})\| \leq cb \|\boldsymbol{\mu}_i(\boldsymbol{r}) - \boldsymbol{\mu}_0(\boldsymbol{r})\|$$

*for all $i \in [N]$.*

*Proof.* Let $\varepsilon \in (0, 1)$. Since $T$ and $C$ are convex$^+$, by Proposition A.10 it is possible to pick $\delta$ such that

$$H_T(\boldsymbol{\mu}) \simeq (1 \pm \varepsilon) H_T(\bar{\boldsymbol{\mu}}) \ , \qquad H_C(\boldsymbol{\mu}) \simeq (1 \pm \varepsilon) H_C(\bar{\boldsymbol{\mu}})$$

whenever $\|\boldsymbol{\mu} - \bar{\boldsymbol{\mu}}\| \leq \delta$. Pick $b_0$ small enough such that $\|\boldsymbol{\mu}^\star(b; C) - \bar{\boldsymbol{\mu}}\| \leq \delta/2$ for all $b \leq b_0$. Fix some $b \leq b_0$ and pick a level set $S$ such that for all $\boldsymbol{r} \in S$, $\|\boldsymbol{\mu}_i(\boldsymbol{r}) - \boldsymbol{\mu}^\star\| \leq \delta/2$ for all $i$ and $\|\boldsymbol{\mu}_0(\boldsymbol{r}) - \boldsymbol{\mu}^\star\| \leq \delta/2$. Thus, for any $\boldsymbol{r} \in S$, we have

$$H_T(\boldsymbol{\mu}_i(\boldsymbol{r})) \simeq (1 \pm \varepsilon) H_T(\bar{\boldsymbol{\mu}}) \text{ for all } i, \qquad H_C(\boldsymbol{\mu}_0(\boldsymbol{r})) \simeq (1 \pm \varepsilon) H_C(\bar{\boldsymbol{\mu}}) \ .$$

Now let $\boldsymbol{r} \in S$ and let $\boldsymbol{r}' = \Psi_i(\boldsymbol{r})$. By the optimality of $\boldsymbol{r}'$, we know that $\nabla_i F(\boldsymbol{r}') = \boldsymbol{0}$. From the mean value theorem applied to $\nabla_i F$, there exists some $\boldsymbol{q}$ on the line segment connecting $\boldsymbol{r}$ and $\boldsymbol{r}'$ such that

$$\boldsymbol{0} = \nabla_i F(\boldsymbol{r}') = \nabla_i F(\boldsymbol{r}) + \nabla_{ii}^2 F(\boldsymbol{q})(\boldsymbol{r}' - \boldsymbol{r}) \ .$$

Since $\boldsymbol{r}'$ and $\boldsymbol{r}$ differ only in block $i$, we obtain

$$P(\boldsymbol{r}_i' - \boldsymbol{r}_i) = -\left(\nabla_{ii}^2 F(\boldsymbol{q})\right)^+ \nabla_i F(\boldsymbol{r}) \tag{79}$$

where $P$ is the projection on $\operatorname{range}(\nabla_{ii}^2 F(\boldsymbol{q})) = \mathcal{G}(C)$. Now applying the mean value theorem to $\nabla F_i$, we obtain that for some $\boldsymbol{q}'$ on the line segment connecting $\boldsymbol{r}$ and $\boldsymbol{r}'$, we have

$$\begin{aligned}
\boldsymbol{\delta}_i(\boldsymbol{r}) &= \nabla F_i(-\boldsymbol{r}_i') - \nabla F_i(-\boldsymbol{r}_i) \\
&= \nabla^2 F_i(-\boldsymbol{q}_i')(-\boldsymbol{r}_i' + \boldsymbol{r}_i) \\
&= \nabla^2 F_i(-\boldsymbol{q}_i') \left(\nabla_{ii}^2 F(\boldsymbol{q})\right)^+ \nabla_i F(\boldsymbol{r}) \ ,
\end{aligned} \tag{80}$$

where in Eq. (80) we used Eq. (79). Now both $\boldsymbol{r}$ and $\boldsymbol{r}'$ are in the level set $S$ and so is the line segment connecting them, which includes the points $\boldsymbol{q}$ and $\boldsymbol{q}'$. Therefore,

$$\nabla^2 F_i(-\boldsymbol{q}_i') = a_i H_T(\boldsymbol{\mu}_i(\boldsymbol{q}')) \preceq a_i(1 + \varepsilon) H_T(\bar{\boldsymbol{\mu}}) \tag{81}$$

and

$$\begin{aligned}
\nabla_{ii}^2 F(\boldsymbol{q}) &= a_i H_T(\boldsymbol{\mu}_i(\boldsymbol{q})) + \frac{1}{b} H_C(\boldsymbol{\mu}_0(\boldsymbol{q})) \\
&\succeq (1 - \varepsilon) \left( a_i H_T(\bar{\boldsymbol{\mu}}) + \frac{1}{b} H_C(\bar{\boldsymbol{\mu}}) \right) \\
&\succeq \frac{1 - \varepsilon}{b} H_C(\bar{\boldsymbol{\mu}}) \ .
\end{aligned}$$

Thus, also

$$\left(\nabla_{ii}^2 F(\boldsymbol{q})\right)^+ \preceq \frac{b}{1 - \varepsilon} H_C^+(\bar{\boldsymbol{\mu}}) \ . \tag{82}$$

Plugging Eq. (81) and Eq. (82) into Eq. (80), we obtain

$$\|\boldsymbol{\delta}_i(\boldsymbol{r})\| \leq a_i(1 + \varepsilon) \|H_T(\bar{\boldsymbol{\mu}})\| \cdot \frac{b}{1 - \varepsilon} \|H_C^+(\bar{\boldsymbol{\mu}})\| \|\nabla_i F(\boldsymbol{r})\| \ ,$$

finishing the proof, since $\nabla_i F(\boldsymbol{r}) = -\boldsymbol{\mu}_i(\boldsymbol{r}) + \boldsymbol{\mu}_0(\boldsymbol{r})$. $\qquad \square$

Using the lemma, we can now prove bounds for ASD:

**Theorem D.17.** *Consider the all-securities dynamics. Let $M_1' \coloneqq P \otimes H_C^+$ and $M_2 \coloneqq (PDP)^+ \otimes H_T^+$. Then for every sufficiently small b, there exists a proper level set S such that*

$$\mathbb{E}\left[(\boldsymbol{g}^{t+1})^\intercal \mathcal{D}(H^\star)^+ \boldsymbol{g}^{t+1} \mid \boldsymbol{r}^t\right] \simeq (\boldsymbol{g}^t)^\intercal \big((1 \pm O(b)) \cdot 2b M_1'\big)\boldsymbol{g}^t \tag{83}$$

$$\mathbb{E}\left[(\boldsymbol{g}^{t+1})^\intercal (H^\star)^+ \boldsymbol{g}^{t+1} \mid \boldsymbol{r}^t\right] \simeq (\boldsymbol{g}^t)^\intercal \big(M_2 \pm O(b)M_1'\big)\boldsymbol{g}^t \tag{84}$$

*whenever $\boldsymbol{r}^{t-1} \in S$. Local strong convexity is bounded from below and above by*

$$\sigma_{\text{low}} = 2b \cdot \lambda_{\min}(PDP) \cdot \lambda_{\min}\big(H_T^{1/2} H_C^+ H_T^{1/2}\big) - O(b^2)$$

$$\sigma_{\text{high}} = 2b \cdot \lambda_{\max}(PDP) \cdot \lambda_{\max}\big(H_T^{1/2} H_C^+ H_T^{1/2}\big) + O(b^2) \ ,$$

*where $\lambda_{\min}(\cdot)$ and $\lambda_{\max}(\cdot)$ denote the smallest and the largest positive eigenvalue of a matrix.*

As mentioned above, the key consequence of Theorem D.17 is the fact that `ASD` converges at the rate $1 - \Theta(b)$. The key difference from Theorem D.15 is in the expression for $\mathcal{D}(H^\star)^+$. While $\mathcal{D}(H^\star)^+ \approx b(I_N \otimes H_C^+)$, as stated in Theorem D.15, when we take an expectation over an update in a single iteration, the action of $\mathcal{D}(H^\star)^+$ is equivalent to that of the matrix $2b(P \otimes H_C^+)$. Thus, the averaging effect of an expectation is to remove one rank from matrix $I_N$ and replace it by the centering matrix $P = I_N - \mathbf{1}\mathbf{1}^\intercal/N$ (while incurring an extra factor of two). This has two consequences. First, the lower bound $\sigma_{\text{low}}$ is a factor of two larger. Second, we can now obtain a non-trivial upper bound $\sigma_{\text{high}}$, which would not be possible via an analog of Eq. (50), because the range of $\mathcal{D}(H^\star)^+$ is too large.

*Proof of Theorem D.17.* Consider $b_0$ and $c$ from Lemma D.16, and let $b \leq b_0$ and $S$ be the level set from the lemma. Assume that $\boldsymbol{r}^{t-1} \in S$. After the update in the iteration $t-1$, it is guaranteed that $\boldsymbol{\mu}_0(\boldsymbol{r}^t) = \boldsymbol{\mu}_j(\boldsymbol{r}^t)$ for some $j$. We analyze the update in the following iteration, i.e., the iteration $t$. We write $\boldsymbol{r}$ for $\boldsymbol{r}^t$ and $\boldsymbol{r}'$ for $\boldsymbol{r}^{t+1}$. Let $\boldsymbol{g} \coloneqq \nabla F(\boldsymbol{r})$, $\boldsymbol{\mu}_j \coloneqq \boldsymbol{\mu}_j(\boldsymbol{r})$ for $j \in [N]$, $\boldsymbol{\mu}_0 \coloneqq \boldsymbol{\mu}_0(\boldsymbol{r})$, and similarly define $\boldsymbol{g}'$, $\boldsymbol{\mu}_j'$, $\boldsymbol{\mu}_0'$ for the iterate $\boldsymbol{r}'$.

Recall from Eq. (65) that the blocks of the gradient are

$$\boldsymbol{g}_j = \boldsymbol{\mu}_j - \boldsymbol{\mu}_0 \ .$$

Also recall that $P = I_N - \mathbf{1}\mathbf{1}^\intercal/N$. A key role in the analysis will be played by the *centered* gradient vector $\boldsymbol{u} \coloneqq (P \otimes I_K)\boldsymbol{g}$ whose blocks are

$$\boldsymbol{u}_j = \boldsymbol{\mu}_j - \hat{\boldsymbol{\mu}}$$

where $\hat{\boldsymbol{\mu}} \coloneqq (\sum_j \boldsymbol{\mu}_j)/N$ is the average among $\boldsymbol{\mu}_j$. As the final part of the setup, let $\rho = \max_j \|\boldsymbol{\mu}_j - \hat{\boldsymbol{\mu}}\|$, and since $\boldsymbol{\mu}_0 = \boldsymbol{\mu}_j$ for some $j \in [N]$, we also have $\rho \geq \|\boldsymbol{\mu}_0 - \hat{\boldsymbol{\mu}}\|$.

By Theorem D.15, we have

$$\mathcal{D}(H^\star)^+ = \big(1 \pm O(b)\big)bM_1 \ , \qquad (H^\star)^+ = M_2 \pm O(b)M_1 \ ,$$

where

$$M_1 = I_N \otimes H_C^+$$
$$M_2 = (PDP)^+ \otimes H_T^+ \ .$$

For the first part of the theorem (Eqs. 83 and 84), it therefore suffices to show that

$$\mathbb{E}\left[(\boldsymbol{g}')^\intercal M_1 \boldsymbol{g}' \mid \boldsymbol{r}\right] \simeq (1 \pm O(b)) \cdot \boldsymbol{g}^\intercal\big(2P \otimes H_C^+\big)\boldsymbol{g} \tag{85}$$

$$\mathbb{E}\left[(\boldsymbol{g}')^\intercal M_2 \boldsymbol{g}' \mid \boldsymbol{r}\right] \simeq \boldsymbol{g}^\intercal M_2 \boldsymbol{g} \pm O(b) \cdot \boldsymbol{g}^\intercal\big(P \otimes H_C^+\big)\boldsymbol{g} \ . \tag{86}$$

Note that $P \otimes H_C^+ = (P \otimes I_K)M_1(P \otimes I_K)$ and $M_2 = (P \otimes I_K)M_2(P \otimes I_K)$, so Eqs. (85) and (86) are equivalent to

$$\mathbb{E}\left[(\boldsymbol{g}')^\intercal M_1 \boldsymbol{g}' \mid \boldsymbol{r}\right] \simeq (1 \pm O(b)) \cdot \boldsymbol{u}^\intercal (2M_1)\boldsymbol{u} \tag{$M_1$-bound}$$

$$\mathbb{E}\left[(\boldsymbol{g}')^\intercal M_2 \boldsymbol{g}' \mid \boldsymbol{r}\right] \simeq \boldsymbol{u}^\intercal M_2 \boldsymbol{u} \pm O(b) \cdot \boldsymbol{u}^\intercal M_1 \boldsymbol{u} \ , \tag{$M_2$-bound}$$

which is what we will show next.

Assume that the $i$th block is chosen for an update in the iteration $t$ and write $\boldsymbol{\delta}_i$ for $\boldsymbol{\delta}_i(\boldsymbol{r})$. Note that

$$\boldsymbol{\mu}'_0 = \boldsymbol{\mu}'_i = \boldsymbol{\mu}_i + \boldsymbol{\delta}_i \ , \qquad \boldsymbol{\mu}'_j = \begin{cases} \boldsymbol{\mu}_j & \text{if } j \neq i, \\ \boldsymbol{\mu}_i + \boldsymbol{\delta}_i & \text{if } j = i. \end{cases}$$

Thus, from Eq. (65), the blocks $\boldsymbol{g}'_j$ can be written as

$$\boldsymbol{g}'_j = \begin{cases} \boldsymbol{\mu}_i - \boldsymbol{\mu}_j + \boldsymbol{\delta}_i & \text{if } j \neq i, \\ \mathbf{0} & \text{if } j = i. \end{cases}$$

Calculate:

$$
\begin{aligned}
(\boldsymbol{g}')^{\mathsf{T}} M_1 \boldsymbol{g}' &= \sum_j (\boldsymbol{g}'_j)^{\mathsf{T}} H_C^+ \boldsymbol{g}'_j \\
&= \sum_{j \neq i} (\boldsymbol{\mu}_i - \boldsymbol{\mu}_j + \boldsymbol{\delta}_i)^{\mathsf{T}} H_C^+ (\boldsymbol{\mu}_i - \boldsymbol{\mu}_j + \boldsymbol{\delta}_i) \\
&= \sum_j (\boldsymbol{\mu}_i - \boldsymbol{\mu}_j + \boldsymbol{\delta}_i)^{\mathsf{T}} H_C^+ (\boldsymbol{\mu}_i - \boldsymbol{\mu}_j + \boldsymbol{\delta}_i) - \boldsymbol{\delta}_i^{\mathsf{T}} H_C^+ \boldsymbol{\delta}_i \\
&= \sum_j \left[ (\boldsymbol{\mu}_i - \boldsymbol{\mu}_j)^{\mathsf{T}} H_C^+ (\boldsymbol{\mu}_i - \boldsymbol{\mu}_j) + 2\boldsymbol{\delta}_i^{\mathsf{T}} H_C^+ (\boldsymbol{\mu}_i - \boldsymbol{\mu}_j) + \boldsymbol{\delta}_i^{\mathsf{T}} H_C^+ \boldsymbol{\delta}_i \right] - \boldsymbol{\delta}_i^{\mathsf{T}} H_C^+ \boldsymbol{\delta}_i \\
&\simeq \sum_j \left[ (\boldsymbol{\mu}_i - \boldsymbol{\mu}_j)^{\mathsf{T}} H_C^+ (\boldsymbol{\mu}_i - \boldsymbol{\mu}_j) \right] \pm \left[ 4N \|\boldsymbol{\delta}_i\| \|H_C^+\| \rho + N \|H_C^+\| \|\boldsymbol{\delta}_i\|^2 \right] && (87) \\
&\simeq \sum_j \left[ (\boldsymbol{\mu}_i - \boldsymbol{\mu}_j)^{\mathsf{T}} H_C^+ (\boldsymbol{\mu}_i - \boldsymbol{\mu}_j) \right] \pm \left[ 8Nbc \|H_C^+\| \rho^2 + 4Nb^2 c^2 \|H_C^+\| \rho^2 \right] && (88) \\
&\simeq \sum_j \left[ (\boldsymbol{\mu}_i - \boldsymbol{\mu}_j)^{\mathsf{T}} H_C^+ (\boldsymbol{\mu}_i - \boldsymbol{\mu}_j) \right] \pm bc_1 \rho^2 \ . && (89)
\end{aligned}
$$

In Eq. (87), we used the fact that

$$\|\boldsymbol{\mu}_i - \boldsymbol{\mu}_j\| \leq \|\boldsymbol{\mu}_i - \hat{\boldsymbol{\mu}}\| + \|\boldsymbol{\mu}_j - \hat{\boldsymbol{\mu}}\| \leq 2\rho \ .$$

In Eq. (88), we used Lemma D.16, which implies that

$$\|\boldsymbol{\delta}_i\| \leq bc \|\boldsymbol{\mu}_i - \boldsymbol{\mu}_0\| \leq bc \left( \|\boldsymbol{\mu}_i - \hat{\boldsymbol{\mu}}\| + \|\boldsymbol{\mu}_0 - \hat{\boldsymbol{\mu}}\| \right) \leq 2bc\rho \ . \qquad (90)$$

And in Eq. (88), we set $c_1 = 8Nc \|H_C^+\| + 4Nb_0 c^2 \|H_C^+\|$.

Recall that $\boldsymbol{u}_j = \boldsymbol{\mu}_j - \hat{\boldsymbol{\mu}}$, so $\boldsymbol{\mu}_i - \boldsymbol{\mu}_j = \boldsymbol{u}_i - \boldsymbol{u}_j$. Thus, taking expectation over $i$, Eq. (89) yields

$$
\begin{aligned}
\mathbb{E}_i \left[ (\boldsymbol{g}')^{\mathsf{T}} M_1 \boldsymbol{g}' \right] &\simeq \frac{1}{N} \sum_{i=1}^N \sum_{j=1}^N \left[ (\boldsymbol{\mu}_i - \boldsymbol{\mu}_j)^{\mathsf{T}} H_C^+ (\boldsymbol{\mu}_i - \boldsymbol{\mu}_j) \right] \pm bc_1 \rho^2 \\
&= \frac{1}{N} \sum_{i=1}^N \sum_{j=1}^N \left[ (\boldsymbol{u}_i - \boldsymbol{u}_j)^{\mathsf{T}} H_C^+ (\boldsymbol{u}_i - \boldsymbol{u}_j) \right] \pm bc_1 \rho^2 \\
&= \frac{1}{N} \sum_{i=1}^N \sum_{j=1}^N \left[ \boldsymbol{u}_i^{\mathsf{T}} H_C^+ \boldsymbol{u}_i - 2\boldsymbol{u}_i^{\mathsf{T}} H_C^+ \boldsymbol{u}_j + \boldsymbol{u}_j H_C^+ \boldsymbol{u}_j \right] \pm bc_1 \rho^2 \\
&= \sum_{i=1}^N \left[ 2\boldsymbol{u}_i^{\mathsf{T}} H_C^+ \boldsymbol{u}_i \right] \pm bc_1 \rho^2 && (91) \\
&= 2\boldsymbol{u}^{\mathsf{T}} M_1 \boldsymbol{u} \pm bc_1 \rho^2 \ , && (92)
\end{aligned}
$$

where Eq. (91) follows because $\sum_i \boldsymbol{u}_i = \mathbf{0}$. To prove ($M_1$-bound), it remains to upper bound $\rho^2$. Let $\sigma$ be the smallest eigenvalue of $H_C^+$ over $\mathcal{G}(C)$; it must be greater than zero because $\text{range}(H_C) = \mathcal{G}(C)$. We can bound $\rho^2$ as

$$\rho^2 \leq \sum_{i=1}^N \|\boldsymbol{\mu}_i - \hat{\boldsymbol{\mu}}\|^2 \leq \sigma^{-1} \sum_{i=1}^N (\boldsymbol{\mu}_i - \hat{\boldsymbol{\mu}})^{\mathsf{T}} H_C^+ (\boldsymbol{\mu}_i - \hat{\boldsymbol{\mu}}) = \sigma^{-1} \boldsymbol{u}^{\mathsf{T}} M_1 \boldsymbol{u} \ . \qquad (93)$$

Plugging this bound back into Eq. (92) yields ($M_1$-bound).

We next prove ($M_2$-bound). Again, consider an update of the block $i$. Let $\boldsymbol{u}' := (P \otimes I_k)\boldsymbol{g}'$ be the centered version of $\boldsymbol{g}'$. Its blocks are

$$\boldsymbol{u}'_j = \begin{cases} \boldsymbol{u}_j - \delta_i/N & \text{if } j \neq i, \\ \boldsymbol{u}_j - \delta_i/N + \delta_i & \text{if } j = i. \end{cases}$$

Thus,

$$\|\boldsymbol{u}' - \boldsymbol{u}\|^2 = (N-1) \cdot \frac{\|\delta_i\|^2}{N^2} + \left(1 - \frac{1}{N}\right)^2 \|\delta_i\|^2 \leq \|\delta_i\|^2 \leq (2bc\rho)^2$$

where the last inequality follows by Eq. (90). Also, from the definition of $\rho$,

$$\|\boldsymbol{u}\|^2 \leq N\rho^2 .$$

Now, we bound $(\boldsymbol{g}')^\mathsf{T} M_2 \boldsymbol{g}'$. Since $M_2 = (P \otimes I_K)M_2(P \otimes I_K)$, we can write

$$\begin{aligned}
(\boldsymbol{g}')^\mathsf{T} M_2 \boldsymbol{g}' &= (\boldsymbol{u}')^\mathsf{T} M_2 \boldsymbol{u}' \\
&= \left(\boldsymbol{u} + (\boldsymbol{u}' - \boldsymbol{u})\right)^\mathsf{T} M_2 \left(\boldsymbol{u} + (\boldsymbol{u}' - \boldsymbol{u})\right) \\
&\simeq \boldsymbol{u}^\mathsf{T} M_2 \boldsymbol{u} \pm \left[2\|\boldsymbol{u}\|\|M_2\|\|\boldsymbol{u}' - \boldsymbol{u}\| + \|M_2\|\|\boldsymbol{u}' - \boldsymbol{u}\|^2\right] \\
&\simeq \boldsymbol{u}^\mathsf{T} M_2 \boldsymbol{u} \pm \left[4bc\sqrt{N}\|M_2\|\rho^2 + 4b^2c^2\|M_2\|\rho^2\right] && (94) \\
&\simeq \boldsymbol{u}^\mathsf{T} M_2 \boldsymbol{u} \pm bc_2\rho^2 , && (95)
\end{aligned}$$

where in Eq. (94) we used the previously derived bounds and in Eq. (95), we set $c_2 = 4c\sqrt{N}\|M_2\| + 4b_0c^2\|M_2\|$. Finally, using the bound on $\rho^2$ from Eq. (93) in Eq. (95) yields ($M_2$-bound).

It remains to prove the bounds $\sigma_{\text{low}}$ and $\sigma_{\text{high}}$. In particular, we will show that if $\boldsymbol{r}^{t-1} \in S$ then

$$\sigma_{\text{low}} \cdot \mathbb{E}\left[(\boldsymbol{g}')^\mathsf{T}(H^\star)^+ \boldsymbol{g}' \mid \boldsymbol{r}\right] \leq \mathbb{E}\left[(\boldsymbol{g}')^\mathsf{T} \mathcal{D}(H^\star)^+ \boldsymbol{g}' \mid \boldsymbol{r}\right] \leq \sigma_{\text{high}} \cdot \mathbb{E}\left[(\boldsymbol{g}')^\mathsf{T}(H^\star)^+ \boldsymbol{g}' \mid \boldsymbol{r}\right] \quad (96)$$

then taking expectation over $\boldsymbol{r} = \boldsymbol{r}^t$, conditionally on $\boldsymbol{r}^{t-1}$, we also obtain

$$\sigma_{\text{low}} \cdot \mathbb{E}\left[(\boldsymbol{g}')^\mathsf{T}(H^\star)^+ \boldsymbol{g}' \mid \boldsymbol{r}^{t-1}\right] \leq \mathbb{E}\left[(\boldsymbol{g}')^\mathsf{T} \mathcal{D}(H^\star)^+ \boldsymbol{g}' \mid \boldsymbol{r}^{t-1}\right] \leq \sigma_{\text{high}} \cdot \mathbb{E}\left[(\boldsymbol{g}')^\mathsf{T}(H^\star)^+ \boldsymbol{g}' \mid \boldsymbol{r}^{t-1}\right]$$

which will yield the desired conclusion by Theorem D.8 (with the level set $S$ and $\ell = 2$).

To prove Eq. (96), we first apply Eqs. (83) and (84):

$$\begin{aligned}
\underbrace{\frac{\mathbb{E}\left[(\boldsymbol{g}')^\mathsf{T}(H^\star)^+ \boldsymbol{g}' \mid \boldsymbol{r}\right]}{\mathbb{E}\left[(\boldsymbol{g}')^\mathsf{T} \mathcal{D}(H^\star)^+ \boldsymbol{g}' \mid \boldsymbol{r}\right]}}_{=:z} &\simeq (1 \pm O(b)) \cdot \frac{\boldsymbol{g}^\mathsf{T}\left(2b(P \otimes I_K)M_1(P \otimes I_K)\right)\boldsymbol{g}}{\boldsymbol{g}^\mathsf{T}\left(M_2 \pm O(b) \cdot (P \otimes I_K)M_1(P \otimes I_K)\right)\boldsymbol{g}} \\
&= (1 \pm O(b)) \cdot \frac{\boldsymbol{u}^\mathsf{T}\left(2bM_1\right)\boldsymbol{u}}{\boldsymbol{u}^\mathsf{T}\left(M_2 \pm O(b)M_1\right)\boldsymbol{u}} \\
&= (1 \pm O(b)) \cdot \frac{2b}{\frac{\boldsymbol{u}^\mathsf{T} M_2 \boldsymbol{u}}{\boldsymbol{u}^\mathsf{T} M_1 \boldsymbol{u}} \pm O(b)} .
\end{aligned}$$

Now note that the blocks of $\boldsymbol{u}$ take form $\boldsymbol{u}_i = \boldsymbol{\mu}_i - (\sum_j \boldsymbol{\mu}_j)/N$, where $\boldsymbol{\mu}_i \in \mathcal{M} \subseteq \text{range}(H_T^+)$, so $\boldsymbol{u} \in \text{range}(P \otimes H_T^+) = \text{range}(M_2) \subseteq \text{range}(M_1)$. This means that

$$0 < \lambda_{\min}(M_1, M_2) \leq \frac{\boldsymbol{u}^\mathsf{T} M_1 \boldsymbol{u}}{\boldsymbol{u}^\mathsf{T} M_2 \boldsymbol{u}} \leq \lambda_{\max}(M_1, M_2) ,$$

so we can write

$$z \simeq (1 \pm O(b)) \cdot 2b \cdot \frac{\boldsymbol{u}^\mathsf{T} M_1 \boldsymbol{u}}{\boldsymbol{u}^\mathsf{T} M_2 \boldsymbol{u}} ,$$

and thus,

$$2b \cdot \lambda_{\min}(M_1, M_2) - O(b^2) \leq z \leq 2b \cdot \lambda_{\max}(M_1, M_2) + O(b^2) .$$

The theorem now follows, because

$$\begin{aligned}
\lambda_{\min}(M_1, M_2) &= \lambda_{\min}\left((M_2^+)^{1/2} M_1 (M_2^+)^{1/2}\right) \\
&= \lambda_{\min}(PDP) \cdot \lambda_{\min}\left(H_T^{1/2} H_C^+ H_T^{1/2}\right) ,
\end{aligned}$$

and similarly for $\lambda_{\max}(M_1, M_2)$. $\qquad\square$

### D.7.2 Tighter Relationship between Suboptimality and Convergence Error for `ASD`

For `ASD`, it is possible to establish a tighter relationship between the convergence error $\|\boldsymbol{\mu}^t - \boldsymbol{\mu}^\star\|$ and suboptimality $F(\boldsymbol{r}^t) - F^\star$ than we proved in Appendix D.2. Specifically, we showed that $\|\boldsymbol{\mu}^t - \boldsymbol{\mu}^\star\|^2 = O(F(\boldsymbol{r}^t) - F^\star)$ for convex$^+$ $C$. Under `ASD`, we obtain a matching lower bound for the one-step expectation, i.e., $\mathbb{E}\left[\|\boldsymbol{\mu}^{t+1} - \boldsymbol{\mu}^\star\|^2 \mid \boldsymbol{r}^t\right] = \Theta(F(\boldsymbol{r}^t) - F^\star)$.

**Theorem D.18.** *Let $C$ be convex$^+$. Under* `ASD`, *with probability one there exists $t_0$ such that for all $t \geq t_0$,*
$$\mathbb{E}\left[\|\boldsymbol{\mu}^{t+1} - \boldsymbol{\mu}^\star\|^2 \mid \boldsymbol{r}^t\right] \simeq (1 \pm O(b)) \cdot \Theta(F(\boldsymbol{r}^t) - F^\star) \ .$$

*Proof.* Recall that in the notation introduced in Appendix D.2, we have $\boldsymbol{\mu}^t = \boldsymbol{\mu}_0(\boldsymbol{r}^t)$. We use a similar notation as in the proof of Theorem D.17. We write $\boldsymbol{r}$ for $\boldsymbol{r}^t$ and $\boldsymbol{r}'$ for $\boldsymbol{r}^{t+1}$. Let $\boldsymbol{\mu}_j := \boldsymbol{\mu}_j(\boldsymbol{r})$ for $j \in [N]$, $\boldsymbol{\mu}_0 := \boldsymbol{\mu}_0(\boldsymbol{r})$, and similarly define $\boldsymbol{\mu}'_j, \boldsymbol{\mu}'_0$ for the iterate $\boldsymbol{r}'$. Finally, write $\boldsymbol{\delta}_i$ for $\boldsymbol{\delta}_i(\boldsymbol{r})$.

We assume that $t_0 \geq 1$. Let $\rho = \max_j \|\boldsymbol{\mu}_j - \boldsymbol{\mu}^\star\|$, and since $\boldsymbol{\mu}_0 = \boldsymbol{\mu}_j$ for some $j \in [N]$, we also have $\rho \geq \|\boldsymbol{\mu}_0 - \boldsymbol{\mu}^\star\|$. We will use the following loose bound on $\rho^2$:
$$\rho^2 \leq \sum_{i=1}^N \|\boldsymbol{\mu}_i - \boldsymbol{\mu}^\star\|^2 \ .$$

We start with the expression for the suboptimality in Eq. (47) and specialize it to `ASD`:

$$F(\boldsymbol{r}) - F^\star \simeq \left(1 \pm \frac{1}{2}\right) \sum_{i=1}^N \frac{1}{2a_i} \left(\boldsymbol{\mu}^\star - \boldsymbol{\mu}_i\right)^\mathsf{T} H_T^+(\boldsymbol{\mu}^\star) \left(\boldsymbol{\mu}^\star - \boldsymbol{\mu}_i\right)$$

$$+ b\left(1 \pm \frac{1}{2}\right) \frac{1}{2} \left(\boldsymbol{\mu}^\star - \boldsymbol{\mu}_0\right)^\mathsf{T} H_C^+(\boldsymbol{\mu}^\star) \left(\boldsymbol{\mu}^\star - \boldsymbol{\mu}_0\right)$$

$$\simeq (c_1 \pm c_2) \left(\sum_{i=1}^N \|\boldsymbol{\mu}^\star - \boldsymbol{\mu}_i\|^2 + b\|\boldsymbol{\mu}^\star - \boldsymbol{\mu}_0\|^2\right) \tag{97}$$

$$\simeq (c_1 \pm c_2)(1 + b) \sum_{i=1}^N \|\boldsymbol{\mu}^\star - \boldsymbol{\mu}_i\|^2 \ . \tag{98}$$

In Eq. (97), we used the fact that $\lambda_{\min}(H_T^+(\boldsymbol{\mu}^\star), P) > 0$ and $\lambda_{\min}(H_C^+(\boldsymbol{\mu}^\star), P) > 0$, where $P$ is the projection on the linear space parallel to $\mathcal{M}$, which implies existence of constants $0 \leq c_2 < c_1$ such that

$$(c_1 - c_2)P \preceq \left(1 \pm \frac{1}{2}\right) \frac{1}{2a_i} H_T^+(\boldsymbol{\mu}^\star) \preceq (c_1 + c_2)P$$

$$(c_1 - c_2)P \preceq \left(1 \pm \frac{1}{2}\right) \frac{1}{2} H_C^+(\boldsymbol{\mu}^\star) \preceq (c_1 + c_2)P \ .$$

In Eq. (98), we used the upper bound $\|\boldsymbol{\mu}^\star - \boldsymbol{\mu}_0\|^2 \leq \rho^2 \leq \sum_{i=1}^N \|\boldsymbol{\mu}_i - \boldsymbol{\mu}^\star\|^2$. We now similarly bound $\mathbb{E}\left[\|\boldsymbol{\mu}^{t+1} - \boldsymbol{\mu}^\star\|^2 \mid \boldsymbol{r}^t\right]$. Recall that in our notation $\boldsymbol{\mu}^{t+1} = \boldsymbol{\mu}_0(\boldsymbol{r}^{t+1}) = \boldsymbol{\mu}'_0$. We assume that $b$ is sufficiently small, so Lemma D.16 applies, i.e., $b \leq b_0$:

$$\mathbb{E}\left[\|\boldsymbol{\mu}'_0 - \boldsymbol{\mu}^\star\|^2 \mid \boldsymbol{r}\right] = \frac{1}{N} \sum_{i=1}^N \|\boldsymbol{\mu}'_i - \boldsymbol{\mu}^\star\|^2$$

$$= \frac{1}{N} \sum_{i=1}^N \|\boldsymbol{\mu}_i + \boldsymbol{\delta}_i - \boldsymbol{\mu}^\star\|^2$$

$$\simeq \frac{1}{N} \sum_{i=1}^N \left[\|\boldsymbol{\mu}_i - \boldsymbol{\mu}^\star\|^2 \pm 2\|\boldsymbol{\delta}_i\|\rho + \|\boldsymbol{\delta}_i\|^2\right] \tag{99}$$

$$\simeq \frac{1}{N} \sum_{i=1}^N \left[\|\boldsymbol{\mu}_i - \boldsymbol{\mu}^\star\|^2 \pm 4cb\rho^2 + 4c^2 b^2 \rho^2\right] \tag{100}$$

$$\simeq \left( \frac{1}{N} \pm c_3 b \right) \sum_{i=1}^{N} \|\boldsymbol{\mu}_i - \boldsymbol{\mu}^\star\|^2 \ . \tag{101}$$

In Eq. (99), we applied the bound $\|\boldsymbol{\mu}_i - \boldsymbol{\mu}^\star\| \leq \rho$. In Eq. (100), we used Lemma D.16 and applied the triangular inequality to obtain $\|\boldsymbol{\mu}_i - \boldsymbol{\mu}_0\| \leq \|\boldsymbol{\mu}_i - \boldsymbol{\mu}^\star\| + \|\boldsymbol{\mu}_0 - \boldsymbol{\mu}^\star\| \leq 2\rho$. Finally, in Eq. (101), we applied the bound $\rho^2 \leq \sum_{i=1}^{N} \|\boldsymbol{\mu}_i - \boldsymbol{\mu}^\star\|^2$ and set $c_3 = 4c + 4c^2 b_0$. The theorem now follows by combining Eq. (98) and Eq. (101). Note that, as before, we suppress the dependence on $N$ and $a_i$ within the $O(\cdot)$ and $\Theta(\cdot)$ notation. □

### D.7.3   Summary of Local Convergence Results

Here we summarize our local convergence results for ASD and SSD. Recall that $D := \operatorname{diag}_{i \in [N]} a_i$, $P := I_N - \mathbf{1}\mathbf{1}^\mathsf{T}/N$, and $\lambda_{\min}(\cdot)$ and $\lambda_{\max}(\cdot)$ to denote the smallest and the largest positive eigenvalues of a matrix.

**Theorem D.19.** *Assume that $C$ is convex$^+$. Let $H_T := H_T(\bar{\boldsymbol{\mu}})$, $H_C := H_C(\bar{\boldsymbol{\mu}})$, and $D_C$ be the diagonal matrix with the diagonal of $H_C$. For all-securities dynamics, local strong convexity is bounded from below and above by*

$$\sigma_{\text{low}}^{\texttt{ASD}} = 2b \cdot \lambda_{\min}(PDP) \cdot \lambda_{\min}\big( H_T^{1/2} H_C^+ H_T^{1/2} \big) - O(b^2) \ ,$$
$$\sigma_{\text{high}}^{\texttt{ASD}} = 2b \cdot \lambda_{\max}(PDP) \cdot \lambda_{\max}\big( H_T^{1/2} H_C^+ H_T^{1/2} \big) + O(b^2) \ .$$

*For single-security dynamics, local strong convexity is bounded from below by*

$$\sigma_{\text{low}}^{\texttt{SSD}} = b \cdot \lambda_{\min}(PDP) \cdot \lambda_{\min}\big( H_T^{1/2} D_C^+ H_T^{1/2} \big) - O(b^2) \ .$$

*Proof.* The theorem follows immediately from Theorems D.15 and D.17. □

Recall that by Theorem D.8, the bounds on local strong convexity translate into bounds on local convergence rate as $\gamma_{\text{high}} = 1 - \sigma_{\text{low}}/N$ and $\gamma_{\text{low}} = 1 - \sigma_{\text{high}}/N$ for ASD, and $\gamma_{\text{high}} = 1 - \sigma_{\text{low}}/NK$ for SSD.

So, for ASD, Theorem D.19 proves linear convergence with the rate $\gamma = 1 - \Theta(b)$. This means that the convergence gets worse as $b \to 0$, leading to a trade-off with the bias, which decreases as $b \to 0$. Our numerical experiments in Section 7 and Appendix E show that these bounds on the convergence rate are empirically quite tight. Below, we show an example when the two bounds match except for the $O(b^2)$ terms: when all traders have identical risk aversions and the cost function is LMSR.

For SSD, we only present a lower bound on the local strong convexity, which suffices to establish a linear convergence rate. This bound is worse by a factor of two than the bound for ASD. This we believe is only an artifact of a looser analysis and we expect that the reasoning that gave rise to a tighter analysis of ASD can be generalized to SSD. Our experiments in Appendix E also suggest that our SSD analysis is looser than the ASD analysis.

**Example D.20** (Convergence of LMSR under ASD)**.** We next demonstrate the tightness of our bounds for ASD. Consider the setting when $N \geq 2$ and the risk aversion of all traders equals $a$. Then $PDP = aPI_NP = aP$, and since $P$ is a non-zero projection matrix, we obtain $\lambda_{\min}(PDP) = a\lambda_{\min}(P) = a$ and similarly $\lambda_{\max}(PDP) = a$. Furthermore, if the cost is LMSR then $H_C = H_T$, and thus $H_T^{1/2} H_C^+ H_T^{1/2} = I_K - \mathbf{1}\mathbf{1}^\mathsf{T}/K$. Therefore, Theorem 6.2 yields the bounds $\sigma_{\text{low}}^{\texttt{ASD}} = 2ab - O(b^2)$ and $\sigma_{\text{high}}^{\texttt{ASD}} = 2ab + O(b^2)$, whose main asymptotic terms match exactly. Thus, the objective decreases at the rate $\gamma^t$ and the convergence error at the rate $\gamma^{t/2}$ with $\gamma = 1 - 2ab/N + O(b^2)$, and the linear term in $b$ cannot be improved.

### D.8   Proof of Theorem 6.2

The theorem follows immediately from Theorem D.19, because $\gamma_{\text{high}} = 1 - \sigma_{\text{low}}/N$ and $\gamma_{\text{low}} = 1 - \sigma_{\text{high}}/N$ for ASD (by Theorem D.8).

## D.9 Proof of Theorem 6.3

*Proof of Theorem 6.3.* Fix the liquidity $b$ for LMSR and $b' = b/\eta$ for IND, where $\eta \in [1, 2]$ (following Theorem 5.7). We begin by deriving the relationship between the upper and lower bounds on the rate of convergence using Theorem 6.2. We will write $\gamma_{\text{high}}$ and $\gamma_{\text{low}}$ for the convergence rate bounds for LMSR and $\gamma'_{\text{high}}$ and $\gamma'_{\text{low}}$ for IND. We will start with LMSR.

Following the same steps as in Example D.20, we have $\lambda_{\min}(PDP) = \lambda_{\max}(PDP) = a$, and since $H_{\text{LMSR}} = H_T$, we obtain

$$\gamma_{\text{high}} = 1 - 2ab/N + O(b^2) \ , \gamma_{\text{low}} = 1 - 2ab/N - O(b^2) \ .$$

For IND, we have $H_{\text{IND}} = D_T$ where $D_T$ is the diagonal of $H_T$. By Lemma D.21 (see below), we obtain that

$$\lambda_{\min}\big( H_T^{1/2} H_{\text{IND}}^+ H_T^{1/2} \big) = \lambda_{\min}\big( H_T^{1/2} D_T^+ H_T^{1/2} \big) \geq 1$$
$$\lambda_{\max}\big( H_T^{1/2} H_{\text{IND}}^+ H_T^{1/2} \big) = \lambda_{\max}\big( H_T^{1/2} D_T^+ H_T^{1/2} \big) \leq 2 \ .$$

Plugging these expressions, alongside $b' = b/\eta$, into Theorem 6.2, we obtain

$$\gamma'_{\text{high}} \leq 1 - 2ab \, / \, \eta N + O(b^2) \leq 1 - ab/N + O(b^2)$$
$$\gamma'_{\text{low}} \geq 1 - 4ab \, / \, \eta N + O(b^2) \leq 1 - 4ab/N - O(b^2)$$

Next, note the following chain of inequalities which we will use to simplify our analysis. Let $\gamma = 1 - \alpha b + O(b^2)$, and let $t \geq t_0$ and $c > 0$. Then

$$
\begin{aligned}
c\gamma^{t-t_0} &= \exp\big\{ (\log c) + (t - t_0)\log\big(1 - \alpha b + O(b^2)\big) \big\} \\
&\leq \exp\big\{ (\log c) - b(t - t_0)\big(\alpha - O(b)\big) \big\} \\
&\leq \exp\big\{ -bt\big(\alpha - O(b) - \varepsilon_t\big) \big\} \ ,
\end{aligned}
$$

where $\varepsilon_t \to 0$ as $t \to \infty$. Similarly, for $\gamma = 1 - \alpha b - O(b^2)$, we can derive the following lower bound

$$
\begin{aligned}
c\gamma^{t-t_0} &= \exp\big\{ (\log c) + (t - t_0)\log\big(1 - \alpha b - O(b^2)\big) \big\} \\
&\geq \exp\big\{ (\log c) - b(t - t_0)\big(\alpha + O(b)\big) \big\} \\
&\geq \exp\big\{ -bt\big(\alpha + O(b) + \varepsilon_t\big) \big\} \ ,
\end{aligned}
$$

where $\varepsilon_t \to 0$ as $t \to \infty$. In the remainder of the proof, we will write $\varepsilon, \varepsilon', \varepsilon''$ etc., to mean quantities that are $O(b) + \varepsilon_t$ with some $\varepsilon_t \to 0$ as $t \to \infty$.

We can now apply our convergence rate bounds to bound the suboptimality of the potential under both costs and liquidities. For each of the two costs $C$, let $F_C$ and $F_C^\star$ denote the corresponding potential and its optimal value, and $r_C^t$ and $\mu_C^t$ be the corresponding iterates and market prices. By Proposition D.7 and Proposition D.13, we obtain that with probability 1, we will reach an iteration $t_0$ such that for all $t \geq t_0$

$$\exp\big\{ -bt\big(2a/N + \varepsilon\big) \big\} \leq \mathbb{E}\big[ F_{\text{LMSR}}(r_{\text{LMSR}}^t) \mid r_{\text{LMSR}}^{t_0} \big] - F_{\text{LMSR}}^\star \leq \exp\big\{ -bt\big(2a/N - \varepsilon\big) \big\}$$
$$\exp\big\{ -bt\big(4a/N + \varepsilon\big) \big\} \leq \mathbb{E}\big[ F_{\text{LMSR}}(r_{\text{IND}}^t) \mid r_{\text{IND}}^{t_0} \big] - F_{\text{IND}}^\star \leq \exp\big\{ -bt\big(a/N - \varepsilon\big) \big\} \ ,$$

where we used our bounds for $c\gamma^{t-t_0}$. By Theorem D.18, we then also have, for some $\varepsilon'$, and all $t > t_0$,

$$\exp\big\{ -bt\big(2a/N + \varepsilon'\big) \big\} \leq \mathbb{E}\big[ \|\mu_{\text{LMSR}}^t - \mu_{\text{LMSR}}^\star\|^2 \mid r_{\text{LMSR}}^{t_0} \big] \leq \exp\big\{ -bt\big(2a/N - \varepsilon'\big) \big\}$$
$$\exp\big\{ -bt\big(4a/N + \varepsilon'\big) \big\} \leq \mathbb{E}\big[ \|\mu_{\text{IND}}^t - \mu_{\text{IND}}^\star\|^2 \mid r_{\text{IND}}^{t_0} \big] \leq \exp\big\{ -bt\big(a/N - \varepsilon'\big) \big\} \ .$$

Now writing $\mathbb{E}_{t_0}[\cdot]$ instead of $\mathbb{E}\big[ \cdot \mid r_{\text{IND}}^{t_0} \big]$, $\mathbb{E}\big[ \cdot \mid r_{\text{IND}}^{t_0} \big]$, we obtain that for a suitable $\varepsilon''$

$$
\begin{aligned}
\mathbb{E}_{t_0}\Big[ \|\mu_{\text{LMSR}}^{2t(1+\varepsilon'')} - \mu_{\text{LMSR}}^\star\|^2 \Big] &\leq \exp\big\{ -2bt\big(2a/N - \varepsilon'\big)(1 + \varepsilon'') \big\} \\
&= \exp\big\{ -bt\big(4a/N - 2\varepsilon' + 4a\varepsilon''/N - 2\varepsilon'\varepsilon''\big) \big\} \\
&\leq \exp\big\{ -bt\big(4a/N + \varepsilon'\big) \big\}
\end{aligned}
\tag{102}
$$

$$\leq \mathbb{E}_{t_0}\left[\|\boldsymbol{\mu}_{\text{IND}}^t - \boldsymbol{\mu}_{\text{IND}}^\star\|^2\right]$$

$$\leq \exp\left\{-bt\big(a/N - \varepsilon'\big)\right\}$$

$$\leq \exp\left\{-bt\big(a/N + \varepsilon'/2 - a\varepsilon''/N - \varepsilon'\varepsilon''/2\big)\right\} \qquad (103)$$

$$= \exp\left\{-b(t/2)\big(2a/N + \varepsilon'\big)(1 - \varepsilon'')\right\}$$

$$\leq \mathbb{E}_{t_0}\left[\left\|\boldsymbol{\mu}_{\text{LMSR}}^{(t/2)(1-\varepsilon'')} - \boldsymbol{\mu}_{\text{LMSR}}^\star\right\|^2\right] \ .$$

It remains to verify that we can choose a suitable $\varepsilon''$ to guarantee inequalities (102) and (103) for a sufficiently small $b$ and large $t$. One possibility is to set $\varepsilon'' = 3N\varepsilon'/(2a)$, because then (for a sufficiently small $b$ and large $t$), we have

$$\frac{4a\varepsilon''}{N} \geq 3\varepsilon' + 2\varepsilon'\varepsilon'' \quad \Longrightarrow \quad -2\varepsilon' + \frac{4a\varepsilon''}{N} - 2\varepsilon'\varepsilon'' \geq \varepsilon' \quad \Longrightarrow \quad (102),$$

$$\frac{a\varepsilon''}{N} \geq \frac{3}{2}\varepsilon' - \frac{1}{2}\varepsilon'\varepsilon'' \quad \Longrightarrow \quad -\varepsilon' \geq \frac{\varepsilon'}{2} - \frac{a\varepsilon''}{N} - \frac{\varepsilon'\varepsilon''}{2} \quad \Longrightarrow \quad (103). \qquad \square$$

**Lemma D.21.** *Let $\boldsymbol{\mu} \in \mathbb{R}^K$ be a probability vector with non-zero entries, let $H := (\operatorname{diag}_{k\in[K]} \mu_k) - \boldsymbol{\mu}\boldsymbol{\mu}^\intercal$ be the covariance matrix of the associated multinomial distribution, and $D := \operatorname{diag}_{k\in[K]} \mu_k(1-\mu_k)$ be the diagonal matrix consisting of the diagonal of $H$. Then*

$$1 \leq \lambda_{\min}(H^{1/2}D^{-1}H^{1/2}) \quad \text{and} \quad \lambda_{\max}(H^{1/2}D^{-1}H^{1/2}) \leq 2 \ .$$

*Proof.* Let $\mathcal{L} := \operatorname{range}(D^{-1/2}HD^{-1/2})$. To prove the lemma, it suffices to show that for all $\boldsymbol{u} \in \mathcal{L}$

$$\boldsymbol{u}^\intercal\boldsymbol{u} \leq \boldsymbol{u}^\intercal D^{-1/2}HD^{-1/2}\boldsymbol{u} \leq 2\boldsymbol{u}^\intercal\boldsymbol{u} \ . \qquad (104)$$

This will imply that $1 \leq \lambda_{\min}(D^{-1/2}HD^{-1/2})$ and $\lambda_{\max}(D^{-1/2}HD^{-1/2}) \leq 2$. Thus, by Eq. (7), we will also have $1 \leq \lambda_{\min}(H^{1/2}D^{-1}H^{1/2})$ and $\lambda_{\max}(H^{1/2}D^{-1}H^{1/2}) \leq 2$.

Let $\boldsymbol{u} \in \mathcal{L}$ and $\boldsymbol{v} = D^{-1/2}\boldsymbol{u}$. Eq. (104) can be rewritten as

$$\boldsymbol{v}^\intercal D\boldsymbol{v} \leq \boldsymbol{v}^\intercal H\boldsymbol{v} \leq 2\boldsymbol{v}^\intercal D\boldsymbol{v} \ .$$

We will next show that both inequalities hold.

**Part 1: $\boldsymbol{v}^\intercal D\boldsymbol{v} \leq \boldsymbol{v}^\intercal H\boldsymbol{v}$.** We first rewrite the constraint $\boldsymbol{u} \in \mathcal{L}$ in terms of $\boldsymbol{v}$. To start, note that $\boldsymbol{u} \in \mathcal{L} = \operatorname{range}(D^{-1/2}HD^{-1/2})$ iff $\boldsymbol{u} \perp \operatorname{null}(D^{-1/2}HD^{-1/2})$. Next, note that $\boldsymbol{y} \in \operatorname{null}(D^{-1/2}HD^{-1/2})$ iff $D^{-1/2}\boldsymbol{y} \in \operatorname{null}(H)$, which is equivalent to $\boldsymbol{y} \in D^{1/2}\operatorname{null}(H)$. Now, $\operatorname{null}(H) = \{c\mathbf{1} : c \in \mathbb{R}\}$, where $\mathbf{1}$ is the all-ones vector. So,

$$\operatorname{null}(D^{-1/2}HD^{-1/2}) = D^{1/2}\operatorname{null}(H) = \left\{\boldsymbol{y} : y_j = c\sqrt{\mu_j(1-\mu_j)} \text{ for } j \in [K], \text{ for some } c \in \mathbb{R}\right\} \ .$$

Therefore, $\boldsymbol{u} \perp \operatorname{null}(D^{-1/2}HD^{-1/2})$ iff

$$\sum_{j\in[K]} u_j\sqrt{\mu_j(1-\mu_j)} = 0 \ . \qquad (105)$$

Since $\boldsymbol{v} = D^{-1/2}\boldsymbol{u}$, we have $\boldsymbol{u} = D^{1/2}\boldsymbol{v}$, i.e., $u_j = v_j\sqrt{\mu_j(1-\mu_j)}$ for $j \in [K]$. Substituting this expression into Eq. (105) yields

$$\sum_{j\in[K]} v_j\mu_j(1-\mu_j) = 0 \ . \qquad (106)$$

When $K = 1$, then $D = H = \mathbf{0}$, so in this case indeed $\boldsymbol{v}^\intercal D\boldsymbol{v} \leq \boldsymbol{v}^\intercal H\boldsymbol{v}$. Next consider $K > 1$. We will use the following identity for $v_1\mu_1$, implied by Eq. (106):

$$v_1\mu_1 = -\frac{1}{1-\mu_1}\sum_{j\geq 2} v_j\mu_j(1-\mu_j) \ . \qquad (107)$$

We next argue that $\boldsymbol{v}^\intercal H \boldsymbol{v} - \boldsymbol{v}^\intercal D \boldsymbol{v} \geq 0$:

$$\boldsymbol{v}^\intercal H \boldsymbol{v} - \boldsymbol{v}^\intercal D \boldsymbol{v} = -\sum_{\substack{j,k\in[K]\\j\neq k}} \mu_j \mu_k v_j v_k \tag{108}$$

$$= -\sum_{\substack{j,k\geq 2\\j\neq k}} \mu_j \mu_k v_j v_k - 2\sum_{k\geq 2} \mu_1 v_1 \mu_k v_k$$

$$= -\sum_{\substack{j,k\geq 2\\j\neq k}} \mu_j \mu_k v_j v_k + \frac{2}{1-\mu_1} \sum_{j,k\geq 2} \mu_j(1-\mu_j)v_j \mu_k v_k \tag{109}$$

$$= -\sum_{\substack{j,k\geq 2\\j\neq k}} z_j z_k + \frac{2}{1-\mu_1} \sum_{j,k\geq 2} (1-\mu_j)z_j z_k \tag{110}$$

$$= -\sum_{\substack{j,k\geq 2\\j\neq k}} z_j z_k + \frac{1}{1-\mu_1} \sum_{j,k\geq 2} \big((1-\mu_j)+(1-\mu_k)\big)z_j z_k \tag{111}$$

$$= \frac{1}{1-\mu_1}\Big[ \sum_{\substack{j,k\geq 2\\j\neq k}} z_j z_k\big(-(1-\mu_1)+(1-\mu_j)+(1-\mu_k)\big) + \sum_{j\geq 2} 2(1-\mu_j)z_j^2 \Big]$$

$$= \frac{1}{1-\mu_1}\Big[ \sum_{\substack{j,k\geq 2\\j\neq k}} z_j z_k\big((1-\mu_1-\mu_j-\mu_k)+2\mu_1\big) + \sum_{j\geq 2} z_j^2\big(2(1-\mu_1-\mu_j)+2\mu_1\big) \Big]$$

$$= \frac{1}{1-\mu_1}\Big[ 2\mu_1 \sum_{j,k\geq 2} z_j z_k + \sum_{\substack{j,k\geq 2\\j\neq k}} (1-\mu_1-\mu_j-\mu_k)z_j z_k + \sum_{j\geq 2} 2(1-\mu_1-\mu_j)z_j^2 \Big]$$

$$= \frac{1}{1-\mu_1}\Big[ 2\mu_1 \sum_{j,k\geq 2} z_j z_k + \sum_{\substack{j,k\geq 2\\j\neq k}} \sum_{\substack{\ell\geq 2\\\ell\neq j,k}} \mu_\ell z_j z_k + \sum_{j\geq 2}\sum_{\substack{\ell\geq 2\\\ell\neq j}} 2\mu_\ell z_j^2 \Big] \tag{112}$$

$$= \frac{1}{1-\mu_1}\Big[ 2\mu_1 \sum_{j,k\geq 2} z_j z_k + \sum_{j,k\geq 2} \sum_{\substack{\ell\geq 2\\\ell\neq j,k}} \mu_\ell z_j z_k + \sum_{j\geq 2}\sum_{\substack{\ell\geq 2\\\ell\neq j}} \mu_\ell z_j^2 \Big]$$

$$= \frac{1}{1-\mu_1}\Big[ 2\mu_1 \sum_{j,k\geq 2} z_j z_k + \sum_{\ell\geq 2} \mu_\ell \sum_{\substack{j,k\geq 2\\j\neq\ell,k\neq\ell}} z_j z_k + \sum_{j\geq 2}\sum_{\substack{\ell\geq 2\\\ell\neq j}} \mu_\ell z_j^2 \Big]$$

$$= \frac{1}{1-\mu_1}\Big[ 2\mu_1 \Big(\sum_{j\geq 2} z_j\Big)^2 + \sum_{\ell\geq 2} \mu_\ell \Big(\sum_{\substack{j\geq 2\\j\neq\ell}} z_j\Big)^2 + \sum_{\substack{j,\ell\geq 2\\\ell\neq j}} \mu_\ell z_j^2 \Big] \geq 0 \; . \tag{113}$$

In Eq. (108), we use the fact that $D$ is the diagonal of $H$, so the right-hand side only sums over off-diagonal entries of $H$. In Eq. (109), we replaced $\mu_1 v_1$ using Eq. (107). In Eq. (110), introduce the substitution $z_j := \mu_j v_j$. In Eq. (111), we use the fact that $2\sum_{j,k\geq 2}(1-\mu_j)z_j z_k = \sum_{j,k\geq 2}(1-\mu_j)z_j z_k + \sum_{j,k\geq 2}(1-\mu_k)z_j z_k$. In Eq. (112), we use the fact that $\sum_{\ell\in[K]}\mu_\ell = 1$ and so $(1-\mu_1-\mu_j-\mu_k) = \sum_{\ell\neq 1,j,k}\mu_\ell$ and $(1-\mu_1-\mu_j) = \sum_{\ell\neq 1,j}\mu_\ell$. Finally, the inequality in Eq. (113) follows because $\mu_1, \mu_\ell \geq 0$.

**Part 2: $\boldsymbol{v}^\intercal H \boldsymbol{v} \leq 2\boldsymbol{v}^\intercal D \boldsymbol{v}$.** We show by direct calculation that $2\boldsymbol{v}^\intercal D \boldsymbol{v} - \boldsymbol{v}^\intercal H \boldsymbol{v} \geq 0$:

$$2\boldsymbol{v}^\intercal D \boldsymbol{v} - \boldsymbol{v}^\intercal H \boldsymbol{v} \geq 0 = 2\Big[ \sum_{j\in[K]} \mu_j v_j^2 - \sum_{j\in[K]} \mu_j^2 v_j^2 \Big] - \Big[ \sum_{j\in[K]} \mu_j v_j^2 - \sum_{j,k\in[K]} \mu_j \mu_k v_j v_k \Big]$$

$$\tag{114}$$

$$= \sum_{j \in [K]} \mu_j v_j^2 - 2 \sum_{j \in [K]} \mu_j^2 v_j^2 + \sum_{j,k \in [K]} \mu_j \mu_k v_j v_k$$

$$= \sum_{j,k \in [K]} \mu_j \mu_k v_j^2 - 2 \sum_{j \in [K]} \mu_j^2 v_j^2 + \sum_{j,k \in [K]} \mu_j \mu_k v_j v_k \tag{115}$$

$$= \sum_{\substack{j,k \in [K] \\ j \neq k}} \mu_j \mu_k v_j^2 + \sum_{\substack{j,k \in [K] \\ j \neq k}} \mu_j \mu_k v_j v_k$$

$$= \sum_{\substack{j,k \in [K] \\ j \neq k}} \left( \frac{1}{2} \mu_j \mu_k v_j^2 + \frac{1}{2} \mu_j \mu_k v_k^2 + \mu_j \mu_k v_j v_k \right) \tag{116}$$

$$= \sum_{\substack{j,k \in [K] \\ j \neq k}} \frac{1}{2} \mu_j \mu_k \left( v_j + v_k \right)^2 \geq 0 \ . \tag{117}$$

In Eq. (114), we just use the definition of $H$ and $D$. In Eq. (115), we use that $\sum_{k \in [K]} \mu_k = 1$. In Eq. (116), we use that by symmetry $\sum_{j \neq k} \mu_j \mu_k v_j^2 = \sum_{j \neq k} \mu_j \mu_k v_k^2$ and so $\sum_{j \neq k} \mu_j \mu_k v_j^2 = \sum_{j \neq k} \mu_j \mu_k (v_j^2 + v_k^2)/2$. The final inequality in Eq. (117) follows because $\mu_j, \mu_k \geq 0$. $\qquad\square$

## E   Additional Numerical Experiments

In Section 7, we demonstrated that our asymptotic theory closely matches simulations for all-securities dynamics and single-peaked beliefs. Here we include experiments for an additional set of beliefs (uniform beliefs, defined below) and single-securities dynamics (defined in Appendix D.1). Once again, we consider a setting in which there is a complete market over $K = 5$ securities with $N = 10$ traders who have exponential utilities, exponential-family beliefs and risk aversion coefficients $a_i = 1$ for $i \in [N]$. Similar to Section 7, we fix the ground-truth natural parameter $\boldsymbol{\theta}^{\text{true}}$ and independently sample the belief $\tilde{\boldsymbol{\theta}}_i$ of each trader from $\text{Normal}(\boldsymbol{\theta}^{\text{true}}, \sigma^2 I_K)$. We consider two settings of the ground truth and beliefs:

- *Uniform Beliefs*:  All outcomes are equally likely. We set $\boldsymbol{\theta}^{\text{true}} = \mathbf{0}$ and $\sigma = 1$.
- *Single-Peaked Beliefs*:  One outcome is more likely than the others. Here we set $\theta_1^{\text{true}} = \log(1 - \nu \cdot (k - 1))$ and $\theta_k^{\text{true}} = \log(\nu)$ for $k \neq 1$. We use $\nu = 0.02$ and $\sigma = 5$.

Fig. 2 shows the trader beliefs and market-clearing equilibrium prices (calculated via Theorem 4.3) for both settings. Note that in Section 7, we gave results for the case of single-peaked beliefs and all-security dynamics whereas here we present results for all four combinations of belief settings and trader dynamics.

**Bias/convergence tradeoffs**   We first examine the tradeoff that arises between market-maker bias and convergence error as the liquidity parameter of the market is adjusted. Since our main interest is in the effect of the cost function $C$ and liquidity parameter $b$ on error, we ignore the sources of error that do not depend on the choice of cost function, such as the sampling error. Fig. 3 shows the combined bias and convergence error, $\|\bar{\boldsymbol{\mu}} - \boldsymbol{\mu}^t(b; C)\|$, as a function of liquidity, for different beliefs and cost functions under ASD after different numbers of trades have occurred. (Other choices of norm lead to similar results.) Similarly, we give results for SSD in Fig. 4. The minimum point on each curve tells us the optimal value of the liquidity parameter $b$ for the particular setting and number of trades. When the market has not been running long, larger values of $b$ lead to lower error. On the other hand, smaller values of $b$ are preferable as the number of trades grows, with the combined error approaching 0 for small $b$. The combined error of LMSR is similar to that of the sum of independent LMSRs (IND) under uniform beliefs, but LMSR produces lower combined error for single-peaked beliefs.

**Market-maker bias**   We next focus in on the market-maker bias to empirically evaluate our bounds from Section 5. From Theorem 5.6, we know that $\|\boldsymbol{\mu}^\star(b; C) - \bar{\boldsymbol{\mu}}\| \approx b(\bar{a}/N)\|H_T(\bar{\boldsymbol{\mu}})\partial C^*(\bar{\boldsymbol{\mu}})\|$. In Fig. 5, we plot the empirical bias $\|\boldsymbol{\mu}^\star(b; C) - \bar{\boldsymbol{\mu}}\|$ as a function of $b$ for both LMSR and IND under

Figure 2: Two sets of beliefs of the $N = 10$ traders about the $K = 5$ outcomes. The beliefs were sampled once and then fixed in all experiments. The gray bars show the market-clearing equilibrium prices $\bar{\boldsymbol{\mu}}$ as in Definition 3.1 and Theorem 4.3.

Figure 3: The tradeoff between marker-maker bias and convergence error for ASD with different beliefs and cost functions. Solid lines show the total bias and convergence error of LMSR after various numbers of trades, averaged over 20 random trade sequences. Dotted lines show the same for IND.

uniform and single-peaked beliefs, and in each case compare this bias with the approximation implied by the theory. We find that although Theorem 5.6 only gives an asymptotic guarantee as $b \to 0$, the approximation above is fairly accurate even for moderate values of $b$. As Theorem 5.7 shows, the bias of IND is higher than that of LMSR at any fixed value of $b$, but by no more than a factor of two. The difference is greater for single-peaked beliefs than uniform beliefs. Note that the bias is unaffected by the choice of trader dynamics.

**Convergence error**    Finally, we turn to the convergence error. We first show that the local linear convergence rate kicks in very quickly—essentially from the start of trade in our simulations. We then examine the tightness of our bounds on the local convergence rate from Section 6.

From Theorem D.3, we know that $F(\boldsymbol{r}^t) - F^\star$ is an upper bound on $\|\boldsymbol{\mu}^t - \boldsymbol{\mu}^\star\|^2$, and under ASD we also have $F(\boldsymbol{r}^t) - F^\star = \Theta(\|\boldsymbol{\mu}^t - \boldsymbol{\mu}^\star\|^2)$ (by Theorem D.18), where we suppress the implicit dependence on $C$ and $b$. Rather than examining the convergence of prices directly, we examine convergence of the objective, which will be more convenient in the discussion below. Fig. 6 shows the empirical value of $\hat{\mathbb{E}}[F(\boldsymbol{r}^t)] - F^\star$, where the expectation is the empirical average over the 20 random sequences, as a function of the number of trades, plotted on a log scale, for our two belief sets and cost functions under the all-securities trade dynamics. In all settings, the log of convergence error appears linear, matching the local asymptotic analysis in Section 6. In other words, there exist some $\hat{c}$ and $\hat{\gamma}$ such that, empirically, we have for all $t$, $\hat{\mathbb{E}}[F(\boldsymbol{r}^t)] - F^\star \approx \hat{c}\hat{\gamma}^t$.

To examine the tightness of the bounds from Section 6, we dig more deeply into the value of this empirical constant $\hat{\gamma}$, which depends on $C$ and $b$. Since this approximation holds for any sufficiently

Figure 4: The tradeoff between marker-maker bias and convergence error for SSD with different beliefs and cost functions. Solid lines show the total bias and convergence error of LMSR after various numbers of trades, averaged over 20 random trade sequences. Dotted lines show the same for IND.

Figure 5: Market-maker bias as a function of $b$ for different beliefs and cost functions.

large $t$, we can define $\hat{\gamma}$ by choosing some $t_1$ and $t_2$ and setting

$$\hat{\gamma} = \left( \frac{\hat{\mathbb{E}}\left[F(\boldsymbol{r}^{t_2})\right] - F^\star}{\hat{\mathbb{E}}\left[F(\boldsymbol{r}^{t_1})\right] - F^\star} \right)^{1/(t_2 - t_1)}.$$

If $\hat{\gamma}$ is the correct asymptotic convergence rate, then from Theorem D.8, we should have $1 - \sigma_{\text{high}}/N \leq \hat{\gamma} \leq 1 - \sigma_{\text{low}}/N$ for values of $\sigma_{\text{high}}$ and $\sigma_{\text{low}}$ that satisfy Eq. (49), since $|\mathcal{A}| = N$ for ASD. Rearranging terms, we would expect that, for sufficiently large $t_1$ and $t_2$,

$$\sigma_{\text{low}} \leq N \left( 1 - \left( \frac{\hat{\mathbb{E}}\left[F(\boldsymbol{r}^{t_2})\right] - F^\star}{\hat{\mathbb{E}}\left[F(\boldsymbol{r}^{t_1})\right] - F^\star} \right)^{1/(t_2 - t_1)} \right) \leq \sigma_{\text{high}}. \tag{118}$$

We refer to this quantity that is upper and lower bounded by $\sigma_{\text{high}}$ and $\sigma_{\text{low}}$ as the *empirical strong convexity* $\hat{\sigma}$. Note that $\sigma_{\text{low}}$ and $\sigma_{\text{high}}$ implicitly depend on $b$ and $C$.

We can now check how well our theoretical lower and upper bounds on local strong convexity bound the empirical strong convexity $\hat{\sigma}$. In Fig. 7, we plot $\hat{\sigma}$ as a function of $b$ using different values of $t_1$ and $t_2$ and compare it with the asymptotic bounds of $\sigma_{\text{low}}$ and $\sigma_{\text{high}}$ computed as in Theorem D.19, dropping the terms that are $O(b^2)$. We would expect to see $\sigma_{\text{low}} \leq \hat{\sigma} \leq \sigma_{\text{high}}$ as $b$ goes to 0, and indeed this is the case. For LMSR, the values of $\sigma_{\text{high}}$ and $\sigma_{\text{low}}$ coincide, and the empirical values for $\hat{\sigma}$ agree for small $b$.

We now turn to our results for single-security dynamics (SSD). Fig. 8 shows the empirical value of $F(\boldsymbol{r}^t) - F^\star$, averaged over 20 random sequences of trade, as a function of the number of trades, plotted on a log scale, for our two belief sets and cost functions under SSD. The plots show the convergence error for LMSR and IND right on top of each other, suggesting that the main asymptotic

Figure 6: Convergence in the objective value for various trader beliefs, cost functions, and liquidity parameters under ASD. Solid lines show the log error in objective for `LMSR`, dotted lines for `IND`.

Figure 7: The empirical strong convexity from Eq. (118) under `ASD` for various values of $t_1, t_2$ represented as dots, and asymptotic bounds for $\sigma_{\text{low}}$ and $\sigma_{\text{high}}$ from Theorem D.19 (ignoring $O(b^2)$ terms) represented as lines.

term is driven by the diagonal of $H_C(\bar{\boldsymbol{\mu}})$, which is the same for both costs, and which appears in the lower bound of Theorem D.19 (with a multiplier that could be possibly improved). Similar to `ASD`, we also evaluate the empirical strong convexity $\hat{\sigma}$. In this case, we only have access to a lower bound (Theorem D.19), which our plots show to be a valid albeit a somewhat loose bound. All the bounds that we used in the `ASD` and `SSD` strong convexity plots are summarized in Table 1.

Figure 8: Convergence in the objective value for various trader beliefs, cost functions, and liquidity parameters under SSD. Solid lines show the log error in objective for LMSR, dotted lines for IND; the IND and LMSR lines are right on top of each other.

Figure 9: The empirical strong convexity from Eq. (118) under SSD for various values of $t_1, t_2$ represented as dots, and asymptotic bound for $\sigma_{\text{low}}$ from Theorem D.19 (ignoring the $O(b^2)$ term) represented as the black solid line.

Table 1: The bounds $\sigma_{\text{low}}$ and $\sigma_{\text{high}}$ in the various cases we consider in our experiments, computed using Theorem D.19.

| Beliefs | Dynamics | $C$ | $\sigma_{\text{low}}$ | $\sigma_{\text{high}}$ |
|---|---|---|---|---|
| Uniform Beliefs | ASD | LMSR | $2b$ | $2b$ |
| | | IND | $2.31b$ | $2.78b$ |
| | SSD | LMSR | $1.01b$ | — |
| | | IND | $1.01b$ | — |
| Single-Peaked Beliefs | ASD | LMSR | $2b$ | $2b$ |
| | | IND | $2.03b$ | $3.78b$ |
| | SSD | LMSR | $1.01b$ | — |
| | | IND | $1.01b$ | — |