[Reviews · NeurIPS 2017]

Reviewer 1



This paper decomposes prediction market forecasting errors into three components, the sampling error, market-maker bias and convergence error. This decomposition provides a meaningful way to compare different cost functions and liquidity levels. Empirical results are presented using a comparison between the Logarithmic Market Scoring Rule and the Sum of Independent LMSRs with a trader model that uses an exponential cost function and varying liquidity parameter. I enjoyed reading the paper. I didn't find anything fundamentally wrong with it. I understand the paper is demonstrating the liquidity parameter, but I think it would have been more interesting to see a variation in the cost function. Further, I think the paper would have been more interesting under a more heterogeneous pool of risk aversion values and a changing number of traders/securities. The model also ignores liquidity gaps that could be the result of market impact or market maker utilities. Some comments: - Why are only 20 random trade sequences considered? - How are you random trade sequences generated? - How many trades are considered in the two left plots in Fig 1? - Explain why you set the risk aversion parameter to 1. - You assume liquidity is without gaps, so I'm wondering how your model responds to gaps or pockets of illiquidity. - What happens to your error terms when liquidity gaps are randomly inserted? Please review your manuscript for minor mistakes that reduce readability, such as, L344 random sequences of trade -> random trade sequences

Reviewer 2



n/a

Reviewer 3



The paper looks at prediction markets. It proposes a decomposition of the forecast error into 3 terms: the sampling error (due to noisy estimates), the market-maker bias (due to the cost function) and the convergence error (due to finite trades). The format definitions of these errors are based on 2 idealized notions of equilibrium, the market clearing equilibrium and the market-maker equilibrium. The results are essentially bounds on the 3 different errors. These bounds allow comparing the trade-offs of different cost functions, which is something not many results of the literature focus on. Some numerical experiments show that the bounds are tight. I find the results interesting as they provide a tool to bound errors for different cost functions and might potentially be used to decide which is the best cost function for a given situation. However, the market-clearing and market-marker equiibria are unique only under the exponential utility function used by the authors, which is problematic: under another utility, what would happen? as I understand it, the error terms would not be well defined. What would happen with the bounds and trade-off discussed? Other comments: - the intuition that connects the market-clearing and market-marker equiibria to the different error terms is not perfectly clear - it would be good that the authors discuss more the implication of their results for the choice of a cost function

Reviewer 4



Summary: This paper presents several results for analyzing the error of forecasts made using prediction markets. These results can be used to compare different choices for the cost function and give principled guidance about setting the liquidity parameter b when designing markets. The first key idea is to decompose the error into three components: sampling error, market-maker bias (at equilibrium), and convergence error. Intuitively, sampling error is due to the empirical distribution of traders' beliefs not matching the unknown true distribution over states, the market-maker bias is the asymptotic error due to the market's rules influencing trader behavior, and the convergence error is the distance from the market's prices to the equilibrium prices after a fixed number of trades. To make this decomposition precise, the authors introduce two notions of equilibria for the market: market-clearing and market-maker equilibria. Intuitively, a market-clearing equilibrium is a set of security prices and allocations of securities and cash to the N traders such that no trader has incentive to buy or sell securities at the given prices (in this case, the traders need not interact via the market!). Similarly, a market-maker equilibrium is a set of prices and allocations of securities and cash to the N traders that can be achieved by trading through the market and for which no trader has incentive to buy or sell securities through the market at the current prices. With this, the sampling error is the difference between the true prices and market-clearing equilibrium prices, the market-maker bias is the difference between the market-clearing and market-maker equilibrium prices, and the convergence error is the difference between the market-maker equilibrium prices and the prices actually observed in the market after t trades. Only the market-maker bias and the convergence error depend on the design of the market, so the authors focus on understanding the influence of the market's cost function and liquidity parameter on these two quantities. For the remainder of the paper, the authors focus on the exponential trader model introduced by Abernethy et al. (EC 2014), where each trader has an exponential utility function and their beliefs are exponential family distributions. The authors then recast prior results from Abernethy et al. (EC 2014) and Frongillo et al. (NIPS 2015) to show that in this case, the market-clearing and market-maker equilibria are unique and have prices/allocations that can be expressed as the solutions to clean optimization problems. From these results, it is possible to bound the market-maker bias by O(b), where b is the liquidity parameter of the market. Unfortunately, this upper bound is somewhat pessimistic and, since it does not have an accompanying lower bound, it can't be used to compare different cost functions for the market. The authors then present a more refined local analysis, under the assumption that the cost function satisfies a condition the authors call convex+. These results give a precise characterization the relationship between the market-clearing and market-maker equilibrium prices, up to second order terms. To study the convergence error, the authors consider the following market protocol: on each round the market chooses a trader uniformly at random and that trader buys the bundle of securities that optimizes their utility given the current state of the market and their own cash/security allocations. Frongillo et al. (NIPS 2015) showed that under this model, the updates to the vector of security allocations to the N traders exactly corresponds to randomized block-coordinate descent on a particular potential function. The standard analysis of randomized block coordinate descent shows that the suboptimality decreases at a rate O(gamma^t) for some gamma less than one. Under the assumption that the market cost function is convex+, this paper shows both upper and lower bounds on the convergence rate, as well as gives an explicit characterization of gamma on the cost function C and the liquidity parameter b. We are able to combine the two above results to compare different cost functions and liquidity parameters. In particular, we can compare two different cost functions by setting their liquidity parameter to ensure the market-maker bias of the two cost functions is equal, and then comparing the rates of the two resulting convergence errors. The authors apply this analysis to two cost functions, the Logarithmic Market Scoring Rule (LMSR) and Sum of Independent LMSRs (IND), showing that for the same asymptotic bias, IND requires at least 1/2 and no more than 2 times as many trades as LSMR to achieve the same error. Their theoretical findings are also verified empirically. Comments: The results in the paper are interesting and nontrivial. One limitation, however, is that they seem to only apply under the exponential trader model, where each trader has an exponential family belief over outcomes and a risk-averse exponential utility function. If the results are more generally applicable (e.g., maybe for other trader models we can have similar characterizations and analysis of the two market equilibria), it may be helpful to include some discussion of this in the paper. Also, is the exponential trader model inspired by the behavior of real traders in prediction markets? In the comparison on LMSR and IND, the authors choose the liquidity parameters of the two markets so that the market-maker bias is equal. An alternative comparison that might be interesting would be to fix a time horizon of T trades and choose the market liquidity parameter to give the most accurate forecasts after T trades. It's also a bit surprising that the analysis seems to suggest that LMSR and IND are roughly comparable in terms of the number of trades required to get the same level of accuracy. This result seems to contradict the intuition that maintaining coherence between the security prices has informational advantages. Is it the case that IND requires the liquidity parameter b to be set larger and this increases the potential loss for the market? It may be easier to understand the definition of market-clearing and market-maker equilibria if the orders of their definition were reversed. The market-clearing equilibrium definition comes immediately after the definition of the market, but does not require that the traders buy/sell securities through the market, and this took me some time to understand. The new upper and lower bounds for randomized block coordinate descent for convex+ functions also sound interesting, and stating them in their general form might help the paper appeal to a wider audience.